# The solute carrier superfamily interactome

Fabian Frommelt [1,7], Rene Ladurner [1,7], Ulrich Goldmann [1], Gernot Wolf [1], Alvaro Ingles-Prieto [1], Eva Lineiro-Retes [1], Zuzana Gelová [1], Ann-Katrin Hopp [1], Eirini Christodoulaki [1], Shao Thing Teoh [1], Philipp Leippe [1], Brianda L Santini [1], Manuele Rebsamen [1], Sabrina Lindinger [1], Iciar Serrano [1], Svenja Onstein [1], Christoph Klimek [1], Barbara Barbosa [1], Anastasiia Pantielieieva [1], Vojtech Dvorak [1], Thomas J Hannich [1], Julian Schoenbett [2], Gilles Sansig [2], Tamara A M Mocking [3], Jasper F Ooms [3], Adriaan P IJzerman [3], Laura H Heitman [3], Peter Sykacek [4], Juergen Reinhardt [2], André C Müller [1], Tabea Wiedmer [1] & Giulio Superti-Furga [1,5,6 ✉]

## Abstract

**Solute carrier (SLC) transporters form a protein superfamily that enables transmembrane transport of diverse substrates including nutrients, ions and drugs. There are about 450 different SLCs, residing in a variety of subcellular membranes. Loss-of-function of an unusually high proportion of SLC transporters is genetically associated with a plethora of human diseases, making SLCs a rapidly emerging but challenging drug target class. Knowledge of their protein environment may elucidate the molecular basis for their functional integration with metabolic and cellular pathways and help conceive pharmacological interventions based on modulating proteostatic regulation. We aimed at obtaining a global survey of the SLC-protein interaction landscape and mapped the protein–protein interactions of 396 SLCs by interaction proteomics. We employed a functional assessment based on RNA interference of interactors in combination with measurement of protein stability and localization. As an example, we detail the role of a SLC16A6 phospho-degron and the contributions of PDZ-domain proteins LIN7C and MPP1 to the trafficking of SLC43A2. Overall, our work offers a resource for SLC-protein interactions for the scientific community.**

**Keywords** AP-MS; Protein–protein Interactions; Proteostasis; SLC Superfamily; Trafficking
**Subject Category** Proteomics

See also: T Wiedmer et al, G Wolf et al & U Goldmann et al

## Introduction

Biological membranes demark organismic, cellular and organellar boundaries and are essential for life. The transport of all sorts of organic and inorganic molecules across membranes by membrane-spanning proteins is regulated at many levels, from transcriptional to post-translational, directed by availability of solutes as well as biosynthetic, metabolic and informational requirements.

Within human cells, it is estimated that between 1500 and 2000 genes contribute to transport across membranes (Ye et al, 2014). Transporters take a key role to enable homeostasis, growth, and proliferation. Solute carrier (SLC) proteins can transport substrates using electrochemical gradients and are the largest transporter class with more than 450 members (Meixner et al, 2020; Ferrada and Superti-Furga, 2022; He et al, 2009; Haas et al, 2014). The SLC superfamily covers a wide range of transported substrates and is differentially expressed across cell types and tissues, and across healthy and disease states (Hediger, 2004; Pizzagalli et al, 2021; Lin et al, 2015). In fact, 228 of 456 SLCs have a link to human diseases (Goldmann et al, 2025). The targets of some of the most important so-called blockbuster drugs, such as serotonin-uptake inhibitors and gliflozins (glucose re-uptake inhibitors) are solute carriers (Wang et al, 2020; César-Razquin et al, 2018). Despite the estimated potential of 75% of all SLCs to carry small organic molecules, most chemical efforts have focused on just a few SLC families, leaving most families and most SLCs untargeted (Dvorak and Superti-Furga, 2023; Carter et al, 2019; Fauman et al, 2011; Galetin et al, 2024; Gyimesi and Hediger, 2022; Wang et al, 2020). SLCs are increasingly considered attractive novel therapeutic targets for modifying disease by modulating metabolism. Strategies for the development of inhibitors have often started from the natural substrate, like in the case of PF-06649298 and SLC13A5, a sodium-coupled citrate transporter primarily expressed in the liver, brain, and other tissues (Huard et al, 2015). In the search for alternative routes, it has also been shown that it is possible to target SLC

[1]CeMM Research Center for Molecular Medicine of the Austrian Academy of Sciences, 1090 Vienna, Austria. [2]Novartis Pharma AG, Novartis Biomedical Research NBR/DSc, CH-4002 Basel, Switzerland. [3]Division of Drug Discovery and Safety, Leiden Academic Centre for Drug Research, Leiden University, Einsteinweg 55, 2333 CC Leiden, The Netherlands. [4]Department of Biotechnology, University of Natural Resources and Life Sciences, 1190 Vienna, Austria. [5]Center for Physiology and Pharmacology, Medical University of Vienna, 1090 Vienna, Austria. [6]Fondazione Ri.MED, Palermo, Italy. [7]These authors contributed equally: Fabian Frommelt, Rene Ladurner. ✉E-mail: gsuperti@cemm.oeaw.ac.at

transporters using Proteolysis Targeting Chimeras (PROTACs), causing efficient degradation of the target (Bensimon et al, 2020; Zhang et al, 2024). However, the majority of SLC-associated diseases represent loss-of-function situations where mutations impair protein levels, such as SLC6A8 and creatine transport deficiency (Ferrada et al, 2024; Valayannopoulos et al, 2013) or SLC39A8, the metal transporter associated with congenital disorder of glycosylation type II and Leigh syndrome (Park et al, 2015). The success in partially restoring the function of hypomorphic alleles of the cystic fibrosis transmembrane conductance regulator (CTFR) ion channel gene by chemical modulators (correctors and potentiators) highlights the possibility that even partial restoration of protein folding and localization can be of clinical benefit (Liu et al, 2024; Fiedorczuk and Chen, 2022), suggesting an avenue for the hundreds of SLC-associated loss-of-function pathologies.

Due to the substantial lack of cellular, molecular and functional data for a large portion of the SLC superfamily (César-Razquin et al, 2015), the RESOLUTE consortium, a public-private partnership, used a systematic multi-omics approach to gather molecular and functional data on SLCs (Fig. 1A), thus complementing the interaction proteomics approach with parallel investigations in metabolomics, transcriptomics (Wiedmer et al, 2025), genetic interactions (Wolf et al, 2025), and multi-omics data analysis (Goldmann et al, 2025). Characterization of the multiprotein complexes to which SLC transporters participate represents an important functional dimension of SLCs because of their potential to reveal connections to the biological and biochemical machinery of the cell. The characterization of the protein interactome of SLCs may inform novel therapeutic approaches (Yan et al, 2020; Gomez et al, 2023; Boeszoermenyi et al, 2023). Further, an interaction proteomics approach can reveal chaperones involved in protein folding, which on top of representing therapeutic intervention routes can improve heterologous expression of SLCs and subsequent structural characterization (Torres et al, 2003).

The protein interactome plays a crucial role for all cell activities, but the interactions of membrane proteins are difficult to study by standard biochemical protein interaction techniques (Schey et al, 2013; Cao et al, 2021). Human proteome-wide interactome studies using mass spectrometry have therefore only reported on a subset of transmembrane domain-containing proteins (TM-proteins) and of SLCs in particular, often with protocols developed to mainly suit the study of soluble proteins (Huttlin et al, 2021, 2015, 2017). Isolated interactome analyses of individual TM-proteins, on the other hand, can miss the necessary context to understand if interactions are specific or are common cellular interaction modalities for multiple-pass TM-proteins localized on all kinds of subcellular membranes. This is a particular challenge for highly sensitive proteomics, for which several frameworks were developed to assign pairwise protein interaction confidence across large-scale affinity purification coupled to mass spectrometry (AP-MS) datasets (Teo et al, 2014; Sowa et al, 2009). AP-MS, as a state-of-the-art method to assign protein interactions offers high sensitivity and reproducibility (Varjosalo et al, 2013), and thus is especially suited for large-scale studies (Gavin et al, 2002; Huttlin et al, 2021; Uliana et al, 2023; Buljan et al, 2020; Salokas et al, 2022).

Studies that focused on single SLCs established the general applicability of MS-based interaction proteomics and linked two transporters to important cellular metabolic machineries. Among these are the interaction of the amino acid transporter SLC38A9

with the Ragulator/LAMTOR complex (Rebsamen et al, 2015; Wang et al, 2015) and the interaction of SLC15A4 with TASL (Heinz et al, 2020). Interaction proteomics studies have also revealed SLC heterodimers, including SLC16-family members with BSG/EMB (Howard et al, 2010; Rusu et al, 2017; Halestrap, 2013), SLC7-family members with SLC3A1/SLC3A2 (Yan et al, 2020; Parker et al, 2021) and within members of the ZIP-family of Zinc transporters (SLC39-family) (Taylor et al, 2016). Other examples showed that protein interactions determine the accurate folding and ER export of SLCs through chaperone interactions (Wiktor et al, 2021; Ohtsubo et al, 2011) or affect the correct trafficking of SLCs (Yang et al, 2018; El-Kasaby et al, 2010). In addition, several SLCs are regulated through proteostatic pathways (Colaco et al, 2023; Xu et al, 2016) or endocytosis (Puris et al, 2022; Rotin and Staub, 2012).

Here, we used a systematic AP-MS approach to determine the protein interaction network of human solute carriers. For this we isolated and analyzed over 400 SLC proteins using improved purification strategies from human embryonic kidney cells. We then compared all individual SLC interactomes and in combination with machine learning (ML), assembled 18,991 interactions for 396 individual SLCs spanning 68 different SLC families.

We integrated gene ontology and metabolic information to understand the nature of these novel interactions and identified individual interacting proteins and protein complexes linked to organelle function, protein trafficking, protein degradation, post-translational modification and signaling. By genetic loss-of-function analysis, we found dozens of interactors that led to a change in protein stability of the interacting transporter, charting the proteostatic regulatory network of SLC transporters.

# Results

## Obtaining a robust protein interactome of SLC transporters

For the systematic characterization of SLC-protein interactions, we aimed to establish a robust and scalable experimental affinity enrichment protocol, combined with a reproducible MS-based proteomics method suitable to map the entire human interactome of SLCs in a specific cellular setting.

All 405 SLCs profiled within this study were subjected to an experimental workflow specifically adapted for TM-proteins (Fig. 1B), and built on previously described AP-MS protocols (Varjosalo et al, 2013; Gavin et al, 2002; Rebsamen et al, 2015; Uliana et al, 2023). Full elution of TM-proteins was achieved by employing an SDS-based elution. For buffer exchange to remove remaining SDS, we adapted an SP3-based sample clean-up protocol (Müller et al, 2020), which enabled scaling up sample preparation. For SLCs localized to mitochondria, we performed a crude mitochondrial enrichment prior to lysis of the mitochondrial fraction by digitonin followed by affinity purification (see "Methods" for details). The resulting AP-samples were acquired by data-dependent acquisition (DDA) in biological duplicates with technical injections.

To perform AP-MS for the human SLC superfamily, the codon-optimized consensus sequence of each SLC bait protein was either N- or C-terminally fused to a double-strep-tag and HA-epitope

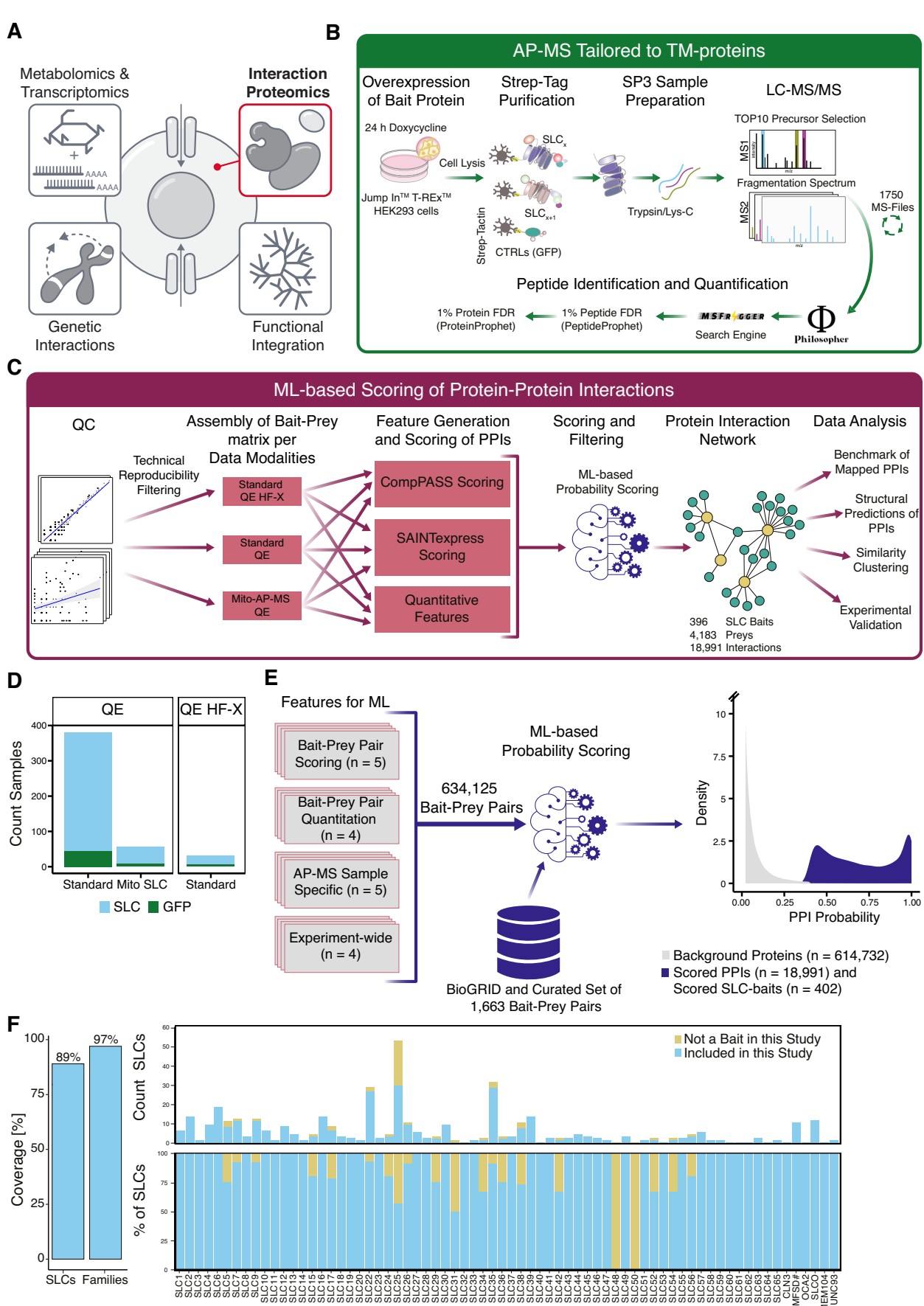

**Figure 1. A systematic AP-MS approach to define the human solute carrier superfamily interactome.**

(**A**) Overview of the -omics layers to characterize SLCs on a molecular level generated within the RESOLUTE project. (**B**) Experimental workflow for native purification of SLC-containing protein complexes from HEK 293 Jump In T-Rex cell lines. (**C**) Modular data analysis pipeline for processing of MS data and scoring of PPIs. (**D**) SLC baits grouped by protocol and MS-platform. (**E**) A set of 18 features (Table EV2) and a curated list of labeled PPIs served as input for the scoring. Distribution of scored interactions and scored SLC baits as preys (dark blue) versus background proteins (gray). For visualization of both distributions, the y axis was cut at a density of 10. (**F**) AP-MS coverage across the SLC superfamily. The left bar chart shows the coverage of SLC baits (396 of 447 SLCs investigated in RESOLUTE) and families (68 of 70 SLC families) included in the SLC interactome. The upper part of the split graph reports the counts of SLC baits used within the study separated per SLC-family. The count of SLCs (y axis) is plotted against the SLC families (x axis). SLCs used as bait in the SLC interactome are marked in blue, SLCs not included in this study or filtered after scoring are colored in yellow. The lower part of the bar chart represents the percentage coverage per SLC-family, indicating in blue the percentage of family members reported in the SLC interactome and in yellow the percent of family members which are not included. The y axis shows percent coverage by family and the x axis indicates SLC families.

(Table EV1) and stably expressed under a doxycycline-inducible promoter in HEK 293 Jump In T-REx cells. HEK 293 were also used as model cell line for many different large-scale interaction proteomics studies focusing on individual protein families such as RTKs (Salokas et al, 2022) or kinases (Varjosalo et al, 2013; Buljan et al, 2020), and for the human proteome-wide series BioPlex studies (Huttlin et al, 2015, 2017, 2021), thus allowing the comparison of the SLC-interactome data to these datasets. A project of this magnitude required a computational pipeline which was able to handle large amounts of MS-injections and integrate a fine-tuned scoring framework to account for differences in expression and subcellular localization of the various SLCs. We set up a modular processing pipeline employing Philosopher (da Veiga Leprevost et al, 2020), which we integrated with an in-house developed scoring approach (Fig. 1C). Our scoring approach considered also the three data modalities, which were obtained by using two different affinity enrichment protocols and two MS-platforms (Fig. 1D). For the ML-scoring, a combination of 18 features was derived for all 634,125 interaction pairs (Fig. 1E; Table EV2), serving as input for an RBF classifier (see "Methods" for details).

In total, we acquired data for 405 SLCs, with 396 SLCs remaining in the interaction proteomics datasets after scoring and filtering of background proteins. The SLC interactome covered 97% of the SLC families and 89% of the 447 SLCs which were investigated by the RESOLUTE consortium (Fig. 1F; Dataset EV1) (Superti-Furga et al, 2020).

## General characteristics of the SLC interactome

To effectively identify co-purified proteins within an AP-MS experiment, the bait protein must be sufficiently expressed. Low bait expression levels result in reduced enrichment of interacting proteins, making it challenging to distinguish bona fide protein interactors from background preys (e.g., proteins with an affinity to the bead material). The bait abundance evaluation is of particular importance for TM-proteins, as overexpression is difficult to achieve due to the need of folding chaperones and extensive PTM and processing (Czuba et al, 2018), including glycosylation (Console et al, 2015) and palmitoylation (Villanueva and Hagenbuch, 2023). We first assessed bait abundance and its influence on total signal quantified within a sample and on the number of interactors. Within the SLC interactome, we obtained a median of 249 spectral counts per SLC bait (Appendix Fig. S1A), with only very few SLCs showing a higher signal. Within the group of highly abundant baits were mainly large SLC transporters with

many tryptic peptides, such as the cation-chloride cotransporters or SLC12-family members, indicating the importance of normalizing for protein length. The abundance of SLCs did not correlate with the total signal per experiment (Pearson correlation coefficient, $R = 0.22$, $P < 0.001$; Appendix Fig. S1B), indicating no loss of signal in less expressed SLCs or suppression of prey quantitation with abundant baits. Similarly, the number of interactors and bait abundance did not correlate (Pearson correlation coefficient, $R = 0.07$, $P = 0.162$; Appendix Fig. S1C).

Further, we compared the log2FC of each SLC bait against GFP controls as well as the log2FC against all other AP-MS samples (Appendix Fig. S1D). The two log2FC significantly correlated with each other, indicating a strong enrichment of the SLC bait in the AP-MS experiment (Pearson correlation coefficient $R = 0.93$, $P$ value <0.01). Several SLCs showed lower log2FC enrichment against the membrane background of other SLC AP-MS samples compared to GFP controls. Each SLC bait demonstrated a strong enrichment compared to GFP negative controls (minimum log2FC of 3.24 across the SLC interactome) and the background of the same SLC identified in other AP-MS samples (minimum log2FC of 1.49 across the SLC interactome). We concluded that AP-MS data from other SLCs provided a more accurate representation of the inherent background co-purified during affinity enrichment. Therefore, we used the entire dataset in the filtering and scoring strategy to correct for co-purified SLCs.

The interactors had a signal that was at least twofold higher than before filtering (Appendix Fig. S2A). Across our dataset, we quantified on average 1566 proteins per SLC, indicative of a highly complex membrane background (Appendix Fig. S2B), albeit with strong variation (a few hundred to 3000 proteins). This may be due to SLCs being expressed in different compartments and the fact that the data were obtained using different MS-platforms. We assigned on average 48 interaction partners per SLC, which roughly corresponds to the number of interaction partners when investigating an individual SLC (Rebsamen et al, 2015) and to other large-scale human interaction proteomics studies (Uliana et al, 2023; Sowa et al, 2009; Ciuffa et al, 2022). A few SLCs retrieved a higher number of interaction partners, maybe reflecting their multiple subcellular localizations (Appendix Fig. S2C). To test for a bias toward frequently or uniquely quantified proteins, we plotted the frequency of observation against the quantitative signal of each interactor (Appendix Fig. S2D). We scored unique as well as often found interaction partners, suggesting no penalty for frequently quantified proteins.

To determine the network properties of the fully assembled SLC interactome, we transformed the obtained network into an

undirected network graph and calculated the degree of connectivity: the number of adjacent edges per bait or interaction partner. In addition, we calculated the Kleinberg's hub centrality score, which is a measure of node influence on connecting other nodes (Kleinberg, 1999).

Within the network, around 25% of the nodes were uniquely connected, whereas most nodes were densely connected with a median of 33 edges (Appendix Fig. S2E). The top 20 most connected interaction partners of the SLC interactome included several interactions that were previously linked to TM-proteins and/or SLC function (Appendix Fig. S2F). Among these were the folding chaperone calmegin (CLGN) (Huttlin et al, 2021), ANKRD13A, an ankyrin repeat domain-containing protein involved in the internalization of receptors (Mattioni et al, 2020), and the ER-localized chaperones ERLIN1/2 (Wiktor et al, 2021). Investigating the most influential hubs across the binary network revealed 54 interaction partners and 46 SLCs among the 100 most important hubs. Among the most connected interactors were proteins involved in the glycosylation machinery (e.g., DDOST, RPN1/2, STT3A, GALNT2/7) and protein trafficking (e.g., GOLGA5, SEC62, CNIH4, Appendix. Fig. S2G).

## Benchmarking of the SLC interactome

To determine the relevance of the data, we benchmarked the interaction data on a PPI level against reported PPIs in databases as well as on a multi-subunit protein complex level. Compared to other TM-proteins or proteins without an annotated TM-domain, few SLCs are characterized by AP-MS and fewer PPIs are reported, making benchmarking challenging (Fig. 2A).

We generated a combined reference PPI library retrieving interactions from multiple sources, including BioGRID (Oughtred et al, 2021), IID (Kotlyar et al, 2022), IntAct (Orchard et al, 2014), and STRING (Szklarczyk et al, 2023). We subsequently filtered the PPI library to contain interactions obtained by AP-MS and equivalent interaction mapping techniques (Table EV3 see "Methods"). The resulting reference PPI library covered 404 (90%) of the 447 SLCs and a total of 16,072 PPIs (Fig. EV1A). The reference PPI library showed considerable variation with 197 SLCs (44%) and only 1214 PPIs (7.55%) overlapping across databases, and nearly half of the PPIs (7899) were only reported by a single database, thus highlighting first the need to combine PPI-libraries and second, to add high-quality SLC interactome data as a new reference point (Fig. EV1B).

We compared the SLC-interactome and reference PPI library using the similarity matrix between the structural models of human SLCs (Ferrada and Superti-Furga, 2022), and represented the relationships by an unrooted structural similarity tree, similar to how the human kinases were shown in the kinome tree (Manning et al, 2002; Karaman et al, 2008). Unlike the kinome tree, there is no common ancestor, and phylogenetic relationships are valid only within one fold (details in (Goldmann et al, 2025)). With each node representing an individual SLC, the edges were colored by the major structural folds of the SLCs, and all SLCs included in the study were colored in blue (Fig. EV2; Dataset EV2). In total, 24 clade classifications with an additional clade classified as unknown were annotated. Clustering and fold classification were as described in Ferrada and Superti-Furga (Ferrada and Superti-Furga, 2022). The SLCome was decorated with the PPIs per clade, scored within

the SLC interactome, and reported in the reference PPI library (see "Methods"). For 23 clades, we found 18,232 novel PPIs, increasing the number of PPIs up to sixfold per clade and indicating that the PPI coverage is mostly independent of structural and evolutionary features.

We found less interactions compared to the reference PPI library for six structural clades, among them the MitC clade (3913 in literature vs. 454 within the SLC interactome) and the SLC56 clade (674 vs. 14). These two clades consist of mitochondrial transporters. This might be because highly abundant mitochondrial carriers are often wrongly assigned as interactors, as suggested by their high CRAPome presence (Mellacheruvu et al, 2013). Eleven SLCs from the reference PPI library had a frequency above 20%, indicating non-specific engagement to the beads-matrix, and of those, eight were mitochondrial SLCs. To investigate if SLCs found with a high CRAPome presence are overrepresented as interaction partners, we calculated the ratio of these SLCs in the bait or prey role for each PPI reported in BioGRID (Fig. EV1C). We found that SLCs with higher presence in the CRAPome database were significantly more often reported as interactors (90.24% vs. 60.45%), indicating that many of the interactions were likely non-specific (Fig. EV1D).

We next investigated the potential bias of currently reported PPIs towards heavily studied superfamily members, as shown for example for kinases (Buljan et al, 2020). As expected, the number of reported interactions and associated studies strongly varied between SLCs (Fig. 2B). We grouped the SLCs according to the number of references: more than 10 references (72 SLCs), less than 10 references (315 SLCs) or no associated references (60 SLCs) (Fig. 2B). Comparing the number of PPIs between highly studied or poorly characterized SLCs revealed a significant difference for the reference PPI library (Welch two-sample $t$ test $P$ value $9.791e^{-12}$), but not the SLC interactome (Welch two-sample $t$ test $P$ value 0.0896), thus supporting that our systematic approach provides novelty across the SLC superfamily independent of prior knowledge (Fig. 2C).

Comparing the SLC interactome to other large-scale studies, we found that 148 baits had not been covered by other large-scale AP-MS (Huttlin et al, 2021) or yeast two-hybrid (Luck et al, 2020) studies (Fig. 2D). This study almost doubles the number of SLC interactomes of previous large-scale studies and contains approximately ten times more PPIs for the SLC superfamily (Fig. 2E). When comparing all three datasets against the reference PPI library, the SLC interactome had, with 3,9%, the highest recovery rate (Fig. 2F) and identified the largest number of PPIs with two or more associated studies (Fig. 2G). Within our dataset we found 740 PPIs that were previously identified, compared to 11 and 8 for HuRI and BioPlex, respectively. A total of 96.1% interactions within the SLC interactome are novel, which is slightly higher than comparable interaction proteomics studies that identified 80–85% novel interactions (Taipale et al, 2014; Huttlin et al, 2015).

To further benchmark our PPI-network, we investigated whether it was enriched for reported protein complexes. With only 19 SLCs covered, SLCs and their complex associations are underrepresented within CORUM (Tsitsiridis et al, 2023), a reference database of protein complexes, in comparison to other TM or soluble proteins (Fig. EV1E). For this, we assumed full connectivity between all interactors of each SLC and of all complex subunits reported in CORUM. We intersected the CORUM-derived

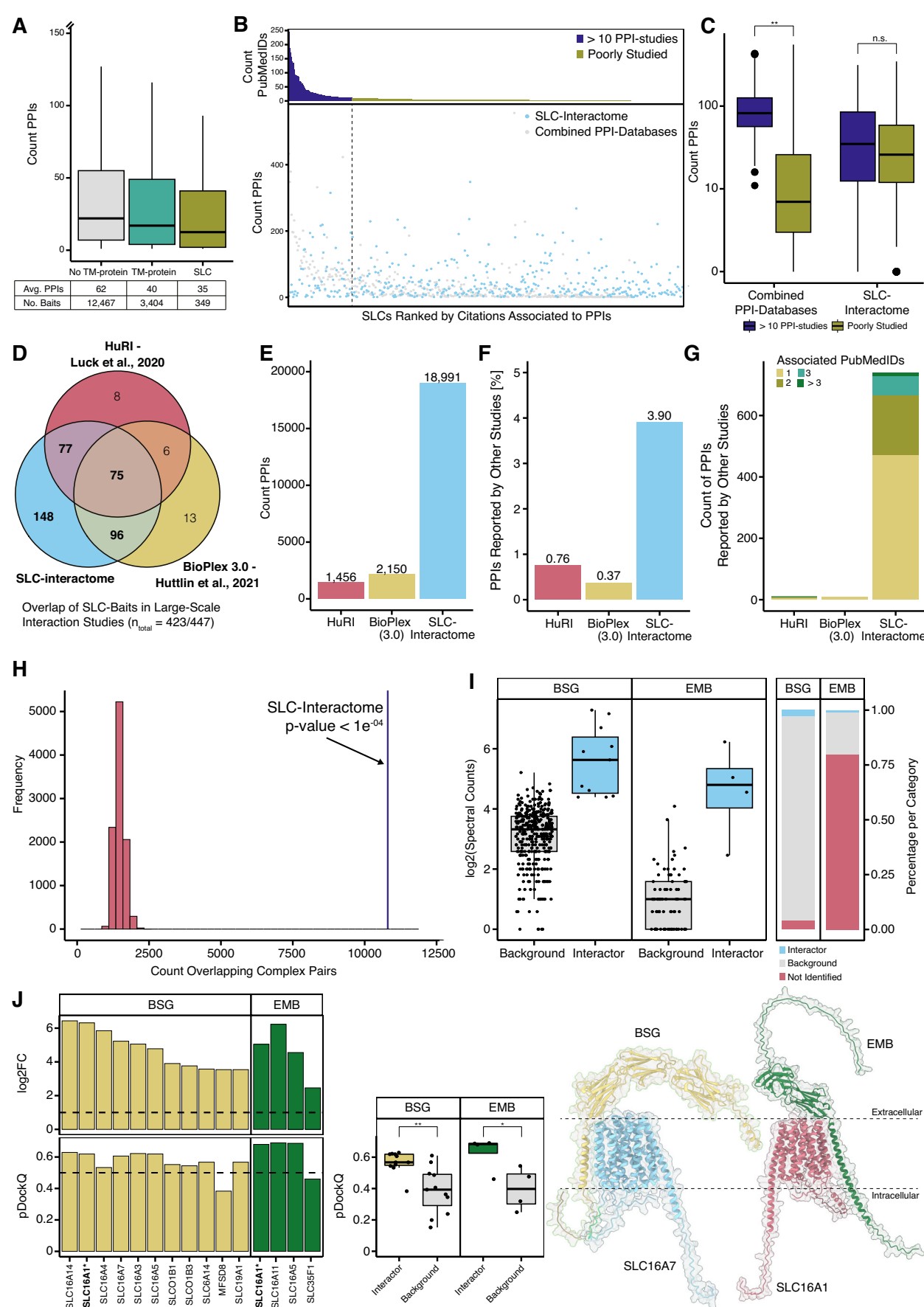

**Figure 2. Assessment of SLC-protein interaction network and data quality.**

(A) Reported protein interactions for SLCs ($n = 349$, olive) versus TM-domain-containing proteins ($n = 3404$, teal) and proteins without annotated TM-domain ($n = 12,467$, gray). Lower and upper hinges of box plots correspond to the 25th and 75th percentiles, respectively. Lower and upper whiskers extend from the hinge to the smallest or largest value no further than the 1.5× interquartile range from the hinge, respectively. The black line represents the median. Outliers were removed and to increase readability the y axis was cut at 150 PPIs. (B) PPIs per SLC reported in the PPI library. The upper part shows the associated studies per PPI, and the lower part the number of reported PPIs (gray dots) and PPIs identified within the SLC interactome (blue dots). SLCs were grouped by the associated publication into a group of SLC which were studied by interaction proteomics (dark blue, >10 referenced studies) and a group of poorly characterized SLCs (yellow, <10 associated studies or none). (C) Comparison of PPIs reported for the poorly studied (yellow) against more often studied SLCs (dark blue). For the SLC interactome no bias was observed (in literature poorly characterized, $n = 332$, average PPIs = 45.4, >10 associated studies, $n = 64$, average PPIs = 61.1, Wilcox rank-sum test P value = 0.3318, indicated as "n.s." in the figure panel) in comparison to the literature reported database for which a statistically significant bias between SLCs was found (poorly characterized, $n = 375$, average PPIs = 21.8, >10 associated studies, $n = 72$, average PPIs = 111, Wilcox rank-sum test P value < 2.2e-16, indicated with "**" in the figure panel). Lower and upper hinges of box plots correspond to the 25th and 75th percentiles, respectively. Lower and upper whiskers extend from the hinge to the smallest or largest value no further than the 1.5× interquartile range from the hinge, respectively. The black line represents the median and the black dots represent outliers. (D) Overlap of SLCs used as baits in the SLC interactome study (blue), in the BioPlex (yellow) and HuRI (red). (E) PPIs reported for SLCs in the BioPlex (yellow), HuRI and the SLC interactome (blue). (F) Fraction of protein interactions reported by BioPlex (yellow), HuRI (red), and the SLC interactome (blue) that were reported by additional studies. (G) PPIs reported in literature and the two large-scale reference studies, and the SLC interactome. The color indicates associated studies for reported PPIs. (H) CORUM-derived interaction pairs are enriched within the SLC interactome in comparison to 10,000 permuted networks with conserved topology and composition. The blue line indicates the overlapping PPIs with reported PPIs of deconvoluted CORUM complex found in the SLC interactome (significantly more PPIs found; P value < 1e$^{-04}$ compared to the permuted PPI-networks; as none of the permuted networks recovered more CORUM PPI pairs than the SLC interactome, we estimated the P value by 1/10,000 to be below 1e$^{-04}$). (I) Distribution of BSG and EMB, two chaperones of SLC16-family members, across the SLC interactome. The left panel shows the log2 transformed SPCs separated by scored interactions (BSG $n = 11$ and EMB $n = 78$, blue) and background (BSG $n = 378$ and EMB $n = 4$, gray) within the SLC interactome. Lower and upper hinges of box plots correspond to the 25th and 75th percentiles, respectively. Lower and upper whiskers extend from the hinge to the smallest or largest value no further than the 1.5× interquartile range from the hinge, respectively. The black line represents the median and the black dots represent single measurements. Bars on the right side indicate how often the chaperones were scored or found as background across the SLC interactome. (J) Upper part shows for SLC-BSG (yellow) and EMB (green) interactions the log2FC against GFP (dotted black line log2FC > 1), and the lower part shows the pDockQ (dotted black line pDockQ >0.5, high-confidence structures). Complexes for which the structure was experimentally solved are marked with an asterisk and are in bold (*). Predicted SLC-chaperone structures ($n = 11$ BSG and $n = 4$ for EMB complexes) were compared against a set of predicted structures of SLC-chaperones for which the chaperones were classified as background ($n = 11$ BSG and $n = 4$ for EMB complexes; unpaired Student t test for BSG P value = 0.001589 and EMB with a P value = 0.02762; independent control sets and tests). In the figure, P values below 0.01 are marked with "**", and P values between 0.01 and 0.05 are marked with "*". Lower and upper hinges of box plots correspond to the 25th and 75th percentiles, respectively. Lower and upper whiskers extend from the hinge to the smallest or largest value no further than the 1.5× interquartile range from the hinge, respectively. Black line represents the median and the black dots represent scores per complex. On the right side predicted structures for SLC16A7-BSG and SLC16A1-EMB complexes are shown.

PPI-network against 10,000 permuted networks with the same topology and composition as the SLC interactome (see "Methods"). CORUM interaction pairs were significantly enriched within the SLC interactome compared to permuted networks (Fig. 2H).

## Orthogonal evidence of SLC-protein interactions by in silico structural modeling

Recent progress in deep-learning methods to predict structures of experimentally determined protein interactions can lead to high-confidence structural models of protein complexes (Burke et al, 2023). However, the success rate for the long C- and N-terminal tails of TM-proteins is limited, as these tails often contain intrinsically disordered regions (IDR) with protein interactions mostly mediated by small linear motifs (SLiMs) (Morris et al, 2021). Despite these limitations, we used AlphaFold-Multimer for structural prediction of PPIs (Evans et al, 2021). For the in silico validation, we decided to start from well-studied SLC-protein interactions and expand to novel interactions of SLCs. First, we focused on heterodimeric complexes of monocarboxylate transporters (MCTs). SLC16-family members were reported to form stable associations with small single-pass transmembrane chaperone proteins Basigin (BSG) and Embigin (EMB) (Bosshart et al, 2021; Wilson et al, 2005; Poole and Halestrap, 1997). Several SLC16-family members, such as SLC16A1, SLC16A3, and SLC16A7, are widely expressed, while others exhibit tissue-specific expression patterns. For instance, SLC16A8 is specifically expressed in the retinal pigment epithelium (Philp et al, 2003). The chaperone BSG is ubiquitously expressed, whereas EMB expression

is more restricted (Guenette et al, 1997). It was further reported that the interaction of SLC16A11 with BSG was dysregulated in Type 2 diabetes variants (Rusu et al, 2017). In our dataset, BSG was detected in 93.33% of all AP-MS samples and scored for eleven SLCs. We found interactions with several SLC16-family members and additionally with SLCO1B1, SLCO1B3, MFSD8, and SLC19A1. Compared to BSG, EMB was only detected in ~20% of all our AP-MS experiments (Fig. 2I). We modeled the structure of the SLC-chaperone interactions found in our SLC interactome (Fig. 2J; Appendix Fig. S3A–D; Dataset EV3). For 13 out of the 15 heterodimers, we retrieved high-confidence structures (pDockQ >0.5), among them SLC16A1 in complex with both chaperones as previously reported (Halestrap, 2013), as well as SLC16A5 and SLC16A11 in complex with BSG, for which no structure was reported.

To provide further benchmarking examples, we obtained structural models from SLC3A2 interactions identified within our dataset. The amino acid transporter heavy chains SLC3A1 and SLC3A2 are well-known for their interactions with members of the SLC7 family. SLC3A2 forms heterodimeric complexes with multiple members of the L-type amino acid transporters (LATs) (Rodriguez et al, 2021; Lee et al, 2022; Oda et al, 2020; Mastroberardino et al, 1998; Estévez et al, 1998; Kandasamy et al, 2018). These complexes are also referred to as heterodimeric amino acid transporters (HAT). SLC3A2 was detected in 87% of all SLC AP-MS experiments and scored as interaction partner for a total of seven SLCs, of which six are members of the SLC7 family (Appendix Fig. S4A). We further modeled the complexes and obtained high-confidence structures for all SLC3A2 interactions

(Appendix Fig. S4B). Comparing predicted chaperone complexes against a set of non-reported chaperone-SLC interactions, randomly sampled from the lowest 25% quantile of SLC3A2 abundance quantified within the SLC interactome, showed that the scored interactions led to significantly higher scoring structures (unpaired Student $t$ test, $P$ value = 0.0009177, Appendix Fig. S4B, right side, S4C,D).

Further, we modeled the interactions of SLCs with the Calcineurin B homologous protein 1 (CHP1). CHP1 was reported to form heterodimeric complexes with SLC9A1 and SLC9A3, stabilizing their plasma membrane localization and increasing pH sensitivity (Dong et al, 2021). We found CHP1 to interact with even more SLC9-family members as well as SLC6- and SLC7-family members (Appendix Fig. S5A). CHP1 was detected at significant levels for SLC9A2–SLC9A4 and with lower abundance for SLC9A5–SLC9A9. The affinity purifications of SLC9A1 did not yield sufficient signal and were thus excluded from the SLC-interactome study (see "Methods" and Table EV1). Structural prediction of these PPIs resulted in high-confidence structures for all CHP1-SLC9-family heterodimers and for four other SLCs with a medium-confidence structure (Appendix Fig. S5B). Also, the obtained structural models for the identified SLC-CHP1 interactions were of significantly higher quality compared to a random set of SLC-CHP1 interactions classified as background (unpaired Student $t$ test, $P$ value = 0.03642, Appendix Fig. S5C–E; Dataset EV3).

We also evaluated novel SLC interactions with GALNT2 (polypeptide N-acetylgalactosaminyltransferase 2), which was so far only reported to interact with three SLCs (NPC1, NIPAL1, SLC25A4; see reference PPI library). GALNT2, expressed ubiquitously and localized to the cis- and trans-Golgi apparatus, where it catalyses the initial steps of the O-linked glycosylation of proteins (Kurze et al, 2019). O-linked glycosylation was found to affect protein stability of receptors, such as EGFR (Wu et al, 2022). GALNT2 was scored as interactor for 21 SLCs (Fig. EV1F), of which eleven were metal transporters of the SLC30 and SLC39 families. (Fig. EV1G, upper part). We obtained 14 high-confidence and seven medium-confidence structures of the SLC-GALNT2 complexes (Fig. EV1G, lower part). We compared the structures of these heterodimers against a set of non-reported SLC-GALNT2 complexes, randomly sampled from the lowest 25% quantile of GALNT2 abundances quantified within our dataset. The structures of scored interactions showed a significantly higher pDockQ score (unpaired Student $t$ test, $P$ value = 0.00695, Fig. EV1G, right side). Lastly, we investigated one of the high-scoring models, the complex of SLC30A1 with GALNT2 (Fig. EV1H; Appendix Fig. S6A,B). SLC30A1 is a $Zn^{2+}/Ca^{2+}$ exchanger localized at the plasma membrane, and forms a homodimer (Sun et al, 2024). The predicted interaction interface of GALNT2 is largely distinct from the homodimer interaction interface. SLC30A1 was reported to be N-glycosylated at position 299 (Asn-299) in the extracellular loop between transmembrane domain, affecting its stability (Nishito and Kambe, 2019). Profiling of O-linked glycosylated proteins identified two O-linked sites on SLC30A1, one being either S300 or T301 (Steentoft et al, 2013), again at the extracellular loop, which is predicted to interact with GALNT2.

In summary, the benchmarking effort highlighted that the SLC interactome recapitulated well-characterized interactions and recovered thousands of novel interactions. The recovery rate of already reported interactions was higher compared to other large-scale studies. Prediction of heterodimers enabled the in silico validation of several known as well as unreported SLC-containing complexes. Finally, this SLC interactome combined with machine learning algorithms is suitable for predicting the structure of interacting surfaces and for the generation of thousands of three-dimensional models, which are useful for interpreting SLC genetic variants associated with disease.

## Clustering of SLC-interaction profiles reveals functionally relevant interaction networks

Faced with the challenge of visualizing the interactome in a manner that is reader-friendly and biologically meaningful, we abstained from a hairball network representation. Clustering of interactor relations is regularly used to deconvolute high-density PPI-networks, obtained by AP-MS (Buljan et al, 2020; Uliana et al, 2023) or in vivo proximity interaction mapping (Go et al, 2021; Salokas et al, 2022), into protein complexes. In addition to this approach, we grouped the SLCs based on their PPI-network similarity to derive common properties such as subcellular location, protein complexes, and family-wide interactions.

For the latter approach, we performed hierarchical clustering based on the overlap of interaction partners, resulting in a dendrogram of 396 SLCs in 38 clusters (Fig. 3A). The number of clusters was chosen using a local peak in the mean silhouette width by estimating a cluster number from $k = 2$ to $k = 110$ (Appendix Fig. S7). The clusters included on average 10 SLCs (min: 2, max: 78) and 319 interactions (min: 2, max: 1185).

For a coarse overview of biological processes associated with the interactome profiles of each cluster, we performed GO biological processing terms enrichment analysis. To reduce complexity and graphically represent functional similarities between different SLCs, we filtered for frequent terms and then performed hierarchical clustering in combination with semantic similarity and reduction analysis to determine important parental terms. We obtained seven representative parental terms to annotate the PPI-network dendrogram (Fig. 3A, outer ring; Appendix Fig. S8A–C; Dataset EV4 and "Methods").

Most of the parental terms were enriched for clusters 17, 18, 31, and 35, indicating that they contain SLCs with large, complex interactomes. Other clusters were only enriched for specific terms; for example, clusters 5 and 34 showed a strong enrichment for COPI-coated Vesicle Budding. The parental terms proteostasis and protein subcellular localization and trafficking were more often found to be significant. Some clusters lack enrichment for any parental term, as we removed child terms to simplify visualization. A heatmap of all enriched terms is shown in Appendix Fig. S8A.

To study the relationship between the clusters, we used all enriched terms (5675 terms, $P$ value < 0.01) and derived a similarity network based on GO semantic similarities (Fig. 3B). The network retained 32 of the 38 clusters, and clusters 31, 17, and 35 were the most connected. Comparing clusters 31 and 35 on SLC level showed that both contained SLC35 family members. After overlapping the PPI-networks of the two clusters, we found that roughly a third (411) of all interaction partners are present in both clusters (Fig. 3C). Among the interactors found in at least 50% of all SLCs in these two clusters were RABL3 and ERMP1, associated with trafficking/signaling and an ER residing endopeptidase.

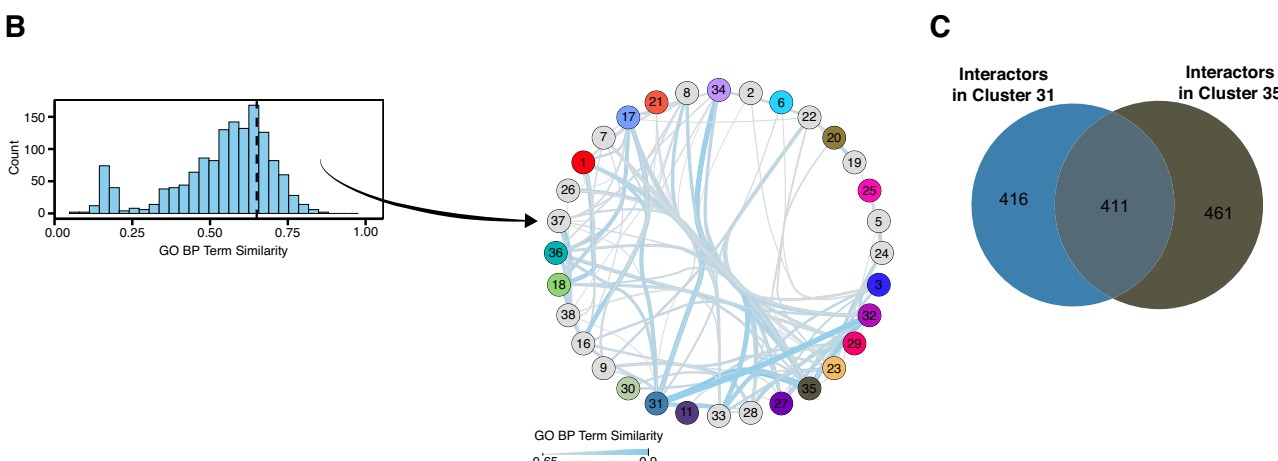

A Protein Localization (GO:0008104)
B Protein Targeting To ER (GO:0045047)
C Intracellular pH Reduction (GO:0051452)
D Proteolysis Involved In Protein Catabolic Process (GO:0051603)
E COPI-coated Vesicle Budding (GO:0035964)
F Cell-Cell Adhesion Via Plasma-Membrane Adhesion Molecules (GO:0098742)
G Organic Substance Transport (GO:0071702)

**Figure 3.   Clustering SLCs by their interactome similarity.**

(A) Dendrogram of hierarchical clustering based on the Jaccard similarity matrix derived from the interactome profiles of 396 SLCs. The heatmap displays the log ratio of the top-level biological process-enriched pathways. Clusters with a significantly enriched SLC functional property (Fisher's test $P < 0.2$) are shown in bold and the respective cluster color. The outer ring shows representative parental GO terms significantly enriched within the cluster ($P$ value $< 0.01$). (B) Distribution of GO semantic similarities between SLC similarity clusters. A similarity threshold of 0.65 (dashed black line) between the clusters was chosen to filter the data. (C) Overlap of protein interactors identified among the SLCs from cluster 31 and cluster 35.

To further compare the SLCs, we tested by enrichment analysis whether SLCs were grouped according to functional properties such as fold, localization, substrate class or family membership ("Methods" for details; SLC annotation from (Goldmann et al, 2025)). For 20 of the 38 clusters (53%) we retrieved at least one significantly enriched functional property (Fisher's test $P < 0.2$) (Fig. EV3A; Dataset EV5) and in total, we found 44 significantly enriched SLC properties that were grouped into five distinct categories: coupled ion, family, fold, location, and substrate (Fig. EV3B). SLC families were overrepresented in several clusters (e.g., SLC4, SLC6, SLC27, and SLC39), indicating that these SLCs share more interactions within their family compared to other SLCs in the SLC interactome (Fig. EV3A). For cluster 30, multiple properties were enriched: coupled ion, substrate, family, and type of fold (Fig. EV3C). The cluster contained eight members from SLC12 and SLC6 families, annotated as chloride-dependent transporters (Meixner et al, 2020), and their combined PPI-network covered a subset of 231 interactions (Fig. EV3D). Enrichment analysis of proteins within the cluster identified cell volume regulation, chloride and ion homeostasis, protein folding and chaperone-mediated protein complex folding as significant terms (Fig. EV3E), thus linking the interaction partners recovered in cluster 30 to the transported substrates of its members (Song et al, 2002; Syringas et al, 2000). 21 out of the 25 most connected interactors in the cluster-specific PPI-network (Fig. EV3F) are reported as physically interconnected (STRING score $>0.4$, filtered for physical interactions; Fig. EV3G). Several of them were folding chaperones, including two HSP90 subunits and four HSP70 isoforms. It was previously found that the immature, ER-resident form of SLC12A1 interacted with HSP70 (Bakhos-Douaihy et al, 2021) as well as other family members (SLC6A14) relying on chaperoning for trafficking (Rogala-Koziarska et al, 2019; Donnelly et al, 2013). These shared chaperone interactions likely represent one layer of ER quality control during the folding process of SLC12 and SLC6 members, possibly because SLC12-family members are large proteins that require specialized stabilization and folding of the LeuT fold.

As HSP70 and HSP90 multichaperones were quantified in many of the SLCs, and particularly high in SLC12-family members, we next investigated if there is a general dependence of finding this interaction and the SLC-protein length. Comparing the protein length of all SLCs interacting with HSP90AA1 with SLCs for which HSP90AA1 was only recovered in the background showed that the median protein length of SLCs interacting with the chaperone was significantly longer (1084 versus 533 amino acids; Student $t$ test $P$ value $= 1.285 \times 10^{-8}$, Fig. EV3H). Next, we tested if the abundance of chaperones correlated with the SLC tail length ("Methods" for details). This revealed that 77% of the chaperones showed a weak to moderate relationship (Fig. EV3I; Dataset EV5).

To reveal complexes covered by multiple SLCs, we clustered interaction partners across the SLC interactome. We pre-filtered all interaction partners by their connectivity, resulting in a subnetwork of 2835 interactors from 371 SLCs, and derived a correlation-based distance matrix. We performed unsupervised hierarchical clustering (Ward D2, silhouette plot Appendix Fig. S9A) resulting in 207 clusters (Appendix Fig. S9B). Next, we performed GSEA for GO biological process terms on the cluster members and found significant terms for 124 (60%) clusters (adjusted $P$ value $\leq 0.01$) (Appendix Fig. S9C). The top five terms were in-line with the coarse overview of biological processes associated with SLC-clusters used in Fig. 3 (Appendix Fig. S9D). To test if interactors of the same cluster form a complex, we intersected CORUM complexes with the interactor clusters, and filtered for completeness of complexes using a 40% complex completeness threshold (Appendix Fig. S9E). For 73% (151) of the clusters, we found a total of 792 CORUM complexes (680 unique CORUM complexes), with the Rag-Ragulator complex being the most complete (Rebsamen et al, 2015). In addition, we found several complexes, associated with trafficking, including multiple small LIN-complexes (Appendix Fig. S9F; Dataset EV6). The LIN-complex will be discussed in more detail in the trafficking result section.

In summary, similarity clustering combined with functional annotation of SLC-cluster-specific networks allowed us to relate SLCs to each other on a functional level. Dissecting one of them led to the identification of a set of chaperones for SLC12 and SLC6 family members, likely involved in their protein quality control. The correlation analysis of interactors identified several protein complexes, despite the sparseness of the SLC interactome, among them the LIN-complex.

## Proteostatic regulation of SLCs

Any functional validation strategy of the SLC interactome, on such a broad spectrum of transporters, localized on different subcellular membranes and likely to transport different substrates, risks of being exemplary and anecdotal in nature. From the pharmacological perspective inspired by the mission of RESOLUTE, modulation of the proteostatic regulation of transporters may restore function to poorly folded or mislocalised gene variants associated with human disease. We therefore considered a strategy allowing to assess new interactors for their ability of affecting the localization and protein levels of transporters. Indeed, GO term enrichment analysis for biological processes of all detected interactions in the SLC interactome identified these biological themes recurring among the top terms (Fig. 3A; Appendix Fig. S8A,B).

We devised a validation workflow that assessed protein localization and degradation. We selected 65 SLCs with interactions involved in these processes and expressed them with an N- or

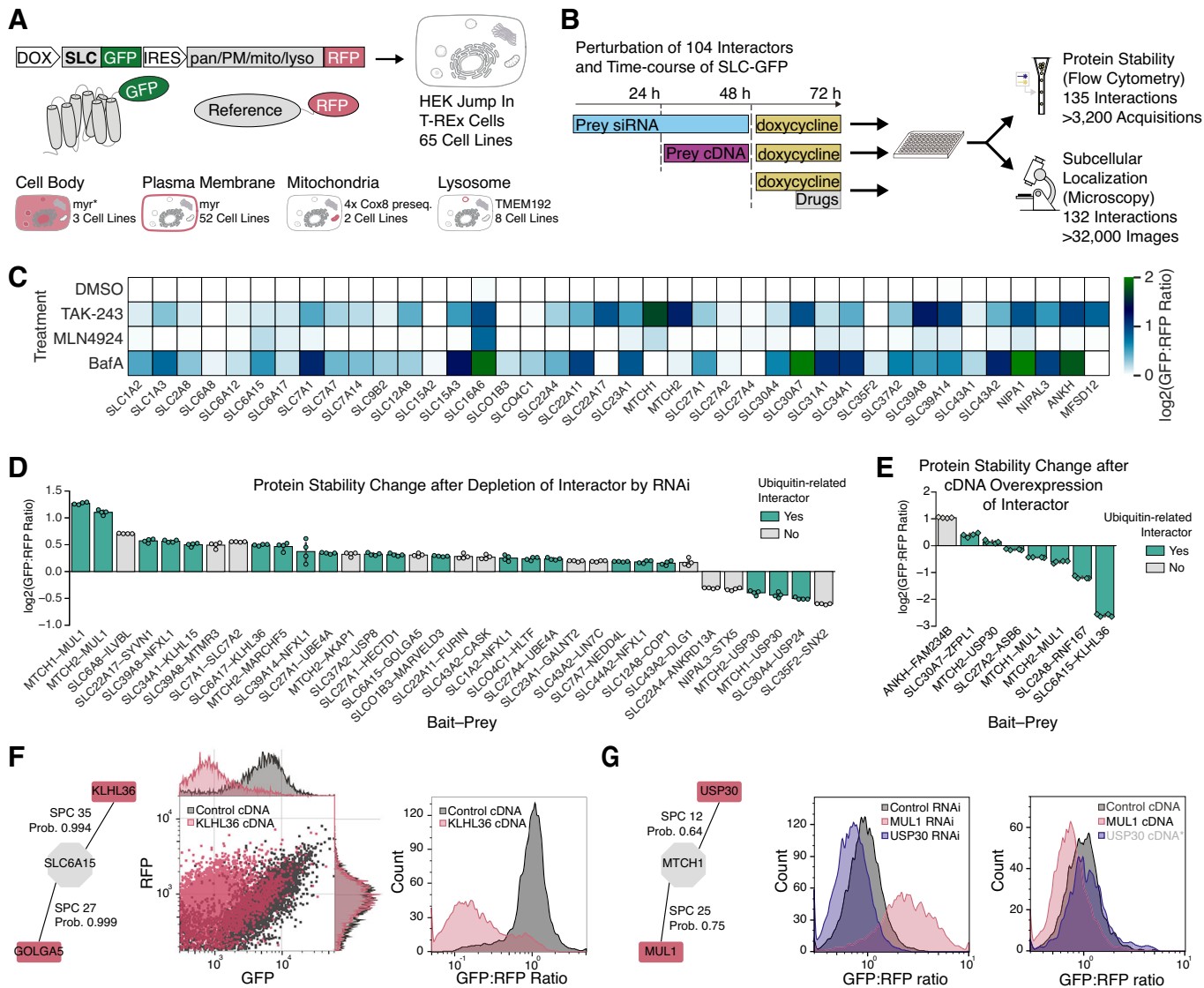

**Figure 4. Proteostatic regulation of SLCs.**

(A) Generation of cell lines conditionally expressing GFP-tagged SLCs together with an RFP reporter modified with membrane-specific tags or proteins that allow localization to biologically relevant membranes. (B) Protein stability and subcellular localization assay to validate SLC interactions. Selected SLCs were conditionally expressed and monitored using flow cytometry and microscopy. GFP:RFP ratio and GFP localization and intensity were measured after perturbation of interactor abundance or drug treatments. (C) SLC-protein stability after drug treatment as measured by flow cytometry showed distinct susceptibility to degradation pathways. Heatmap of log2 transformed GFP:RFP ratios measured after 24 h of induction and drug treatment in the last 6 h (normalized to DMSO control; $n = 3$ wells per sample). (D) Median GFP:RFP ratios for 37 out of 135 tested SLC-protein interactions that were found to be significantly different from control treatment ($n = 4$ wells per condition, $P$ value < 0.01, unpaired independent $t$ test; GFP:RFP and GFP:GFP ratios compared to control treatment changed at least 10%; see "Methods" for details and Dataset EV7 for exact $P$ values). Bars and error bars represent the mean of the median GFP:RFP ratios and the 95% confidence intervals. (E) Results for 9 SLC-protein interactions after interactor cDNA overexpression out of 18 tested interactions analyzed as in (D). (F) KLHL36 and GOLGA5 were strongly enriched in SLC6A15 purifications compared to other SLCs or GFP ($n = 2$ biologically independent replicates). In flow cytometry experiments, SLC6A15-GFP was destabilized by overexpression of the ubiquitin ligase adapter KLHL36 ($n > 4600$ events per sample). (G) MTCH1 was destabilized by E3 ligase MUL1 and stabilized by deubiquitinase USP30. In flow cytometry experiments, MTCH1-GFP protein stability directly correlated with depletion by RNAi or cDNA overexpression of its interactors and their biological functions in the ubiquitin-dependent degradation pathway ($n > 2500$ events per sample). Asterisk and gray figure label of cDNA USP30 condition indicates an effect that was below the set threshold of 10% change in GFP:RFP ratio. See also Fig. EV4.

C-terminal GFP tag from an inducible promoter (Fig. 4A). To measure protein stability by flow cytometry (Fig. 4B), we added an RFP reference protein linked by an internal ribosome entry site to the expression cassette (Yen et al, 2008). To measure changes in subcellular localization of the SLC by microscopy, we further modified the reference RFP to localize either to (1) the entire cell

body; (2) the plasma membrane; (3) mitochondria; or (4) lysosomes, depending on the SLC localization ((Goldmann et al, 2025); see also Table EV4). By using RNA-based interference (loss-of-function) or cDNA overexpression (gain-of-function) of the interaction, we tested effects on localization or abundance of the cognate SLC.

To characterize the cell lines and better understand the effects of protein degradation on the SLCs, we treated 40 of these cell lines with inhibitors of the ubiquitin pathway (TAK-243; (Hyer et al, 2018)), the neddylation pathway (MLN4924; Soucy et al, 2009), or lysosomal acidification (Bafilomycin A1), and measured changes to the GFP:RFP ratio relative to DMSO treatment (Fig. 4C). SLC levels were either unchanged or stabilized by these inhibitors. Applying a threshold of 10% increase in protein levels, only SLC16A6 was stabilized by all three inhibitors, whereas most SLCs were specifically stabilized by either inhibiting ubiquitination, lysosome acidification, or both (Fig. EV4A). Five SLCs showed no significant stabilization after any of the selected treatments, suggesting that they are not regulated by these degradation pathways in HEK 293 cells (Fig. EV4A).

Next, we measured the protein stability of 65 GFP-tagged SLCs after 134 treatments with RNAi and 18 with transient cDNA overexpression, resulting in 152 conditions that we classified into changed or unchanged protein levels. We set a threshold of 10% change in mean GFP:RFP ratio and overall GFP levels compared to control-treated samples at a $P$ value < 0.01. We also excluded measurements if relative RFP levels changed 50% or more of relative GFP levels in the same direction (Dataset EV7). We thus found 37 interactions that showed robust changes in GFP:RFP ratios after RNAi-mediated depletion (Fig. 4D) and 9 interactions after cDNA-mediated overexpression (Fig. 4E). These observations include 29 protein interactions that are functionally linked to ubiquitination, where we found that depletion of E3 ligases and adapter proteins increased SLC levels and depletion of DUB proteins decreased levels in all but three cases. Notably, eight of these interactions were previously reported in large-scale studies (of which four PPIs were identified in AP-MS; Dataset EV7). To our knowledge, none had been functionally assessed before.

We also tested pharmacological inhibition of degradation in addition to depleting or overexpressing the SLC interactor in HEK 293 cells (Appendix Figs. S10 and S11). The increase in protein levels observed after RNAi-mediated depletion was not further increased in some cases, suggesting that the main degrader of a specific SLC had already been eliminated (e.g., Appendix Fig. S10B, SLC2A8–RNF167) or that the interactor was not directly linked to these degradation pathways (e.g., Appendix Fig. S10D, SLC6A15–GOLGA5). In other cases, SLC levels were additionally increased by inhibiting ubiquitination or lysosome acidification (e.g., MTCH2–MUL1; SLC39A8–NFXL1), indicating either incomplete depletion, additional degraders or other stabilization mechanisms.

SLC6A15 was stabilized after treatment with Bafilomycin A1 and MLN4924 (Fig. EV4A) and showed the strongest decrease in stability upon overexpression of its interactor KLHL36 with a reduction of the SLC6A15-GFP to RFP ratio by sixfold (Fig. 4E,F). KLHL36 is a potential cullin-based E3 ligase adapter, and uniquely interacts with SLC6A15 and SLC6A17 in the SLC-interactome dataset (Fig. EV4B). Addition of any of the three inhibitors 6 h preceding analysis partly stabilized the GFP signal, indicating that SLC6A15 protein stability is regulated by a KLHL36-associated cullin-based RING E3 ligase and the autophagy-lysosomal pathway (Fig. 4C). Using an antibody against endogenous SLC6A15 confirmed the degradation by KLHL36 overexpression (Fig. EV4C). These results suggested a protein degradation mechanism that specifically employs a Kelch domain protein, cullin-RING-ligases,

and lysosomes. Other SLCs may also involve a combined degradation mechanism since close to half of the SLCs tested were stabilized by both Bafilomycin A1 and TAK-243 (Fig. EV4A).

The mitochondrial carrier protein MTCH1 was the most strongly stabilized SLC following pharmacological inhibition of ubiquitination (Fig. 4C). When depleting the interactors mitochondrial ubiquitin ligase activator of NFKB1 (MUL1) and Ubiquitin carboxyl-terminal hydrolase 30 (USP30) by RNAi, MTCH1-GFP was stabilized 2.4-fold and destabilized by 1.4-fold, respectively (Fig. 4G). Transient overexpression resulted in the opposite effect, with the E3 ligase reducing and the deubiquitinase increasing MTCH1-GFP levels (Fig. 4G). We obtained similar results for MTCH2-GFP which also interacted with MUL1 and USP30; however, we found that MTCH2 stability was also regulated by another E3 ligase, MARCHF5 (Figs. EV4D and 4D,E). To investigate this further, the stabilization of MTCH2 was also assessed in a HAP1 cell model where endogenous MTCH2 was tagged at its N-terminus with HA-dTAG (HA-FKBP12$^{F36V}$). After depletion of MARCHF5 or MUL1, endogenous MTCH2 stabilization was observed in bulk via western blotting (Fig. EV4E) and specifically at mitochondria via immunofluorescence (Fig. EV4F).

These results highlight the complexity of SLC proteostasis, involving many different protein classes across a wide range of SLC families and suggesting a prominent role of both the proteasome and the lysosome in SLC degradation.

## Destabilization of monocarboxylate transporter SLC16A6 by a phospho-degron

The strong increase of SLC16A6 (MCT7, Monocarboxylate transporter 7) protein levels caused by the neddylation inhibitor MLN4924 suggested involvement of the SCF (SKP1-CUL1-F-box) ubiquitin ligase known to be activated by neddylation (Fig. 4C). The interactome of SLC16A6 contained the CUL1 subunit and its two F-box adapter proteins BTRC and FBXW11 (Frescas and Pagano, 2008) (Fig. 5A), which were unique to SLC16A6 (Appendix Fig. S12A,B). The essential SCF subunit SKP1 was found in the SLC16A6 purification, but since it was found in most SLC purifications and is considered a common contaminant (Mellacheruvu et al, 2013), it was removed post-acquisition (Appendix Fig. S12B). Enrichment analysis of SLC16A6 interactors for GO biological processes resulted in a significant enrichment for the SCF-dependent proteasomal ubiquitin-dependent process term (Appendix Fig. S12C).

Depletion of FBXW11 increased the median levels of GFP-SLC16A6 by 8.5%, below our set threshold for robust changes (Appendix Fig. S12D). Because BTRC and FBXW11 (also known as β-TrCP and β-TrCP2, respectively) are paralogs thought to have redundant roles (Frescas and Pagano, 2008), we analyzed SLC16A6 protein stability after single and combined RNAi treatment of the two F-box proteins (Fig. 5B). Combined treatment with RNAi against BTRC and FBXW11 stabilized SLC16A6 by 1.5-fold compared to control RNAi treatment (Fig. 5B). Additional treatment with MLN4924 did not significantly increase protein levels (Appendix Fig. S12D), indicating that distribution of the SCF$^{β-TRCP}$ complex was mainly responsible for the observed stabilization of GFP-SLC16A6.

Protein degradation via E3 ligases often occurs via specific linear sequence motifs or degrons (Sherpa et al, 2022). Substrates of

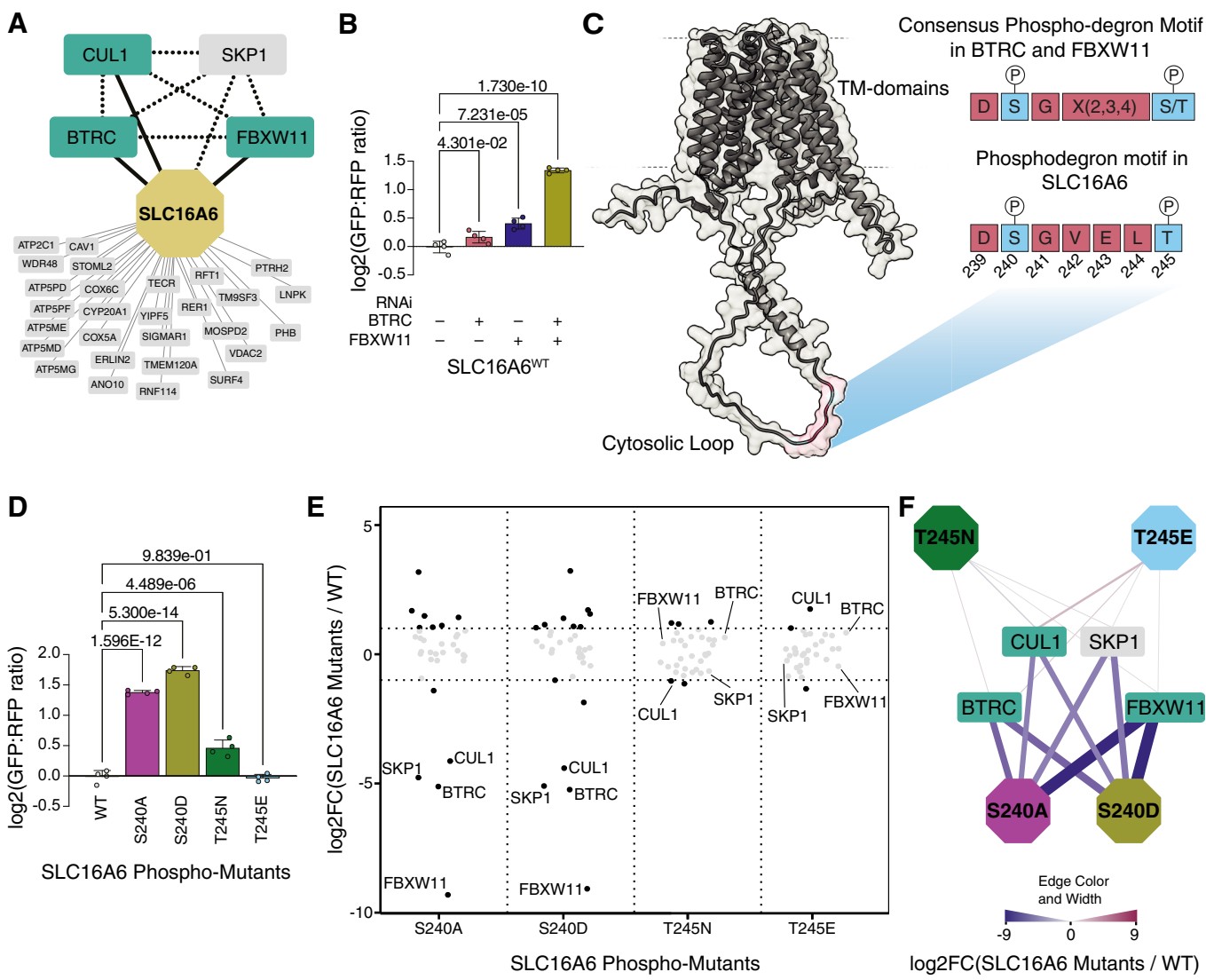

**Figure 5. SLC16A6 stability is regulated by a phospho-degron.**

(A) SLC16A6 interacts with subunits of E3 ubiquitin ligase SCF. SLC16A6 colored in beige, interactors in green. SKP1 (gray) was quantified but not scored and added to the network. Novel edges depicted by solid black lines and edges obtained from literature in black dotted lines. (B) Protein stability of SLC16A6 after RNAi treatment of adapter proteins showing strong stabilization after co-depletion of BTRC and FBXW11 (mean ± SD; $P$ values were calculated using one-way ANOVA and Dunnett test; $n = 4$ replicates). (C) Alphafold predicted structure of SLC16A6 with highlighted cytosolic loop containing the phospho-degron. The βTrCP consensus motif and SLC16A6 phospho-degron motif are visualized on the right side. (D) Protein stability assay of SLC16A6 phosphomutants compared to SLC16A6$^{WT}$ (mean ± SD; $P$ values were calculated using one-way ANOVA and Dunnett test; $n = 4$ replicates). (E) Quantitative comparison of SCF complex subunits across SLC16A6 phosphomutants ($x$ axis) against SLC16A6$^{WT}$. The log2FC ($y$ axis) were derived for the 31 interaction partners against their abundance in SLC16A6$^{WT}$. The dashed black line indicates a ± 1 log2FC threshold. Labeled dots represent SCF complex members ($n = 4$). (F) PPI-Network of SLC16A6 phospho-mutant interactions with SCF complex subunits. The color and thickness of edges indicate the mean log2FC against SLC16A6$^{WT}$. For all AP-MS data shown in the figure panels, $n = 2$ biologically independent replicates with $n = 2$ technical injections were used.

BTRC and FBXW11 share the consensus sequence DSGX(2,3,4)S/T in which the serine residues are phosphorylated to allow binding and subsequent ubiquitination and degradation (Appendix Fig. S12E) (Cardozo and Pagano, 2004; Low et al, 2014); interestingly, SLC16A6 contains a DSGVELT motif at positions 239–245 (Fig. 5C). Experimental structures of SLC16A6 are not yet available, but a structural model by AlphaFold mapped this motif to a cytosolic loop of the transmembrane protein where it could potentially be phosphorylated and recognized by the SCF$^{βTrCP}$ complex (Fig. 5C).

To investigate whether the motif in SLC16A6 is a phosphorylated degron, we individually mutated serine 240 to alanine (S240A) and threonine 245 to asparagine (T245N). We also generated a serine to aspartate (S240D) and a threonine to glutamate (T245E) mutant to potentially mimic phosphorylated residues at these positions and elicit increased degradation. The non-phosphorylatable S240A and T245N mutations increased SLC16A6 levels assessed by flow cytometry, suggesting that the motif targets SLC16A6 for degradation (Fig. 5D). The S240D mutant was the most stable, suggesting that the aspartate

substitution cannot substitute for phosphorylated serine. This is in-line with previous observations that aspartate at the serine position of the DSG motif cannot mimic phosphorylated serine, presumably because the phosphate group is required for hydrogen bonding with βTrCP (Zhao et al, 2010; Wu et al, 2003). The T245E mutant was degraded as efficiently as unmodified GFP-SLC16A6 (Fig. 5D). Assessing SLC16A6 localization by live cell microscopy showed accumulation of the S240A and S240D mutants at the plasma membrane (marked by expression of modified RFP), in-line with the flow cytometry results. While the T245N mutant only showed slight stabilization, the T245E mutant was even less abundant at the plasma membrane than wild-type SLC16A6, suggesting either better recognition by the SCF and improved degradation, or changes in subcellular SLC16A6 distribution (Appendix Fig. S12F).

We depleted BTRC and FBXW11 in the mutant cell lines to ensure that no other degrons were operational (Appendix Fig. S12G). Simultaneous depletion of the two F-box proteins by RNAi did not further stabilize the S240A and S240D mutants, suggesting that serine 240 is the crucial residue determining SLC16A6 stability. The T245N and T245E mutants and wild-type SLC16A6 were stabilized by BTRC and FBXW11 depletion, suggesting that all three proteins encode functional degrons. To investigate whether degron mutations decrease the interaction with SCF proteins, we performed AP-MS on the SH-tagged SLC16A6 mutants. BTRC, CUL1, and FBXW11 were strongly depleted or absent from serine mutants, while the T245E mutant showed increased CUL1 and equal levels of the other SCF subunits as wild-type SLC16A6 (Fig. 5E,F; Appendix Fig. S12H; Dataset EV8). The differential interactome data further emphasized the requirement for SLC16A6 serine 240 and its phosphorylation to ensure efficient recognition of the resulting degron and direct binding by BTRC and FBXW11.

## Proteins affecting the trafficking and subcellular localization of SLCs

To reach their site of action, transmembrane proteins interact with a diverse set of proteins and traverse several subcellular compartments. To examine if this is reflected in the SLC interactome, we performed gene set enrichment analysis of cellular compartment ontology terms and summarized the terms into ten subcellular environments ($P < 0.05$; Fig. EV5A). More than half of SLCs interacted with proteins associated with the ER, with vesicles, and with the plasma membrane where the majority of SLCs localize (Meixner et al, 2020). On average, each SLC interactome was associated with 4.5 out of 10 subcellular compartments, independent of reported intracellular or plasma membrane localization (Fig. EV5B,C), suggesting that AP-MS purification represented a composite of several SLC life cycle locations. Clustering the SLCs based on their respective enriched compartments revealed, among others, subsets of mitochondrial (cluster 20) and lysosomal (cluster 24) SLC interactomes (Fig. EV5D).

To understand their role in trafficking of SLCs, we depleted individual interactors by RNAi and analyzed the subcellular localization of GFP-tagged SLCs in live cells, with strong to moderate correlation between our cytometry and microscopy assays (Fig. 4B; Appendix Fig. S13). Using a co-expressed reference RFP, we calculated the overlap between SLC-GFP and reference signals and measured changes in GFP intensity at RFP-positive and -negative regions. Out of 132 unique interactions analyzed, 73 interactions showed no changes to the SLC-GFP signal after depletion of the interactor. For 59 interactions, corresponding to 36 SLCs, we observed significant changes to the GFP levels or subcellular GFP signal distribution (Dataset EV7).

Among them, thirteen SLCs localized to a single subcellular compartment or were covered by the cell body reference RFP. Interactor depletion for 25 SLC-interactor combinations led to SLC-GFP intensity changes at this location (Fig. 6A). This is exemplified by SLC22A4 and its interaction with the trafficking regulator YIPF3 that when depleted, significantly reduced the SLC22A4-GFP signal at the plasma membrane (Fig. 6A). In another case, we measured SLC1A2 signal at the plasma membrane after depletion of three of its interactors. GFP intensity decreased after GDPD1 and NEK6 RNAi (Fig. 6B).

We used an impedance-based assay of cell morphology (Sijben et al, 2022) to monitor transporter activity of SLC1A2, SLC1A3 (Appendix Fig. S14A–F) and SLC22A3 (Appendix Fig. S14G–I). We tested how glutamate uptake is affected after overexpression of SLC1A2 or SLC1A3 in HEK 293 Jump In T-REx cells. For this we measured the effects of glutamate exposure in SLC1A3 over-expressing cells after depletion of IQSEC1, KCTD2, KCTD5, KCTD17, and NFXL1. From the tested interactors, only depletion of IQSEC1 increased impedance in the assay, suggesting that more glutamate was transported by SLC1A3 in the absence of IQSEC1 (Appendix Fig. S14A). Further, we found that GDPD1 RNAi significantly decreased glutamate uptake in HEK 293 Jump In T-REx cells overexpressing SLC1A2 ($P$ value $= 0.0121$) (Fig. 6C). We also tested whether newly identified interactors of the monoamine transporter SLC22A3 were functionally involved in its transport activity. Using the neurotoxin substrate MPP+, we measured changes to cellular impedance (Mocking et al, 2022) after depletion of the two interactors FRMD5 and HOOK2. When we depleted the membrane-localized and cytoskeleton-associated protein FRMD5, SLC22A3-mediated uptake of MPP+ was significantly reduced (Appendix Fig. S14G,I). Similarly, when we depleted the microtubule binding HOOK2 protein, cellular impedance in response to neurotoxin exposure was greatly reduced (Appendix Fig. S14H). This indicates that SLC22A3 transport activity requires distinct cytoskeletal proteins.

For 22/34 interactions of 23 SLCs localized to multiple compartments, SLC-GFP levels changed indiscriminately at all subcellular locations (Fig. 6D). For instance, plasma membrane- and ER-associated GFP signals for SLC39A8 increased upon depletion of NFXL1. We further tested if the increase affected SLC39A8 transport in a cadmium uptake assay. While NFXL1 RNAi did not change cadmium uptake, depletion of the interactors GOLM1 and, to lesser extent, sorting nexin 25 (SNX25) conferred a significant reduction in fluorescent signal, corresponding to reduced cadmium uptake (Fig. 6E; Appendix Fig. S15). Since GOLM1 and SNX25 did not measurably change SLC39A8 localization or protein levels, they likely regulate the metal transport function of SLC39A8 in a different way.

SLC35F2-GFP localized to the plasma membrane and at the Golgi (Fig. EV5E). In cytometry assays, depletion of sorting nexin 2 (SNX2) in SLC35F2-GFP cells showed the strongest GFP reduction of all 134 protein–protein interactions tested (Fig. 4D). At the subcellular level, SNX2 depletion similarly reduced overall SLC35F2-GFP intensity (Fig. EV5E, quantified in Fig. 6D).

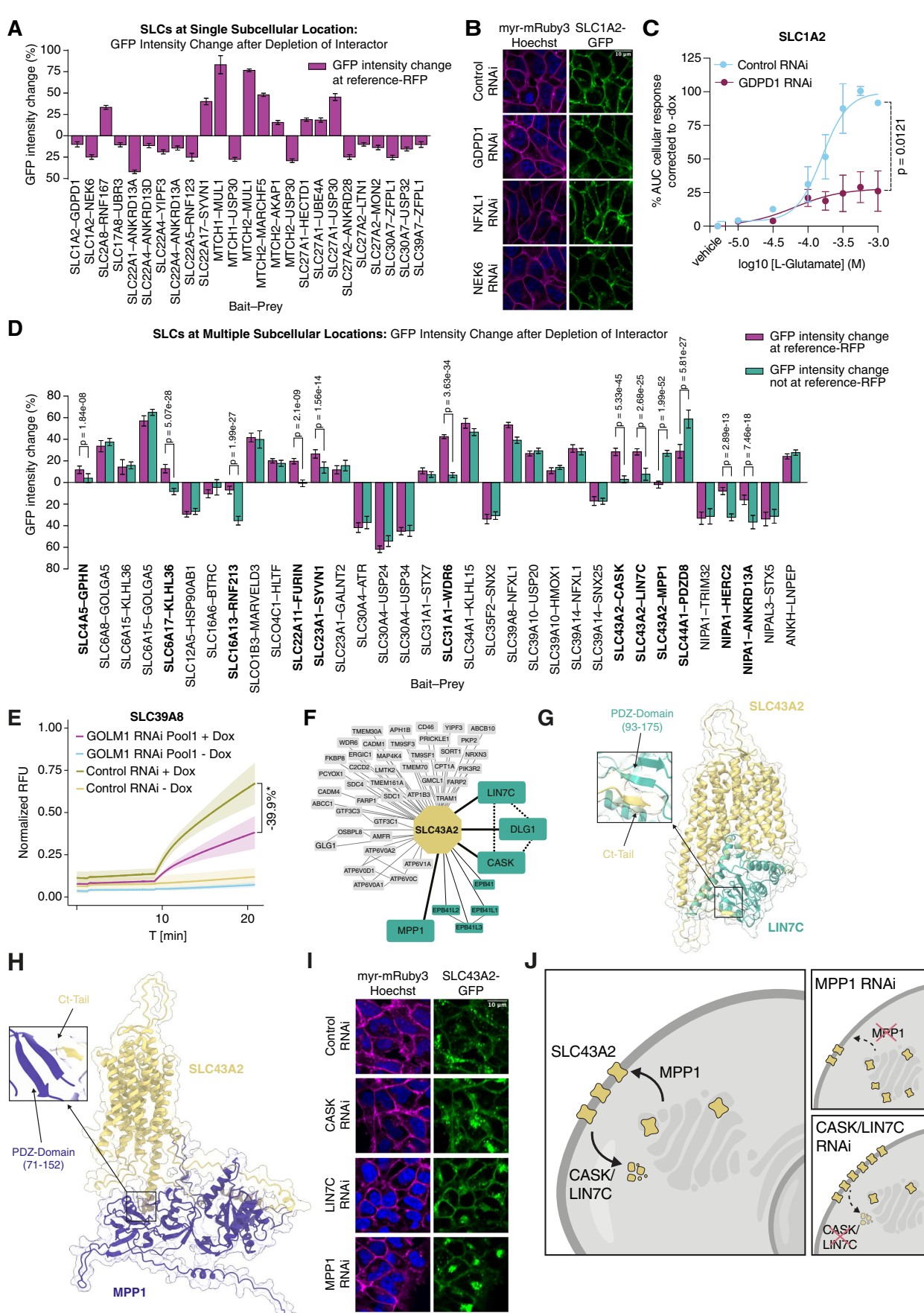

◀ **Figure 6. Proteins affecting trafficking and subcellular localization of SLCs.**

(A) Relative GFP intensity changes at reference regions after interactor RNAi compared to control RNAi are plotted as a percentage ($n = 160$ images per condition; error bars denote 95% confidence intervals). The graph includes SLCs residing at single subcellular locations and SLC30A7 (since vesicles and Golgi locations are both covered by expression of cell body RFP). GFP intensity changes at the respective reference signal were thresholded at 10% increase or decrease over control treatment with $P$ value < 0.01 (independent $t$ test). Subcellular compartment location of SLC-GFP and reference RFP as well as exact $P$ values are indicated in Dataset EV7. (B) Representative images of plasma membrane-localized SLC1A2-GFP after depletion of three interactors (scale bar, 10 µm). GFP signal intensity was quantified using myr-mRuby3 to identify plasma membrane pixels. (C) Quantification of cellular impedance of HEK 293 SLC1A2-SH cells treated with increasing concentrations of glutamate after GDPD1 RNAi ($n = 3$ independent experiments). Data were corrected against samples without doxycycline due to significant ʟ-glutamine transport by endogenous SLC1A3 in HEK 293 cells and are shown as background corrected mean area under the curve (AUC) quantification relative to control RNAi ±SEM. $P$ value was calculated using unpaired $t$ test. (D) Relative GFP intensity changes at the reference region (purple) and outside the reference region (teal) for 34 SLC-interactor RNAi combinations for SLC-GFP signals covering more than one subcellular compartment (as indicated in Dataset EV7; $n = 160$ images per condition; bars denote mean ± 95% confidence intervals). The graph includes interactions with GFP intensity changes exceeding 10% at the reference region or 25% outside the reference region with $P$ values < 0.01 compared to control RNAi (independent $t$ test). For 13 interactions highlighted in bold, GFP intensity changes at reference and non-reference regions differed significantly ($P$ values as indicated, independent $t$ test with Benjamini–Hochberg correction for multiple tests; relative change at least 33.3% of absolute higher value), indicating location-specific changes. GFP intensity changes for other interactions were similar for reference and non-reference regions, indicating general changes in SLC abundance after interactor depletion. (E) Relative fluorescence unit (RFU) curve to measure cadmium uptake of SLC39A8 after RNAi against GOLM1. Traces represent mean min-max normalized RFU values across replicates ±SD (curve shades; $n = 3$ biologically independent replicates with 8 technical replicates). Results are compared to control RNAi using mean AUC calculated from normalized RFU traces considering all replicates (Appendix Fig. S15). (F) SLC43A2 interactions with trafficking proteins CASK, DLG1, LIN7C, and MPP1 (teal; $n = 2$ biological replicates with $n = 2$ technical replicates). Novel edges are depicted by solid lines and edges supplemented from BioGRID are shown with dotted line. (G) AlphaFold model of SLC43A2–LIN7C complex with section of the interaction of the LIN7C PDZ domain and the intracellular C-terminal tail of SLC43A2 (Weighted Score 0.57). (H) AlphaFold model of SLC43A2–MPP1 interaction between the MPP1 PDZ domain and the intracellular C-terminal tail of SLC43A2 (Weighted Score 0.51). (I) Representative images of plasma membrane and vesicle localized SLC43A2 after depletion of three interactors (scale bar, 10 µm). Depletion of CASK or LIN7C increased plasma membrane-associated GFP signal whereas RNAi of MPP1 increased the intensity and relative distribution of SLC43A2-GFP away from the plasma membrane (quantified in (D)). (J) Schematic of SLC43A2 trafficking regulation.

SLC35F2 is the main importer of the experimental antitumor compound YM-155 (sepantronium bromide; (Winter et al, 2014)). In a viability assay based on transport of the toxic compound YM-155 by SLC35F2, we observed that expression of SLC35F2-GFP enhanced toxicity under drug treatment conditions, whereas depletion of SNX2 partially alleviated some of the effect, likely by decreasing the SLC35F2 levels at the plasma membrane (Fig. EV5F).

For 12/34 interactions of 23 SLCs localized to multiple compartments, SLC-GFP levels changed specifically at certain subcellular compartments (Fig. 6D). This included the organic anion transporter SLC22A11 and its interaction with the endoprotease FURIN (Fig. EV5G). Depletion of FURIN increased SLC22A11-GFP intensity specifically at the plasma membrane and not at the Golgi. Another example was the vitamin C transporter SLC23A1 localized in its GFP-tagged form at the plasma membrane and the lysosome (Fig. EV5H). While depletion of GalNAc transferase GALNT2 increased GFP intensity at the lysosome and at non-lysosomal regions (positive and negative for RFP, resp.), depletion of ubiquitin ligase SYVN1 predominantly increased lysosomal SLC23A1-GFP intensity (Fig. 6D), suggesting that ER quality control by SYVN1 is required for trafficking to the plasma membrane. Similarly, the copper importer SLC31A1 was found to be differentially regulated by two of its interactors. SLC31A1-GFP localized at the plasma membrane and in cytoplasmic vesicles. Depletion of syntaxin-7 (STX7) significantly increased both of these pools, while depletion of SLC31A1 interactor WDR6 increased its levels at the plasma membrane (Fig. EV5I, quantified in Fig. 6D).

Besides sorting nexins, proteins of the MAGUK (membrane-associated guanylate kinase) family were also strongly featured in the SLC interactome, most notably associating with the amino acid transporter SLC43A2. In its conditionally expressed GFP-tagged form, SLC43A2 localized to the plasma membrane and to cytoplasmic vesicles (Table EV4). In our co-purification analysis (Appendix Fig. S9), prominent interactions emerged with the members of the LIN- and CASK/DLG1 complex (LIN2/CASK, LIN7/LIN7C, SAP97/DLG1 complex, CORUM (Tsitsiridis et al, 2023)), and with MPP1

complex members (GPC, 4.1 R/EPB41, p55/MPP1 complex, CORUM) (Fig. 6F; Dataset EV7). Proteins in both complexes are characterized by PDZ, SH3 or L27 interaction domains, are kinases or kinase-like proteins, and can interact with the actin or microtubule cytoskeleton to concentrate, polarize or recycle proteins at the plasma membrane (Alewine et al, 2007). These functions are thought to have emerged early in evolution to allow tissue formation (Baines et al, 2014) and are disease-related in the context of brain development (CASK) and hematology (EPB41) (Dubbs et al, 2022). Interestingly, the interactions between the cytoplasmic C-terminal tail of SLC43A2 and the PDZ domains of LIN7C and MPP1 were modeled with medium confidence using AlphaFold (Fig. 6G,H; Appendix Fig. S16). When we depleted the protein complex subunits by RNAi, SLC43A2-GFP was affected differently depending on which PDZ-domain protein was targeted. While depletion of LIN7C and CASK increased plasma membrane-associated SLC43A2 by almost 30%, depletion of MPP1 did not affect plasma membrane-associated SLC43A2 (Fig. 6D,I). Instead, we found locally increased signals in cytoplasmic structures reminiscent of accumulating SLC43A2 in the Golgi (Fig. 6I). This suggested that we observed two distinct interaction hubs involved in trafficking: one containing the membrane-associated guanylate kinase MPP1, possibly involved in shuttling SLC43A2 from the Golgi to the plasma membrane, and another containing LIN7C and CASK, potentially increasing turnover of SLC43A2 at the plasma membrane (Fig. 6J).

In summary, we showed that many interactors identified in this study contributed to the function of SLC transporters, by influencing localization, protein levels or activity. Our findings can serve as a starting point for more in-depth research and for possible therapeutic modalities.

## Discussion

The ensemble of proteins encoded by the large number of human genes (1500–2000) estimated to govern the transport of chemical substances across all kinds of cellular membranes can be termed the

human transportome (Huang et al, 2004). It can be reasonably assumed that most chemical integration of cells, organs and organisms with the respective environments is orchestrated by this transportome. Yet we lack answers to even some of the simplest questions about this interface: what do the individual gene products transport? How is their activity regulated? Does integration of function occur within each system? Characterization of the protein environment of any given protein should, without much prior knowledge on function, represent a valuable way to provide links to the biochemical and biological processes to which the gene product is involved. The protein partners shed light on the subcellular environment, participation to specific molecular machines, potential obligate and facultative partners, and possible modifiers of activity, stability, and localization (Marsh and Teichmann, 2015; Gavin and Superti-Furga, 2003).

Among the transportome, solute carriers represent the largest superfamily with roughly 450 members. As part of a larger effort to start elucidating the transportome systematically, we present here the first comprehensive protein interaction study of the human solute carrier superfamily. We generated tagged versions of each SLC under a doxycycline-inducible promoter, using codon-optimized sequences, and used an experimental AP-MS workflow tailored to TM-proteins. Throughout the study, we address commonly experienced limitations when studying TM-proteins, including expression of the tagged protein, lysis conditions, purification and elution of enriched complexes. To predict protein interactions in a reproducible manner, we devised a machine learning-based method which integrates common PPI-scores (including SAINT, CompPASS) with several quantitative and uniqueness features, thereby overcoming limitations of previous studies which relied solely on GFP-expressing controls or only a few other transporters (Rebsamen et al, 2015; Heinz et al, 2020). Together, these methodological developments allowed us to identify thousands of novel PPIs and generate the first comprehensive protein interaction landscape of SLCs.

Was the effort successful? We would argue that the effort has been technically successful, even if very labor-intensive, and that it represents a very significant resource for the community. Has the guilt-by-association concept allowed us to "deorphanize" many transporters? The simple answer is no. Unlike GPCRs or nuclear hormone receptors, SLCs are thought to be much more promiscuous, as the sheer number of diverse chemical substances entering the human body vastly exceeds the number of transportome genes. Moreover, the functional proteomics approach presented here is not suitable for assigning relationships to potential cargos directly. However, the combination of similarity clustering, enrichment analysis for SLC properties and subsequent analysis of cluster-specific PPI-networks allowed us to derive novel insights into functions of several SLC families. The similarity clustering used in our study is limited due to the heterogenous expression and localization of SLCs and the lack of reciprocal AP-MS experiments, inherently resulting in a low-density network. It was previously shown that high-density networks are needed to deconvolute binary PPI-networks to protein complexes (Uliana et al, 2023; Buljan et al, 2020; Salokas et al, 2022). Similarity-based clustering on the interactome profile of each tagged protein is, on the other hand a rarely used approach for interaction proteomics, as it is strongly penalized by data sparseness. Despite these shortcomings, the clustering strategy of our AP-MS strategy was successful in

positioning each SLC within a particular protein environment that provides important insights into its involvement in cellular and biochemical processes.

By detecting interactors across the SLC lifespan, the SLC interactome can serve as a roadmap to modulate targeting, trafficking, activity and abundance levels of SLCs through their interaction partners. This is of critical interest for diseases associated with pathogenic SLC transporter expression linked to folding-deficient variants (Wiktor et al, 2021; Ohtsubo et al, 2011), trafficking/targeting errors (Rogala-Koziarska et al, 2019) or proteostatic pathways (Colaco et al, 2023; Xu et al, 2016). We thus envision that the identified interaction partners may serve as drug discovery route to modulate SLC-activity indirectly, by rescuing folding and/or assemblies by correctors, pharmaco-chaperones and potentiators (Gautherot et al, 2012; Bhat et al, 2021, 2019).

Our validation efforts assess the value and quality of the SLC-interactome dataset by testing subcellular localization and relative protein levels as a proxy for protein stability. We tested the functional impact of newly discovered interactions for 137 SLC-protein interactions, a large number for AP-MS studies.

Follow-up studies using endogenously modified proteins, such as shown in this study for MTCH2 and its interactors, in a cellular system expressing both interaction partners will be required to confirm the function of these interactions. Currently, scarcity of well-working antibodies against SLCs is a limiting factor. We have recently promoted the development of recombinant high-affinity binders directed against SLCs as alternative (Gelová et al, 2024). Using available antibodies, we were able to confirm that SLC6A15 and its interaction with KLHL36 occur at physiological SLC expression level. The sodium-dependent neutral amino acid transporter SLC6A15 and E3 ligase adapter KLHL36 could also be linked via disease susceptibility. While loss of SLC6A15 function was reported to be associated with depression (Kohli et al, 2011), increased KLHL36 expression has been tentatively correlated with suicide attempts (Han et al, 2023). KLHL36 SNPs have been linked to increased resilience in US Army soldiers (Stein et al, 2019). Similar individual, or systematic investigations of such connections can become valuable tools for research and discovery.

Ectopic expression of bait proteins for AP-MS can also have unexpected benefits. βTrCP proteins play essential cellular roles such as cell cycle control, DNA damage repair, metabolism and signaling, and are considered both oncogenes and tumor suppressor genes. While their substrates have been studied in great detail using AP-MS (Kim et al, 2015; Dorrello et al, 2006; Low et al, 2014) and BioID (Coyaud et al, 2015), SLC16A6 has not previously been reported as an SCF$^{βTrCP}$ substrate. Nevertheless, our data clearly suggest a physiologically relevant relationship between SLC16A6 and SCF$^{βTrCP}$. Interestingly, SLC16A6 expression is confined to certain neuronal and reproductive human tissues and its expression is hardly detectable in common human cell lines, including most cell lines used in the RESOLUTE project (Karlsson et al, 2021). This suggests that screening approaches using ectopic expression may yield biologically relevant insights on protein homeostasis that are difficult to detect using only endogenous components.

Overall, we conclude that SLCs are extensively regulated by the cellular proteostatic machinery, affecting their levels and localization.

There are some limitations to our study that need to be considered. First, we relied for extraction of SLC complexes on NP-

40/Ipegal as detergent, except for mitochondrial-localized transporters, when we used digitonin. Previous interaction proteomics studies focusing on transmembrane proteins used DDM (n-dodecyl-β-maltoside) or C12E8 (octaethylene glycol monododecyl ether) to extract under native lysis conditions TM-localized complexes (Celis-Gutiérrez et al, 2019). We used NP-40 to have comparable lysis conditions with other large-scale studies (Huttlin et al, 2021, 2015). Certainly, different detergents will result in different interactomes.

Second, we used a modified AP-MS workflow to recover low-abundant mitochondrial baits by performing organellar enrichment for mitochondria prior to affinity enrichment (see "Methods" for details). However, we did not implement subcellular enrichment for SLCs localized to other membrane-bound organelles. The diverse subcellular localization—ranging from a single organelle to multiple compartments—and the broad range of expression levels across the SLC superfamily may have impacted the purification of low-expressed SLCs using our standard high-throughput AP-MS workflow. To partially mitigate this, we increased the lysate input for low-expressed SLCs (see "Methods" for details). Third, for generating our interactome dataset, we relied on an overexpression system for the tagged SLC baits. Although this is comparable to other PPI-mapping approaches (BioID, APEX) and necessary due to the lack of sufficient affinity binders for SLCs, it may not accurately represent endogenous expression levels and lead to identifying PPIs related to overexpression, such as trafficking and folding-associated interactors. Using cultured cells, while beneficial for systematically profiling the entire family in one background, may not fully capture the complexity and physiological relevance of PPIs of not endogenously expressed SLCs. Several well-characterized interactions of SLCs are tissue/cell line-specific, such as TASL-SLC15A4 (Heinz et al, 2020) or ACE2-SLC6A19 (Danilczyk et al, 2006), which we missed with our approach.

Fourth, we focused our analysis on the consensus protein sequence for SLCs and interactors, therefore excluding isoforms. It was previously shown that SLC isoforms vary not only in their expression patterns regarding cell type or tissue, but they also can have different lengths of C- and N-terminus, which may affect SLC localization and interactions (Shirakabe et al, 2006; Mazurek et al, 2010; Yoo et al, 2020). Finally, because of the elaborate and conservative scoring of interaction partners, we might have misassigned true-positive interaction partners wrongly as background. Benchmarking of our dataset was hampered due to low coverage of SLCs in protein complex databases and considerable variability of SLC-protein interactions reported in public PPI-databases.

A valuable additional layer in the characterization of the molecular environment of transporters would be the assessment of lipid-protein interactions (Corradi et al, 2019). There are emerging data supporting the importance of phospholipid composition affecting activity of transmembrane transporters (Hresko et al, 2016). Recent developments in proteome-wide mapping of protein–metabolite interactions by LiP-MS and other methods have great potential in this sense (Piazza et al, 2018). The public release of our data will further facilitate integration of SLC-protein interactions, likely to offer new appreciation of the modular organization of the interactome. Notably, our dataset serves as a starting point for orthogonal protein interaction mapping approaches, such as APEX (Hung et al, 2016), MaMTH (Petschnigg et al, 2014), and SIMPL (Yao et al, 2024), to further validate and complement the SLC interactome. Complementation/split assays can capture transient interactions which are often missed by AP-MS. A combination of approaches enables to generate a more comprehensive PPI-landscape of SLCs.

We think that it would be intriguing to study the SLC interactome across different cellular states, thus using AP-MS to identify dynamic rewiring of the PPI-network upon changes of the transporter activity or in metabolic state. Attractive possibilities include changes in media compositions, for example, alternative carbon sources or concentrations of amino acids, and performing similar mapping campaigns across these different settings. Such experiments may lead to the identification of links between cellular signaling and modulation of transporter activity.

A searchable version of the SLC-interactome dataset is available at https://re-solute.eu/resources/dashboards/proteomics/network/. The resource allows exploration of interactions, further filtering and integration of interactions obtained from BioGRID and CORUM to allow visualization of protein complexes.

Our study should contribute to the general understanding of the interactome of membrane-embedded proteins in human cells. Seminal previous studies have focused on a clinically relevant section of GPCR using yeast two-hybrid (Sokolina et al, 2017) or have been dedicated to individual baits or specific relationships. Thus, it can be expected that this work may represent a blueprint for the systematic mass spectrometry-based characterization of the interactome of several other groups of multipass TM-proteins, such as transporters of the ABC, P-type ATPase and aquaporin groups, ion channels, mechanosensitive receptors, and GPCRs (von Heijne, 2007). Because of the diversity of the biochemical and biophysical properties of the membranous environment, including the dimensional limitations to diffusion, it would not be surprising if protein–protein interaction principles established mainly with soluble proteins would be different for multipass membrane proteins. We are now using artificial intelligence-driven approaches to extract interaction modules or "archetypes" that may allow to better predict interactions of membrane proteins generally.

# Methods

**Reagents and tools table**

| Reagent/resource | Reference or source | Identifier or catalog number |
|---|---|---|
| **Experimental models** | | |
| Jump In T-REx HEK 293 cells | Invitrogen™ | CVCL_0045 |
| Huh-7 | JCRB | CVCL_0336 |
| SK-MEL-28 | ATCC | CVCL_0526 |
| 1321N1 | ECACC | CVCL_0110 |
| LS180 | ECACC | CVCL_0397 |
| MDA-MB-468 | MD ANDERSON | CVCL_0419 |
| HCT 116 | ATCC | CVCL_0291 |
| HAP1 (C631) | Horizon Discovery | C631 |
| **Recombinant DNA** | | |
| pJTI R4 DEST CMV-TO pA vectors-CT | Sijben et al, 2022 | NA |

| Reagent/resource | Reference or source | Identifier or catalog number |
|---|---|---|
| pJTI R4 DEST CMV-TO pA vectors-NT | Sijben et al, 2022 | NA |
| **Antibodies** | | |
| Rabbit anti-SLC6A15 | Abcam | Cat# ab191192 |
| Rabbit anti-CUL3 | Bethyl Laboratories | Cat# A301-109A |
| Mouse anti-alpha tubulin | Abcam | Cat# ab7291 |
| Mouse anti-FLAG | Sigma-Aldrich | Cat# F1804 |
| Anti-HA peroxidase conjugate | Sigma-Aldrich | Cat# H6533 |
| Mouse anti-alpha 1 Sodium Potassium ATPase | Abcam | Cat# ab7671 |
| **Oligonucleotides and other sequence-based reagents** | | |
| RNAi pools | This study | Table EV5 |
| PCR primers | This study | Table EV6 |
| cDNAs | This study | Table EV7 |
| SH-quant ([NH2] AADITS(L)Y(K)[COOH]: Heavy Leu: 13 C(6)15 N(1) [ + 7]) and SH-quant* ([NH2]AADITS(L)Y(K) [COOH]: Heavy Leu: 13 C(6) 15 N(1)[ + 7] and heavy C-term Lys: 13 C(6)15 N(2) [ + 8]) | Wepf et al, 2009 | NA |
| **Chemicals, enzymes, and other reagents** | | |
| PNGase F | NEB | P0704 |
| **Tissue culture** | | |
| Dulbecco's Modified Eagle's Medium – high glucose (DMEM) | Sigma | D5796-500mL |
| Fetal Bovine Serum (FBS) | Biowest | S1810-5000 |
| Penicillin-Streptomycin (10,000 U/mL) | Gibco | 15140-122 |
| RPMI-1640-Medium | Sigma-Aldrich | R8758-500ML |
| McCoy 5A-Medium | Sigma-Aldrich | M8403-500ML |
| Plasticidin HCl | InvivoGen | ant-bl-5b |
| G418 disulfate salt (Geneticin) | Sigma-Aldrich | A1720-5G |
| Doxycycline hyclate | Sigma-Aldrich | D9891-1G |
| 10 cm cell culture dishes | Corning | 430167 |
| 15 cm cell culture dishes | Corning | 430599 |
| 1× Dulbeccos PBS without Ca & Mg 1×500 ml | Sigma | D8537-500ML |
| Cell Scrapers | VWR | 734-2604 |
| 15 mL Falcon tubes | Falcon | 352096 |
| 50 mL Falcon tubes | Falcon | 352070 |
| Trizma-Base | Sigma | T1503-5KG |
| **MS sample preparation** | | |
| Covaris S2 high-performance ultrasonicator | Covaris | NA |
| 4-Morpholinepropanesulfonic acid (MOPs) | Ambion | AM9570 |
| Ethylenediaminetetraacetic acid (EDTA) | Sigma | E6758-500G |
| Sucrose | Merck | 1.07654.1000 |
| Branson digital sonicator | Branson | digital Sonifier |

| Reagent/resource | Reference or source | Identifier or catalog number |
|---|---|---|
| 1.5 mL tubes | Eppendorf | 0030 120.086 |
| 2 mL tubes | Eppendorf | 0030 120.094 |
| Water, for HPLC LC-MS Grade | VWR | 83645.320 |
| Snap Cap Low Retention Microcentrifuge Tubes | Thermo Fisher Scientific | #3448 |
| HEPES | Carl Roth | 9105.3 |
| NaCl | Sigma-Aldrich | s7653-5K |
| Alternative NP-40 | Merk | 492016-100 mL |
| Digitonin | Sigma | D141-500MG |
| cOmplete™, EDTA-free protease inhibitor cocktail | Roche | 05056489001 |
| PMSF | BioChemica | A0999.0005 |
| Avidin | IBA | 2-0204-015 |
| Phosphatase inhibitor cocktail 2 | Sigma | P5726-5ML |
| Phosphatase inhibitor cocktail 3 | Sigma | P0044-5ML |
| Sodium lauryl sulfate solution 10% | Merck | 71736-500 m |
| StrepTactin Sepharose (50% suspension) | IBA | 2-1201-010 |
| Bio-Rad Protein Assay Dye Reagent Concentrate | Bio-Rad | 5000006 |
| Micro BCA™ Protein Assay Kit | Thermo Fisher Scientific | 23235 |
| Protein standard I | Bio-Rad | 5000005 |
| Mini Bio-Spin™ Chromatography Columns | Bio-Rad | 732-6207 |
| Sodium dodecyl sulfate (SDS) | Sigma | 436143 |
| DTT (DL-Dithiothreitol), for molecular biology, minimum 99% | Sigma-Aldrich | D9779-5G |
| Urea | Sigma-Aldrich | U0631-500G |
| Tris(hydroxymethyl) aminomethane (Tris/HCl) ultrapure grade ≥99.9% | Sigma-Aldrich | 154563 |
| Microcon 30, Ultracel YM-30 | Merck Millipore | MRCF0R030 |
| Iodacetamide (IAA), SigmaUltra | Sigma-Aldrich | I1149-5G |
| Triethylammonium Bicarbonate Buffer (TEAB), 1 M, pH 8.5 | Sigma-Aldrich | 17902 |
| SpeedBeads™magnetic carboxylate modified particles | GE Healthcare | 45152105050250 |
| SpeedBeads™magnetic carboxylate modified particles | GE Healthcare | 65152105050250 |
| Acetonitril for HPLC LC-MS Grade | VWR Chemical | 83640.32 |
| Ehanol absolute Product: 1.00983.1000 | MERCK | 1.00983.1000 |
| Trypsin/Lys-C Mix, Mass spec Grade | Promega | # V5073 |
| Sequencing Grade Modified Trypsin | Promega | # V5113 |

| Reagent/resource | Reference or source | Identifier or catalog number |
|---|---|---|
| TFA Uvasol (trifluoroacetic acid) | MERCK | 1.08262.0100 |
| Empore™SPE Disksmatrix active group C18, diam. 47 mm, pk of 20 | Sigma-Aldrich | 66883-U |
| MicroSpin columns, SS18V, 30 –300 µg capacity | Nest Group Inc | NC0194358 |
| Gemini-NX C18 column | Phenomenex, Inc. | 00B-4453-E0 |
| Agilent 1200 series HPLC | Agilent | NA |
| Pierce™ Quantitative Peptide Assays & Standards | Thermo Fisher Scientific | 23290 |
| **MS data acquisition** | | |
| Dionex Ultimate 3000 RSLCnano system | Thermo Fisher Scientific | NA |
| Nanospray Flex Ion Source | Thermo Fisher Scientific | NA |
| PepMap 100 C18, 5 µm, 5 × 0.3 mm | Thermo Fisher Scientific | NA |
| 0.1% TFA | Sigma-Aldrich Chemie, GmbH, Germany | # 34978-2.5L-R |
| Suprapur® Formic acid 98–100% | MERCK KgaA | 1.11670.1000 |
| Fused Silica Capillary ID 75 µm, 50 cm length | Polymicro Technologies LLC | TSP075150 |
| ESI Emitter Fused Silica: 20 µm ID × 7 cm L × 365 µm OD; Orifice ID: 10 µm | CoAnn Technologies, United States | TIP1002010C-5 |
| ReproSil-Pur 120 C18-AQ, 3 µm | Dr. Maisch, Ammerbuch-Entringen, Germany | # r13.aq. |
| Sonation column oven PRSO-V1 | Sonation lab solutions | NA |
| Q Exactive Orbitrap Mass Spectrometers | Thermo Fisher Scientific | NA |
| Q Exactive HF-X Hybrid Quadrupole-Orbitrap MS System | Thermo Fisher Scientific | NA |
| Orbitrap Fusion Lumos Tribrid Mass Spectrometer | Thermo Fisher Scientific | NA |
| PepMap C18, 500 mm × 75 µm ID, 2 µm, 100 Å | Thermo Fisher Scientific | # 164570 |
| Butterfly heater, PST-BPH | Phoenix S&T, Inc | NA |
| Column heater controller, PST-CHC | Phoenix S&T, Inc | NA |
| **Validation by transporter assay** | | |
| AquaMax DW4 Microplate Washer | Molecular Devices | NA |
| FLIPR® Calcium 5 Assay Kit | Molecular Devices | R8186 |
| FDSS (HAMAMATSU, Functional Drug Screening System) 7000EX | Hamamatsu Photonics K.K. | NA |
| **Validation of interactions by cytometry** | | |
| BD LSRFortessa | BD Biosciences | NA |
| BD High Throughput Sampler | BD Biosciences | NA |
| GFP filter (B/E AF 488, 505LP and 530/30 filters) | BD Biosciences | NA |
| RFP (YG/D PE-TexasRed, 600LP and 610/20 filters) | BD Biosciences | NA |

| Reagent/resource | Reference or source | Identifier or catalog number |
|---|---|---|
| BFP (V/C Pacific Blue, 450/50 filter) | BD Biosciences | NA |
| TAK-243 | Selleckchem | S8341 |
| MLN4924 | MedChemExpress | 905579-51-3 |
| Bafilomycin A1 | Enzo LifeSciences | BML-CM110 |
| **Validation of interactions by microscopy** | | |
| Opera Phenix | Perkin Elmer | NA |
| Hoechst signal acquisition at 405 nm excitation (435–480 nm emission filter) | Perkin Elmer | NA |
| GFP signal acquisition at 488 nm excitation (500–550 nm emission filter) | Perkin Elmer | NA |
| RFP signal acquisition at 561 nm excitation (570–630 nm emission filter) | Perkin Elmer | NA |
| **Full transcriptome profiling and RT-qPCR for DsiRNAs** | | |
| RLT buffer | Qiagen | 79216 |
| β-mercaptoethanol | Sigma | 444203-250 ML |
| RNAeasy Mini Kit | Qiagen | 74104 |
| DNase I | Qiagen | 79254 |
| Bioanalyzer RNA Analysis | Agilent | 5067-1511 |
| NEBNext® Ultra™ II Directional RNA Library Prep Kit for Illumina® | New England Biolabs | E7760S |
| NEBNext® rRNA Depletion Kit | New England Biolabs | E6310L |
| RiboLock RNase inhibitor | Thermo Fisher Scientific | EO0382 |
| DNase I, RNase-free (1 U/µL) | Thermo Fisher Scientific | EN0521 |
| RevertAid First Strand cDNA Synthesis Kit | Thermo Fisher Scientific | K1622 |
| Luna® Universal qPCR Master Mix | New England Biolabs | M3003S |
| **Databases and online resources** | | |
| BioGRID | https://thebiogrid.org/ Oughtred R et al, 2021 | NA |
| BioPlex | https://bioplex.hms.harvard.edu/ Huttlin EL et al, 2021 | NA |
| STRING | https://string-db.org/ Szklarczyk D et al, 2023 | NA |
| IID | https://iid.ophid.utoronto.ca/ Kotlyar et al, 2016, 2022 | NA |
| IntAct | https://www.ebi.ac.uk/intact/home N Del Toro et al, 2022 | NA |
| UniProtKB | https://www.uniprot.org/ The UniProt Consortium et al, 2023 | NA |
| CORUM | https://mips.helmholtz-muenchen.de/corum/ Tsitsiridis G et al, 2023 | NA |
| **Software** | | |
| XCalibur™ Software (version 4.1.31.9) | Thermo Fisher Scientific | NA |

| Reagent/resource | Reference or source | Identifier or catalog number |
|---|---|---|
| XCalibur version 4.3.73.11 | Thermo Fisher Scientific | NA |
| Tune 2.9.2926 | Thermo Fisher Scientific | NA |
| Tune 2.13.3162 | Thermo Fisher Scientific | NA |
| MSConvert (3.0.21128-7376ae988). | https://proteowizard.sourceforge.io Chambers et al, 2012 | NA |
| Philosopher v3.4.13 | https://philosopher.nesvilab.org da Veiga Leprevost et al, 2020 | NA |
| MSFragger-3-3 | https://msfragger.nesvilab.org/ Kong et al, 2017 | NA |
| Trans-Proteomic Pipeline (TPP v5.2.1-dev Flammagenitus) | http://www.tppms.org/ Deutsch et al, 2023 | NA |
| PeptideProphet | http://www.tppms.org/ Keller et al, 2002 | NA |
| ProteinProphet | http://www.tppms.org/ Nesvizhskii et al, 2003 | NA |
| Spectronaut (version 17.1.221229.55965, Quasar | Biognosys | NA |
| CompPASS | https://github.com/dnusinow/cRomppass Sowa et al, 2009 | NA |
| SAINTexpress_v3.6.3 (2018-03-09) | https://saint-apms.sourceforge.net/Main.html Teo et al, 2014 | NA |
| Cytoscape version 3.0.8 | http://www.cytoscape.org Shannon et al, 2003 | NA |
| AlphaFold2 multimer (version 2.3) | google-deepmind Evans et al, 2021 | NA |
| AlphaFold 3 (version 3.0.0) | google-deepmind Abramson et al, 2024 | NA |
| UCSF ChimeraX 1.7.1 | https://www.rbvi.ucsf.edu/chimerax/ Pettersen et al, 2021 | NA |
| Cutadapt (1.18) | Martin M, 2011 https://cutadapt.readthedocs.io/en/stable/ | NA |
| STAR (2.6.1a) | Dobin et al, 2013 https://github.com/alexdobin/STAR?tab=readme-ov-file | NA |
| Kallisto (0.44.0) | Bray et al, 2016 https://pachterlab.github.io/kallisto/ | NA |
| Prism 9.0 | GraphPad software https://www.graphpad.com/ | NA |
| R4.3.1 | https://www.r-project.org/ | NA |
| RStudio 2023.06.1 | https://posit.co/download/rstudio-desktop/ | NA |
| FlowJo™ version 10.9.0 | BD Biosciences | NA |

## Cell line generation

Jump In T-REx HEK 293 cells (Thermo Fisher; RRID:CVCL_YL74) were co-transfected with pJTI R4 DEST CMV-TO pA vectors containing strepII-HA-tagged (C-terminal) or HA-strepII-tagged (N-terminal), codon-optimized SLC cDNA sequences and the pJTI

R4 Int vector that encodes for the R4 Integrase according to the manufacturer's instructions (for all SLC cDNA sequences see Table EV1). For more information, see (Wiedmer et al, 2025).

For validation assays, HEK 293 Jump In T-REx cells were co-transfected with the pJTI R4 Int vector and with pJTI R4 DEST CMV-TO pA vectors containing GFP-tagged SLC sequences, and a reference RFP separated by an internal ribosome entry (IRES) sequence. The RFP (mCherry or mRuby3) was preceded either by a myristylation tag for plasma membrane localization, four COX8 presequences for mitochondrial localization, the TMEM192 coding sequence for lysosome localization, or a myristylation tag with an in-frame upstream start codon for pan-cellular localization (GFP cell lines see Table EV4, localization tag determined based on (Meixner et al, 2020) and (Goldmann et al, 2025). The GFP tag was located at the C-terminus except for SLC6A2, SLC6A3, SLC9B2, and SLC30A4 where N-terminal tagging did not result in GFP positive cells, and for SLC16A6 (wild-type and mutants) where the N-terminal GFP fusion protein showed slightly stronger plasma membrane localization. In these cases, the RFP reporter was expressed upstream of an IRES sequence followed by the GFP-SLC fusion.

To generate point mutations in the SLC16A6 coding sequence at the phospho-degron motif DSGVELT (239–245), mutagenesis primers were designed using online tools (NEBaseChanger, NEB) and introduced into a pDONR plasmid using a Q5 site-directed mutagenesis kit (NEB). The resulting plasmids containing S240A, S240D, T245E, and T245N modified SLC16A6 were recombined in a Gateway LR reaction to express doxycycline-inducible SLC16A6-strepII-HA for validation experiments.

To generate cells expressing FLAG-KLHL36, HEK 293T cells were transduced with a lentiviral vector expressing KLHL36 under an EF-1α promoter. The HAP1 HA-dTAG-MTCH2 knock-in single-cell clone cell line was generated by microhomology-mediated end joining with the PITCh system as previously described (Bensimon et al, 2020).

All cell lines underwent routine testing to confirm the absence of mycoplasma contamination. Authentication of cell lines was performed, including RNA sequencing (RNA-seq) for comprehensive verification https://doi.org/10.5281/zenodo.5566804.

## Cell culture and mitochondrial fractionation

HEK 293 WT OE cells were grown on 15 cm cell culture dishes in high-glucose DMEM medium supplemented with 10% FBS and 1% Pen-Strep at 37 °C. SLC expression was induced by the addition of 1 µg/mL Doxycycline for 24 h. Depending on the expression levels 40, 80 or 160 million cells corresponding to two, four, or eight 15 cm plates per replicate were harvested.

For the standard AP-MS procedure, cells were scrapped in PBS, pelleted at $600 \times g$ at 4 °C for 10 min and frozen at −80 °C until further usage. For mitochondrial-localized SLC baits, a mitochondrial enrichment was performed. Scraped cells were transferred to a 50 mL Falcon tube to pellet cells. Next, pelleted cells were resuspended in freshly prepared isolation buffer (IBc, 10 mM Tris-MOPS, 1 mM EDTA-Tris, 200 mM sucrose, $H_2O$, pH 7.4, supplemented with 1× PI-cocktail) and incubated for 10 min on ice. Lysis was achieved by sonication with a Branson digital sonicator (cycle time of 30 s, 10% amplitude and 0.5 s on/off). The cell homogenate was centrifuged at $600 \times g$ at 4 °C for 10 min to remove un-lysed cells. The supernatant was collected and transferred to a 15 mL Falcon tube and centrifuged at

10,000×*g* at 4 °C for 15 min. The supernatant was discarded, and the mitochondrial fraction was washed with 500 μL IBc followed by another centrifugation step. The mitochondrial fraction was frozen at −80 °C until further usage.

## Affinity purification of SLC baits

We first affinity-purified the C-terminal fusion protein, and for SLCs which failed (e.g., expression levels), we tested the N-terminal construct. The QC procedure for each SLC bait involved several steps. First, we evaluated SLC bait expression using western blotting of full lysate extracts, cross-referenced with high-throughput imaging results (Goldmann et al, 2025). We assessed 686 cell lines (Table EV1) expressing 447 SLCs and two GFP cell lines for AP-MS. C-terminal tagged versions were used as default. N-terminal constructs were generated if low expression was observed. The second QC step was after the affinity enrichment. Roughly 2% of the AP-eluate was used to assess SLC enrichment by western blotting. Low-performing SLC baits were excluded from further sample processing. Post data acquisition, baits with low signal intensity (<24 SPC) and samples with high technical variability were excluded. In total, for AP-MS, 526 cell lines expressing 439 SLC baits were tested, with 119 rejected due to low expression or better performance of N-terminal tagged constructs.

We selected HEK 293 as the cellular model, as among 1206 cell lines reported in the human protein atlas HEK 293 cells express an average number of SLCs (see Appendix Fig. S17). For SLCs localized at the plasma, lysosomal, Golgi Apparatus (Golgi), Endoplasmic Reticulum (ER) or in vesicular membranes, cell pellets were lysed in freshly prepared lysis buffer containing 50 mM HEPES pH 8.0, 150 mM NaCl, 5 mM EDTA, 0.5% NP-40, 1 mM PMSF, 1× PI-cocktail, avidin (1 μg/mL), 1× phosphatase inhibitor cocktail 2 and 1× phosphatase inhibitor cocktail 3. The ratio of cells to lysis buffer was kept constant (2 plates = 1.8 mL lysis buffer). Lysates were clarified (14,000×*g* at 4 °C for 20 min) before they were incubated with 150 μL pre-equilibrated StrepTactin Sepharose (50% suspension) beads for 2 h at 4 °C. Beads were gently centrifuged at 1000×rpm for 1 min at 4 °C and the flowthrough was discarded. Beads were then washed twice with 1 mL lysis buffer and centrifuged. Next, beads were resuspended in 1 mL of washing buffer (50 mM HEPES pH 8.0, 150 mM NaCl, 5 mM EDTA) and transferred to BioSpin membrane columns (Bio-Rad). Beads were washed 3× with 1 mL washing buffer. Finally, proteins were eluted by incubation of beads with 125 μL 2% v/v SDS in washing buffer for 15 min.

For mitochondria-localized SLCs the enriched mitochondrial pellets were resuspended in 3 mL freshly prepared lysis buffer with the same composition as above except that instead of NP-40, 1% digitonin was used as detergent. To increase protein extraction yields, samples were briefly sonicated in a Branson sonicator with microtip (setting: 10 s sonication time, 0.5 s on/off cycles with 10% amplitude). After this step, mitochondrial samples were handled as outlined above for SLCs localized at other cell membranes.

Detailed descriptions of the AP-MS protocols were made publicly available under https://zenodo.org/records/7457416 and under https://zenodo.org/records/7462207 for the mitochondria-localized SLCs.

## Sample preparation for AP-MS

Sample preparation was performed by a single-pot solid-phase-enhanced sample preparation (SP3) procedure following an adapted protocol (Müller et al, 2020). In short, eluted proteins were reduced with DTT (10 mM) and incubated on a shaker at 56 °C at 600 rpm agitation for 1 h. Cysteine residues were alkylated with IAA (50 mM) followed by incubation in the dark at RT for 30 min. In all, 8.8 μL of Mag-beads prepared according to the manufacturer's instructions were added. Next, proteins were precipitated with 300 μL Acetonitrile (ACN) and the resulting solution was incubated for 10 min on a shaker followed by a 10 min incubation on the bench. Beads were magnetized and the buffer was removed, after which three washing steps with 200 μL 80% (v/v) ethanol with 2 min incubation in-between followed. Next, beads were washed twice with 180 μL ACN with 2 min incubation between washing steps. For digestion, beads were resuspended in 100 μL 50 mM Tris-HCl (pH 8) with 1 μg Trypsin/Lys-C Mix (37 °C at 600×rpm on). Samples were acidified with 2 μL of 30% TFA in H$_2$O and after magnetizing beads the supernatant was collected and centrifuged for 1 min at 14,000×*g* to remove remaining beads. Peptides were purified by solid-phase extraction C18 stage-tips protocol. After washing, peptides were eluted with 2 × 50 μL of 60% ACN, 0.1% TFA. Samples were dried under reduced vacuum at 45 °C and stored at −20 °C until analysis. Peptides were reconstituted in 30 μL MS-buffer of 0.1% TFA. A more detailed procedure for the MS-acquisition can be found under: https://zenodo.org/records/7462253.

## AP-MS data acquisition

Mass spectrometry data were acquired on a Q Exactive Orbitrap MS coupled to a Dionex Ultimate 3000 RSLCnano system interfaced with a Nanospray Flex Ion Source. Peptides were loaded onto a trap column (Pepmap 100 5 μm, 5 × 0.3 mm) at a flow rate of 10 μL/min 0.1% TFA. After loading, the trap column was switched in-line with the analytical column (50 cm, 75 μm inner diameter analytical column packed in-house with ReproSil-Pur 120 C18-AQ, 3 μm) kept at 40 °C to which an ESI Emitter Fused Silica was fitted. Data acquisition was conducted at a flow rate of 230 nL/min with a 120 min gradient (4–24% solvent B (90% ACN 0.4% FA) in 86 min, 24–36% solvent B within 8 min and 36–100% solvent B within 1 min, 100% solvent B for 6 min, 100–4% solvent B in 1 min and 4% solvent B for 18 min). Eluted peptides were ionized in a positive mode (1.8 kV). MS analysis was performed in a data-dependent acquisition (DDA) mode. Full MS scans were acquired in a precursor mass to charge (*m/z*) range of 375–1650 *m/z* in the orbitrap at a resolution of 70,000 (at 200 Da). Automatic gain control (AGC) was set to a target of $1 \times 10^6$ with a maximum injection time of 55 ms. Precursor ions were selected by a Top10 approach using a quadrupole isolation window width of 1.6 Da and higher energy collision-induced dissociation (HCD) at a normalized collision energy (NCE) of 28%. The MS2 AGC target was set to $1 \times 10^5$ with a maximum injection time of 110 ms and an orbitrap resolution of 17,500 (at 200 Da). Dynamic exclusion for selected ions was set to 40 s. A single lock mass at *m/z* 445.120024 and XCalibur version 4.1.31.9 and Tune 2.9.2926 were used.

Part of the AP-MS data were acquired on a Q Exactive HF-X Hybrid Quadrupole-Orbitrap MS. Peptides were separated by reverse-phase chromatography using a nano-flow HPLC (Ultimate 3000 RSLC nano system). Per injection, 3 μL sample was loaded onto a trap column (PepMap C18, 5 mm × 300 μm ID, 5 μm particles, 100 Å pore size) at a flow rate of 10 μL/min using 0.1% TFA. The trap column was switched in-line with the analytical column (PepMap C18, 500 mm × 75 μm ID, 2 μm, 100 Å) for analysis. During the next 10 min a flow rate of 50 μL/min with 0.1% FA 70% Methanol was applied. Elution was achieved at constant T of 40 °C (external butterfly heater controlled by column heater controller). For separation, solvent A with 0.4% FA and organic solvent B with 0.4% FA 90% ACN were used. Flow rate was set to 230 nL/min and a 120 min gradient was used (4–24% solvent B within 81 min, 24–36% solvent B within 8 min and 36–100% solvent B within 1 min, 100% solvent B for 6 min before equilibrating at 4% solvent B for 18 min). Eluted peptides were ionized in a positive mode (1.8 KV) using an Easyspray nanospray Source.

Data acquisition was performed in DDA mode. Full MS scans were acquired with a $m/z$ range of 375–1650 $m/z$ in the orbitrap at a resolution of 60,000 (at 200 Da). AGC was set to a target of $1 \times 10^6$ with a maximum injection time of 55 ms. Precursor ions for MS2 analysis were selected using a Top10 approach using a quadrupole isolation window of 1.6 Da and HCD at a NCE of 28%. AGC target was set to $1 \times 10^5$ with a maximum injection time of 55 ms and an orbitrap resolution of 30,000 (at 200 Da). Dynamic exclusion for selected ions was 15 s. A single lock mass at $m/z$ 445.120023 was employed for internal recalibration during the run. XCalibur version 4.3.73.11 and Tune 2.13 Build 3162 were used to operate the instrument.

## Processing of AP-MS

MS raw files were converted to mzML files with MSConvert (3.0.21128-7376ae988) (Chambers et al, 2012). Acquired spectra were searched using Philosopher's pipeline mode with MSFragger (da Veiga Leprevost et al, 2020). The identification search was performed against the canonical human proteome obtained from UniProtKB (downloaded on the 2021-04-01, status reviewed). The database was supplemented with 48 lab contaminants. The search parameters were set to include fully tryptic peptides, allowing for up to two missed cleavage-sites, with carbamidomethylation on cysteine residues as static modification. Further up to five variable modifications including oxidation on methionine and N-terminal acetylation were enabled. Mass tolerance was set to 50 ppm for precursor ions and 20 ppm for fragment ions. The peptide length was restricted from five to 63. A MS1 $m/z$ range from 375 to 5000 $m/z$ and precursor charges states +1 up to +4 were set. Peptide assignment validation was performed with PeptideProphet and protein inference with ProteinProphet. FDRs were controlled at 1% peptide FDR and 1% protein FDR, respectively. Files were searched one at a time similar to other large-scale interaction proteomics studies (Huttlin et al, 2021). Of note, no blinding was performed in this study. All analyses were conducted without blinding.

The assembled bait-prey matrix for each data modality served as input for feature generation (Fig. 1E). The 1432 MS-injections covering 358 SLCs and 28 GFP controls prepared with the standard protocol and measured on the QE were assembled. For the QE HF-X matrix, the data of 23 SLCs and 4 GFPs (107 MS-injections, 1

technical injection was missed) were combined. Lastly, the mitochondrial-localized SLCs bait-prey matrix was assembled, containing 208 MS-injections of 47 SLCs and mitochondria targeted GFPs. The data were independently assembled to account for the difference in the background. Next, the data per bait were grouped together, and reproducibility filtering strategy was employed, to remove sparse or wrongly assigned protein identifications, thereby lower protein FDR below 1%. The mean SPC per biological replicate was derived only for proteins which were quantified across technical injections. After this filtering step, the protein FDR was re-estimated by counting passing decoys, resulting in 0.64% FDR for the standard QE modality (6962 proteins, 45 decoys), 0.14% FDR for the mitochondrial modality (3709 proteins, 5 decoys), and a 0.82% FDR for the QE HF-X modality (5434 and 45 decoys).

## Assembly of feature matrix for scoring of PPIs

First, a set of scoring features was generated. Interaction scores for each protein were generated using CompPASS (Sowa et al, 2009) and SAINTexpress (Teo et al, 2014). As inputs, the bait-prey matrix per data modality was used. For CompPASS the normalization factor was set to 0.95. For the SAINTexpress scoring, we generated 100 in silico controls, specific for each SLC. To achieve this, we randomly sampled SPCs for each unique protein identifier from SLCs and GFP controls, only considering SLCs with similar summed up signal across all proteins. For this we ranked the absolute delta difference of the mean SPC to other SLCs and GFPs within each data modality. For the random sampling, only data from AP-MS of SLCs with similar summed up signal were considered, resulting in randomized controls. For the standard QE modality, 20% of all samples were used for random sampling, whereas for the mito and QE-HF-X datasets, which had fewer samples, up to 40% of all samples were used. For the sampling, the abundance of the SLC bait protein was excluded. These controls were used for SAINTexpress scoring with default parameters.

Second, a set of quantitative, annotation and experiment-wide features was derived (see Table EV2). For each protein pair of each SLC, additional quantitative features were derived, including the normalized spectral abundance factor (NSAF) (Neilson et al, 2013) and the ratio of proteotypic to total spectral counts. Further, dataset wide features per interaction partner were added, including the sum of total SPC and the ratio of total SPC against the data-wide summed up signal. In addition, SLC-protein pair feature was derived, including the FC against GFP (per data modality), and an additional FC against the SPC of each protein of all other SLC and GFP samples (leaving out the signal in the SLC experiment itself for which the FC was derived). As sample-specific quantitative features, the summed mean spectral counts, the number of quantified proteins, and the SLC bait abundance of each sample were added. Finally, as deterministic features, the experimental protocol ("standard" or mitochondrial AP-MS) and the MS-platforms were added. Finally, all data modalities were assembled (Dataset EV9).

## Curation of labels for SLC-protein interactions

For the ML-based probability scoring of interactions, two sets of labeled PPIs were generated. The first set contained overlapping

PPIs with BioGRID (v.4.4.223) (Oughtred et al, 2021), filtered to contain only the experimental category identifiers "Affinity Capture-MS", "Affinity Capture-Western", "Reconstituted Complex" and "Co-crystal Structure". All matched pairs were set as true. For the second set, we used SLC-protein pairs reported in PDB. This list was expanded by including bait self-loops and curated interactions. The curated PPIs were selected considering PPIs recovered within the SLC interactome and BioGRID, taking the abundance and specificity of each SLC-protein pair into account. The final feature matrix contained 693 true and 970 false labels.

## ML-based scoring of PPIs

The aim of the ML learning was to integrate the features (see above) in one model to calculate an interaction probability for each SLC-protein combination. First, the features were pre-processed. To map all input features to real numbers, each feature was either log- or z-transformed, or taken as-is. Non-finite values were imputed to complete the feature matrix. To assure numerical stability during model fitting, the transformed features were subsequently normalized to have zero mean and unit standard deviation. To provide reliable probabilities whether a protein is an interactor, an ensemble of 30 Radial Basis Function (RBF)-based classifiers is fitted iteratively on different random subsets of all speculatively labeled interaction partners. Using speculative labels allows for potential mistakes in database entries and is considered in model fitting by allowing for BioGRID-based reports of interaction between interactors and SLC as well as lack of reported interaction to be wrong. We assume however that a small set of exquisitely curated interaction states can be trusted without doubt. While always trusting the curated labels, model fitting redraws novel BioGRID-derived interaction labels according to the RBF-based interaction probabilities. This process is repeated until the respective labels show less than 1% differences in two consecutive runs. Against the curated PPI dataset averaged ensemble predictions reached an accuracy of 95%, with a sensitivity of 0.89 and specificity of 1.0. We used a combination of thresholds (RBF probability ≥0.4, log2FC against GFP > 2, FC against other samples >1.5, entropy filter <0.75, quantification frequency <99%) to classify proteins as interactors or background. From the initial 634,125 protein pairs roughly 2.99% or 18,991 interactions were assigned as PPIs.

## Analysis of SLC16A6 phospho-mutant AP-MS dataset

AP-MS data of SLC16A6 phosphomutants were generated as described. Data were searched together with the original SLC16A6$^{WT}$ AP-MS raw files and 24 GFP controls using Philosopher. Peptides were quantified with freequant (default parameters). For further analysis, the peptide-level output was used. The SLC16A6 phosphomutants and SLC16A6$^{WT}$ samples were filtered and the Top3 most intense peptides across the experiment were used to infer protein abundances. Protein abundances were normalized using total signal scaled to the median signal across the experiment. Next, abundances were log2 transformed, and missing values were imputed sampling from a normal distribution around the lowest 5% quantile of all abundances. For each phospho-mutant, the log2FC against the

SLC16A6$^{WT}$ was derived. A log2FC of ±1 was used to assign enriched or depleted interactions, respectively. The results are reported in Dataset EV8.

## Curation of a PPI-reference library of SLCs

To benchmark our SLC interactome, we assembled a reference PPI library covering multiple PPI-databases. For this, we mined BioGRID (Oughtred et al, 2021), IID (downloaded on the 11.05.2021) (Kotlyar et al, 2022), IntAct (Orchard et al, 2014) (downloaded on the 11.09.2023) and STRING v12 (Szklarczyk et al, 2023). BioGRID was filtered to include PPIs from the categories "Affinity Capture-MS", "Affinity Capture-Western", "Reconstituted Complex" and "Co-crystal Structure". IID was limited to experimental data and human proteins. PPIs without an associated reference were removed from IID and experimental methods were limited to AP-MS. STRING was filtered to cover physical interactions with a confidence threshold of 0.4. IntAct was filtered to contain only human proteins in combination with the method identifiers: "MI:0006", "MI:0007", "MI:0019", "MI:0096", "MI:0114", and "MI:0676". For the reference count, interactions obtained from STRING were not considered. For categorization of proteins containing TM-domains, the UniProtKB database annotations for topological domain "transmembrane" (downloaded on the 2022-05-13) were used.

## Benchmarking of SLC interactome

We benchmarked our dataset on PPI and protein complex levels. On PPI level, we performed the benchmark against BioPlex (Huttlin et al, 2015, 2017, 2021) and HuRI (Luck et al, 2020), whereas on complex level we used the CORUM complex (Tsitsiridis et al, 2023) database, filtered for human complexes and heteromers. CORUM complexes were translated to a PPI-network, assuming full connectivity of all subunits. Next, we supplemented interactions between all interactors for each SLC and generated a supplemented PPI-network. This network was used to calculate the overlap with CORUM interaction pairs. To estimate if CORUM interaction pairs were enriched within the original SLC co-purification PPI-network, we permutated network nodes. For permutation, we sampled from all potential interactors and SLCs present in our dataset but conserved the network topology. This permutation was repeated 10,000 times, and the *P* value was derived by a rank-based test. To estimate the *P* value, the number of perturbed networks with a higher number of recovered CORUM PPI pairs compared to the SLC interactome was divided by the number of permutations. Next, we used the CORUM data to derive counts of complexes and subunits present in the SLC interactome. For counting complex subunits, we considered subunits present in multiple assemblies. CRAPome 2.0 database (Mellacheruvu et al, 2013) was used to retrieve proteins regularly found in AP-MS negative controls.

## Construction of the SLCome

To construct a structure-based phylogenetic tree of the human SLCs, we retrieved the similarity matrix of protein structure models from (Ferrada and Superti-Furga, 2022), restricted to the 446 human SLCs which the RESOLUTE dataset studied. SLC22A20P

was not included in the structural analysis. Distances were obtained by subtracting similarities from 1 and used to cluster SLC structures hierarchically (hclust function in R, version 4.3.3, with Ward's method). An unrooted tree was constructed using square root scaled branch lengths (ggraph library, version 2.2.1). Node positions were adjusted manually to reduce overlaps and optimize overall arrangement. In the final unrooted phylogenetic tree, which we termed SLCome, edges were color-coded by structural folds (Ferrada and Superti-Furga, 2022), and SLC nodes were colored if they were part of the SLC interactome. The number of novel and reported interactions were reported for each structural fold, using the reference PPI library assembled for this study.

## Prediction of structures of protein interactions by AlphaFold

We used AlphaFold multimer (version 2.3 and 3.0.0) (Evans et al, 2021) to model binary PPIs. For each prediction, at least 25 models were generated. Amber was set to TRUE. Predicted structures were assessed with the pDockQ score similar to another study (Burke et al, 2023). A pDockQ score >0.5 was considered as high-confidence model, whereas a pDockQ <0.5 but >0.23 were classified as medium-confidence structure. A pDockQ score <0.23 was considered as low confidence structure. Negative SLC-chaperone interaction pairs for structural modeling were randomly sampled considering the lowest 25% quantile of the quantitative signal across our data. Statistics were obtained using an unpaired two-sided $t$ test. Structures were visualized with UCSF ChimeraX 1.7.1 (Pettersen et al, 2021), by selecting the structure with the highest pDockQ.

## Clustering analysis of the SLC interactome

To cluster SLC interactomes, we performed hierarchical clustering on the Jaccard distance of all interactors (hclust function in R, with Ward's method). The selected number of clusters ($k = 38$) was identified based on the mean silhouette width (Appendix Fig. S7). GO enrichment analysis against GO Biological Process 2023 (GO:BP:2023) was performed with Enrichr (Kuleshov et al, 2016) for individual clusters. Of the 1986 unique terms, only 25% were found in more than three clusters. This resulted in 681 terms used for hierarchical clustering (Euclidean distance, hclust function in R, with Ward's method, $k = 44$, Appendix Fig. S8A). For the identification of representative GO terms covering major biological functions, we used GO semantic similarity analysis with the R-package GoSemSim (Yu, 2020). Similarities were calculated per cluster using the relevance method. Subsequently, we used the reduce function to identify parental terms per cluster (score=size, cutoff of 0.7). The selected terms were visualized as functional cluster annotations in Fig. 3A. Similarities between enriched GO terms were derived by using the Wang method with a similarity cutoff of 0.65 (Fig. 3B).

To classify protein functions associated with the SLC interactome in Fig. EV4A, we used the gene set enrichment tool Enrichr and the gene set GO:BP:2023. The top ten terms were ranked by adjusted $P$ value and consolidated into three major categories. $P$ values were aggregated using P Fisher's combined probability test. To classify subcellular protein localizations associated with the SLC interactome in Fig. EV5, we used Enrichr and the gene set GO

Cellular Component 2023 (GO:CC) with individual SLC interactomes. To consolidate terms into ten subcellular localizations (Fig. EV5A), we generated curated lists of GO:CC terms for each subcellular localization and sequentially and conditionally evaluated each term or its parental terms for association with these subcellular localizations. $P$ values were consolidated as outlined above for GO:BP with a $P$ value cutoff <0.05.

## SLC property enrichment analysis

As a measure of functionality, we tested clusters for enrichment of SLC functional properties of different classes (coupled ion, family, fold, location, substrate class). Respective classes and annotations are described in (Goldmann et al, 2025). Fisher's exact test for overrepresentation was performed for each functional property in each cluster, only considering properties with at least three annotations. Resulting $P$ values were corrected for multiple testing using the Benjamini–Hochberg procedure for each class separately, and enrichments of functional SLC properties in clusters were called at 20% FDR.

## Co-purification analysis of the SLC interactome

Clustering of prey-preys was performed on the ratios of the prey to bait NSAF values. This normalizes for the bait and prey abundances considering the length and with the ratios against the prey also for the bait abundance within each AP-MS. For the clustering, preys which were only identified in one single AP-MS experiment were removed, to look for protein complexes or functional modules which were co-purified in two or more AP-MS experiments. A second filtering step removed all SLCs which had only one prey left after the initial filtering. The input matrix comprised 2835 interactors from 371 SLC baits. The Pearson correlation coefficient was derived employing complete linkage. The correlation was converted to a distance matrix by $dist = \sqrt[2]{1 - |r|}$. For unsupervised hierarchical clustering, the Ward's clustering criterion (Ward, 1963) (Ward d2) was used, and the best number of clusters (k) was found by using mean silhouette width to identify a local maximum at $k = 207$ (Appendix Fig. S9A). Clusters were analyzed by performing GO term biological processes using the tool Enrichr (Xie et al, 2021). Next, we overlapped the protein identifiers per cluster with the CORUM complex database (Tsitsiridis et al, 2023), applying the same pre-processing as for the benchmarking experiment. Matched CORUM complexes were filtered by their subunit completeness.

## Protein stability and subcellular localization assay

For all protein stability and subcellular localization assays, cells were passaged onto poly-L-lysine-coated and tissue culture-treated 96-well plates at 10,000 and 25,000 cells per well in DMEM supplemented with 10% FBS, 1% penicillin/streptomycin and 1 µg/mL doxycycline 24 h before analysis. For flow cytometry, cells were PBS washed, trypsinized, and resuspended in FACS buffer (5% FBS, 1 mM EDTA in PBS) before analysis on a BD LSRFortessa cytometer. Signals were acquired for GFP, RFP, and BFP as outlined in the Reagent and Tool Table. Cytometry analysis was performed using FlowJo by gating for healthy and single cells (usual range 5,000–10,000 per sample) and additionally for BFP-positive cells in the case of cDNA overexpression.

Median fluorescent values were processed using Python software. Significant changes had to fulfill three criteria: (1) The mean of the median GFP:RFP ratio and the mean of the median GFP values had to change by at least 10% compared to the respective mean of the median parameter following control RNAi; (2) Statistical significance was defined as a *P* value less than 0.01 in an unpaired independent *t* test comparing GFP:RFP ratios between control and interactor RNAi (see Dataset EV7 for exact *P* values); (3) The change in GFP values, relative to the mean of the median GFP values after control RNAi, had to occur in the opposite direction to the change in RFP values, relative to the mean of the median RFP values; or the mean change in RFP values had to be no more than half of mean GFP value change.

For high-throughput imaging, media were exchanged with fresh DMEM containing 10% FBS, 1% penicillin/streptomycin and 1 μg/mL Hoechst 33342 and imaged on an Opera Phenix with 40x water objective and signal acquisition for Hoechst, GFP and RFP as outlined in the Reagent and Tool Table. Image analysis was performed using Fiji (Schindelin et al, 2012) with custom macros to measure signal Pearson correlation and thresholded signal areas, intensities and overlaps (BioImage Archive submission S-BIAD1105) and processed using Python software. Significant changes had to fulfill three criteria: (1) For SLCs residing at a single subcellular location and for the cell body reference RFP, as well as for GFP signals at the reference RFP, the mean GFP intensity had to change by more than 10% compared to the mean GFP intensity after control RNAi, and more than 25% for GFP signals not overlapping with the reference RFP; (2) Statistical significance was defined as a *P* value less than 0.01 in an independent *t* test comparing GFP intensity between control and interactor RNAi (see Dataset EV7 for exact *P* values); (3) For SLCs residing at a single subcellular location and for the cell body reference RFP, as well as for GFP signals at the reference RFP, the mean GFP intensity change had to occur in the opposite direction to the mean RFP intensity change at the reference RFP. For interactions that significantly changed multi-location SLCs at a specific subcellular location, two additional criteria were applied: (1) Statistical significance was defined as a Benjamini/Hochberg corrected *P* value less than 0.001 in an independent *t* test comparing relative GFP intensity changes between GFP positive pixels that overlapped with, and those that did not overlap with, the reference RFP; (2) the mean GFP intensity change at reference RFP pixels had to occur in the opposite direction to the mean GFP intensity change at non-reference RFP pixels, or the difference between the GFP intensity changes had to be at least one-third of the absolute value of the larger change.

For drug treatment, cells were plated on treated 96-well plates at 10,000 cells per well 2 days before analysis and induced with doxycycline-containing media 24 h before analysis. An additional 10 μL drug-containing media (0.5 μM TAK-243; 0.5 μM MLN4924; 100 nM Bafilomycin A1; or 0.05% DMSO in full media) were supplemented 4.5–6 h before analysis.

For RNAi-mediated depletion, cells were first seeded on tissue culture-treated six-well plates at 200,000 cells per well a day prior, or at 400,000 cells per well on the day of transfection in antibiotics-free media. A transfection mix of 400 μL OptiMEM with 1 μL DsiRNA (Integrated DNA Technologies; stock concentration 20 μM prediluted in duplex buffer from three individual DsiRNAs per gene, see Table EV5 for a list of DsiRNAs) and 4 μL Lipofectamine RNAiMAX Transfection Reagent (Thermo Fisher Scientific) was incubated for

10 min before dropwise addition to cells and incubation for 48 h. Cells were trypsinized, counted and seeded with doxycycline-containing media on 96-well plates with 8 wells per sample. To assess protein stability by cytometry, four wells were treated with DMSO or inhibitors to ubiquitination, neddylation or lysosome acidification 4.5–6 h before analysis. See additional details regarding DsiRNA preparation, validation by RT-qPCR and DIA-based profiling of RNAi-treated cells in "Methods" and Appendix Figs. S18 and S19; Table EV6; Datasets EV10 and EV11.

Interactor cDNA was obtained from Addgene, from Novartis or ordered from Genscript (see Table EV7 for details including references and sequences). Coding sequences were cloned into Gateway™ pDONR™221 (Thermo Fisher Scientific) and recombined into a constitutive expression vector containing an EF-1α promoter, the interactor cDNA, an IRES, and a TagBFP-NLS sequence to control for expression. For cDNA overexpression, HEK 293 cells were first seeded on tissue culture-treated 12-well plates at 100,000 cells per well a day prior, or at 200,000 cells per well on the day of the transfection. Untagged coding sequences were expressed from a constitutive EF1alpha promoter followed by a BFP-NLS reporter separated by an IRES. Transfection was performed by mixing 50 μL OptiMEM with 3 μL Lipofectamine 3000 (Thermo Fisher Scientific), 5 min of incubation, and mixing the Lipofectamine solution with 50 μL OptiMEM supplemented with 1 μg plasmid DNA and 2 μL Lipofectamine P3000 reagent before dropwise addition to cells. All assay results are summarized in Table EV8.

## Cadmium uptake assay

To assess changes in SLC39A8 transporter activity, a combination of a cadmium uptake assay with RNAi of interactors was used. Experiments were conducted in HEK 293 WT OE cell lines, applying the same conditions as described for the protein level and localization assays. After 48 h RNAi treatment, cells were seeded on 384-well plates and expression of SLC39A8 was induced for 24 h. Next, cells were washed six times with 20 μL uptake buffer using AquaMax DW4 (AquaMax DW4 Microplate Washer), leaving 20 μL residual buffer. 20 μL of component A (FLIPR® Calcium 5 Assay Kit) were added, and plates were centrifuged for 1 min at 1000× g followed by 2 h incubation at RT in the dark. The assay was performed on an FDSS 7000EX with an excitation wavelength of 480 nm and an emission of 540 nm with an exposure time of 200 ms for 20.5 min. After 50 sec, 20 μL uptake buffer (117 mM NaCl, 4.8 mM KCl, 1 mM $MgCl_2$, 10 mM glucose, 10 mM HEPES pH 7.4) was added. Next, cells were incubated with 5 μM $CdCl_2$ in uptake buffer at second 351 for 15 min and images were acquired every 2 s. Edge effects were mitigated by omitting border columns and rows. The mean fluorescence signal across replicates (Biological replicate 1 had 12 wells whereas biological replicate 2 and 3 each 8 wells) were derived and min-max scaled. The mean AUC was calculated by the trapezoid rule. Significance was tested across mean AUC using an unpaired *t* test. Cell viability was measured by CellTiter-Glo (Promega). Results are summarized in Appendix Fig. S15 and Dataset EV12.

## TRACT assay

HEK 293 Jump In T-REx cells expressing SLC1A3, SLC1A2, or SLC22A3 were assessed by Impedance-based Transport Activity

through receptor Activation (TRACT) assays using MP real-time cell analyser (RTCA) as described previously (Sijben et al, 2022). HEK 293 Jump In T-REx cells were seeded at 100,000 cells/well. The next day, cells were transfected with DsiRNA mix (10 nM) targeting the interactor of interest in OptiMEM using Lipofectamine RNAiMax and grown for an additional 48 h.

In brief, siRNA transfected cells overexpressing SLC1A3 or SLC1A2 were seeded on a 96-well E-plate PET in absence and presence of 1 μg/mL doxycycline to induce expression of SLC1A3 or SLC1A2, respectively. Cell growth was monitored for 23 h after which cells were stimulated with increasing concentrations of L-glutamate and cellular response was measured for 2 h.

SLC22A3-mediated uptake of MPP$^+$ was measured using xCELLigence MP RTCA as described previously (Mocking et al, 2022). In short, HEK 293 Jump In T-REx SLC22A3 cells were seeded (60,000 cells/well) on a 96-well E-plate PET containing 1 μg/mL doxycycline to induce SLC22A3 expression and cell growth was monitored for 23 h. Cells were stimulated with increasing concentrations of MPP$^+$ and cellular response was detected for a total of 1 h by measuring impedance every 15 s for the first 25 min, followed by every minute for 10 min and then every 5 min for 1 h total.

Data analysis was performed with Prism 9.0. Assay data were presented as mean ± SEM of at least three experiments performed in duplicates. For impedance-based XCELLigence assays, RTCA software pro v2.8.0 (Agilent, CA, USA) was used to record and normalize the data to the time-point prior to stimulation with substrate to obtained delta cell index (dice). Vehicle-induced responses were subtracted to correct for any ligand independent effects. Vehicle-corrected data was analyzed by calculating the area under the curve (AUC) over the first 60 min or 120 min for SLC22A3and EAATs, respectively. Concentration-response curves were analyzed by fitting a non-linear regression three or four parameter response model for SLC22A3or EAATs, respectively, to obtain pEC$_{50}$ values. SLC22A3data was normalized by setting 1 mM MPP$^+$ to 100%. EAAT data was normalized by setting the top fitting of L-glutamate response on siRNA control cells at 100%. Significant differences between potency (pEC$_{50}$) and E$_{max}$ of control and interactor RNAi were determined using unpaired $t$ test (Appendix Fig. S14).

## Full proteome profiling of HEK 293 Jump In T-REx cell line and six other human cell lines

Based on a publicly available RNA-Seq dataset of 675 cell lines (Klijn et al, 2015) a set of 6 adherent human cell lines (HCT 116, Huh-7, LS180, MDA-MB-468, SK-MEL-28, 1321N1) cumulatively covering expression (TPM > 1) of about 80% of all human SLCs were selected as model cell lines in the RESOLUTE project. In addition, the HEK 293 Jump In T-REx cells, which were used in this study to generate SLC-overexpression cell lines, were profiled employing an offline high-pH fractionation prior to MS-data acquisition.

The HEK 293, Huh-7, SK-MEL-28, and 1321N1 cell lines were grown in DMEM - high glucose (Sigma) supplemented with 10% FCS and 5% Pen-Strep, whereas LS180 and MDA-MB-468 were grown in RPMI-1640 Medium (Sigma) supplemented with 10% FCS (Biowest) and 5% Pen-Strep (Gibco). HCT 116 were grown in McCoy's 5 A Medium supplemented with 10% FCS (Biowest) and 5% Pen-Strep (Gibco). Cell lines were grown in 10 cm cell culture

dishes until confluent. Media was removed, and cells were washed once with ice-cooled PBS. Cells were harvested by scraping in PBS supplemented with cOmplete Protease Inhibitor (1x; Roche). Cells were pelleted at 400×$g$ for 10 min at 4 °C and snap-frozen in liquid nitrogen. Cell pellets were lysed by the addition of 200 μL lysis buffer (50 mM HEPES, with 2% SDS supplemented with 1 mM PMSF and protease inhibitor cocktail). Following an incubation for 20 min on ice, lysates were sonicated (Covaris S2) for 150 s and cleared on a table-top centrifuge at 14,000×$g$ for 15 min at 4 °C. For each cell line, 250 μg of total protein (measured by BCA) were digested employing an adapted FASP protocol (Wiśniewski et al, 2009). Reduction of proteins was achieved by the addition of DTT (final concentration of 83.3 mM) and incubation at 99 °C for 5 min. Following the reduction, samples were mixed with 200 μL freshly prepared UA buffer (8 M urea in 100 mM Tris-HCl at pH 8.5) and added onto the FASP filter columns (Merck Millipore). All the following buffer exchange steps commenced by centrifugation at 14,000×$g$ for 15–20 min at 20 °C. First the sample lysis buffer was removed, and samples were washed twice with 200 μL UA buffer. Proteins were alkylated by the addition of IAA (final concentration of 50 mM) and incubation in the dark for 30 min at RT. Following the alkylation, samples were washed three times with 200 μL UA buffer, followed by three more washes with 100 μL 50 mM TEAB buffer (Sigma-Aldrich). Samples were then digested overnight with the addition of trypsin at 37 °C with mild agitation. In the morning, 50 μL of 50 mM TEAB buffer were added to each sample and peptides were collected. Filter units were washed once with 50 μL 0.5 M NaCl and flowthroughs were pooled. After digestion, peptides were cleaned up employing C18 Solid Phase Extraction (SPE, Nerst Group). Peptides were dried under reduced pressure at 46 °C. For high-pH fractionation, peptides were resuspended in 20 mM ammonium formate pH 10. Peptides were separated into 96 time-slice fractions using a C18 column connected to an HPLC. Solvent A consisted of 5% acetonitrile in 20 mM ammonium formate pH 10, and solvent B consisted of 90% acetonitrile in 20 mM ammonium formate, pH 10. The HPLC was operated at constant flow rate of 100 μL/min and peptides were eluted using a separation gradient from 0 to 100% solvent B. The 96 fractions per cell line were reconstituted into 36 fractions in a concatenated fashion. Peptides were dried again before they were reconstituted in 10 μl 5% formic acid.

The MS data was acquired on a Q Exactive hybrid quadrupole-Orbitrap mass spectrometer (ThermoFisher Scientific) coupled to a Dionex Ultimate 3000RSLC nano system (ThermoFisher Scientific) via nanoflex source interface. Tryptic peptides were loaded onto a trap column (Pepmap 100 Å, 5 μm, 5× 0.3 mm, ThermoFisher Scientific) at a constant flow rate of 10 μL/min using 2% ACN and 0.1% TFA as loading buffer. After loading of the peptides, the trap column was switched in-line with the analytical column (20 cm, 75 μm inner diameter) packed in-house (ReproSil-Pur 120 C18-AQ, 3 μm). As solvent A, 0.4% FA in water and as solvent B, 0.4% FA in 90% ACN and 10% water were used. The HPLC was operated at a constant flow rate of 230 nL/min and peptide separation was achieved employing a 90 min gradient (6% to 30% solvent B within 81 min, 30% to 65% solvent B within 8 min, and 65% to 100% solvent B within 1 min, 100% solvent B for 6 min) before the column was re-equilibrated at 6% solvent B for 18 min. The MS was operated in positive mode employing data-dependent acquisition (DDA). The precursor mass range for full MS scans was set to a

mass range of 375–1650 $m/z$ in the orbitrap at a resolution of 70,000 (at 200 Th). Automatic gain control (AGC) was set to a target of $1 \times 10^6$ and a maximum injection time of 55 ms was allowed. Precursor ions for MS2 analysis were selected using a Top15 dependent scan approach using a quadrupole isolation window of 1.6 Da and higher energy collision-induced dissociation (HCD) at a normalized collision energy (NCE) of 38%. For MS scans, the AGC target was set to $1 \times 10^5$ with a max. injection time of 50 ms and an Orbitrap resolution of 17,500 (at 200 Th). Dynamic exclusion for selected ions was set to 60 s. A single lock mass at m/z 445.120024 was used.

Raw files of each cell line were processed together using the software tool Proteome Discoverer (v.2.3). For the analysis, a LFQ workflow was set. If not explicitly stated, the default parameters were enabled. The Spectrum files RC was set to 50 ppm precursor and 0.02 Da fragment ion mass tolerance, respectively. The Spectrum Selector module was used with a minimum peak count threshold of 10. The search was performed employing Sequest HT against the canonical human reference proteome obtained from UniProtKB (February 2019, status reviewed) supplemented with a list of common lab contaminants (cRAP version 2012). Spectra were evaluated with Percolator using a PSM and peptide FDR of 0.01, respectively. The Minora Feature Detector was enabled. The consensus workflow was based on the default settings and was employed to integrate the results of all raw files per individual cell line. The PSM grouper was enabled, and only high-scoring peptides were further filtered with the Peptide Validator employing a strict FDR of 0.001. Only peptides and proteins with at least medium confidence were used for further scoring in the protein scorer (used with default parameters). Furthermore, a Protein FDR of 0.01 was set. For quantification, only unique peptides were used. The MS raw data are available via the ProteomeXchange Consortium via the PRIDE partner repository with the dataset identifier PXD053130 (Data ref: Goldmann and Superti-Furga, 2024).

### Transcriptome profiling of HEK 293 WT cell lines and additional RESOLUTE model cell lines

In addition to the full proteome profiling, we performed transcriptomics analysis by RNA-seq for the same set of 6 adherent human cell lines and HEK 293 Jump In T-REx ($n = 3$ biologically independent replicates). The cell lines were grown under the same conditions as outlined for the full proteome profiling. For RNA isolation, $1 \times 10^6$ cells were washed with PBS and lysed with RTL buffer (Qiagen) supplemented with 0.143 M β-mercaptoethanol and RNA was isolated with the RNeasy Mini Kit (Qiagen). The RNA isolation included a DNAse I digestion according to the manufacturer's instructions. For analysis, all samples had to reach a high RNA integrity (mean RIN 9.6, ± 0.21 SD) assessed using a Bioanalyzer instrument (Agilent). For mRNA-Seq library preparation the Ultra II Directional RNA library Prep Kit from Illumina (New England Biolabs) in combination with rRNA depletion kit (New England Biolabs) were used.

To quantify the transcriptome of these reference cell lines, we performed RNA-seq on Illumina HiSeq 4000 with 80 bp single-read setup. Illumina adapters were trimmed and clipped using Cutadapt (1.18), and reads were mapped to h38 genome with STAR (2.6.1a). Quantification of transcript abundance was performed with the pseudoalignment approach to transcriptome (ENSEMBL GRCh38

94 cDNA) using Kallisto (0.44.0). Fastq files were deposited in the public domain (Data ref: Sedlyarov and Superti-Furga, 2019).

### Validation of RNAi

The 27mer duplex DsiRNAs were designed by and obtained from Integrated DNA Technologies in 96-well deep-well plates in lyophilized form as the top three candidate DsiRNAs (for some targets the top six candidate DsiRNAs were obtained). After dissolution to 100 µM in molecular biology grade water, stocks were stored at −20 °C. Working DsiRNA stocks were generated by combining 48 µL IDT duplex buffer (Integrated DNA Technologies) with 4 µL DsiRNA stock solution of DsiRNAs 1, 2, and 3 of the target gene to obtain 60 µL of a 20,000× working stock and stored at −80 °C. Where applicable, stocks 4, 5, and 6 were combined for a second pool.

To confirm the knockdown efficiency of our approach, we validated a subset of 9 of the 105 targeted genes using RT-qPCR. For the target genes LTN1, NFXL1, and ZFPL1 we used HEK 293 Jump In T-REx cells and processed them alongside samples for DIA-based full proteome profiling. For the target genes GBA2, ILVBL, and SIGMAR1, we used the SLC6A8-GFP cell line (Resolute ID CE07SA-K), for target gene GOLGA5 we used the SLC6A15-GFP cell line (Resolute ID CE07VF-2) and for the target genes KCTD5 and IQSEC1 we used the SLC1A3-GFP cell line (Resolute ID CE07TV-9) and processed them alongside protein stability and subcellular location experiments summarized in Dataset EV7. Briefly, cells were seeded in DMEM media supplemented with 10% FBS without antibiotics in six-well TC plates 80–96 h before harvesting and transfected 72 h before harvesting with DsiRNAs at final concentrations of 10 nM with 0.2% Lipofectamine RNAiMax (Thermo Fisher Scientific). Where applicable, SLC-GFP expression was induced using doxycycline (1 µg/mL) 24 h before harvesting. Finally, cells were washed with PBS, resuspended with 350 µL buffer RLT (Qiagen) by pipetting, transferred to a tube, and frozen at −20 °C before further processing.

RNA was extracted according to manufacturer's recommendation. Briefly, samples were thawed, vortexed and mixed with 350 µL 70% ethanol. The homogenate was applied to RNeasy columns, washed with buffer RW1 and buffer RPE and eluted using 30 µL molecular biology grade water. RNA concentration was then measured using NanoDrop. Genomic DNA contamination was digested using DNase (Thermo Fisher Scientific) in the presence of RiboLock RNase inhibitor (Thermo Fisher Scientific). Reactions were inactivated using EDTA and incubation at 65 °C.

The mRNA was reverse transcribed using RevertAid (Thermo Fisher Scientific) and oligo dT primers according to the manufacturer's recommendations. The qPCR was performed using Luna Universal qPCR Master Mix (NEB) according to the manufacturer's specifications with gene-specific primers (Table EV6) and control HPRT1 housekeeping control primers in triplicates. Knockdown efficiency was calculated using the ΔΔCq method (Livak and Schmittgen, 2001).

### DIA-based full proteome profiling of RNAi-treated cells

To validate RNAi downregulation of gene expression on protein level, we assessed protein abundance of targeted genes 72 h after DsiRNA

transfection. For the assessment, 200,000 HEK 293 Jump In T-REx cells were seeded in 10 cm TC-treated plates in triplicates for each target gene pool and non-targeting control and grown in antibiotics-free DMEM (Sigma), supplemented with 10% FBS (Biowest). The next day, cells were transfected with DsiRNA pools or control DsiRNA using OptiMEM (final concentration of 20% v/v) with DsiRNA (final concentration 10 nM) and Lipofectamine RNAiMAx (Thermo Fisher Scientific; final concentration 0.2% v/v) according to the Lipofectamine protocol. Cells were incubated for 3 days at 37 °C before harvesting by washing with PBS and collecting cells by scraping them of the plate. Cells were pelleted by centrifugation at 1000 rpm at 4 °C for 10 min and frozen in liquid nitrogen.

## MS sample preparation

Cells were lysed in 200 µL lysis buffer (50 mM HEPES, with 2% SDS, 1 mM PMSF, and protease inhibitor cocktail). First, cells were incubated for 20 min at RT, before samples were heated up to 95 °C for 5 min. To ensure complete cell lysis, samples were sonicated (Covaris S2) for 150 s. Lysates were clarified with centrifugation at $16,000 \times g$ for 10 min at RT. For each sample, 100 µg of total protein (BCA) were digested employing a filter-aided sample preparation (FASP) protocol adapted from the published procedures (Wiśniewski et al, 2009) and described in more detail above in the appendix method section for the full proteome profiling of model cell lines. Proteins were enzymatically digested to peptides by the addition of 1.5 µg sequencing-grade modified trypsin at 37 °C overnight. Peptides were de-salted by C18 spin-columns (Thermo Scientific). Cleaned-up peptides were vacuum dried under reduced pressure at 46 °C and stored at −80 °C till further processing. Peptides were resuspended in 0.1% TFA in HPLC-grade $H_2O$. MS data were acquired on an Orbitrap Fusion Lumos Tribrid mass spectrometer (ThermoFisher Scientific) coupled to a Dionex Ultimate 3000 RSLCnano system (ThermoFisher Scientific) interfaced with a Nanospray Flex Ion Source (ThermoFisher Scientific). First, peptides were loaded on a trap column (PepMap 100 C18, 5 µm, 5 × 0.3 mm, ThermoFisher Scientific) at a 10 µL/min flow rate of 0.1% TFA in water. After loading of the peptides, the trap column was switched in-line, and peptides were separated on an in-house packed analytical column (50 cm, 75 mm inner diameter, packed with ReproSil-Pur 120 C18-AQ, 3 µm) fitted to an ESI emitter fused silica (20 µm ID × 7 cm L × 365 µm OD; Orifice ID: 10 µm, CoAnn Technologies). The column was kept in a column oven at 50 °C. 0.4% formic acid (FA) in HPLC-grade $H_2O$ and 0.4% FA in acetonitrile (ACN) was used as running buffer A and B, respectively. Peptide separation was achieved by a four-step gradient over 120 min (4% solvent B from 0 to 4 min, followed by 4% solvent B to 24% from 4 to 86 min, 24% to 36% solvent B from 86 to 94 min, with a step increase to 100% solvent B within 1 min and a hold at 100% solvent B till 101 min followed by a decrease to 4% solvent B within 1 min and an additional 18 min re-equilibration at 4% solvent B). The mass spectrometer was operated in positive mode employing a data-independent acquisition (DIA) method. The MS1 precursor survey scan was performed in a mass to charge range from 350 to 1650 $m/z$ in the Orbitrap at 120,000 resolution. As default charge state +1 was set. The normalized AGC was set to 200% and the max. injection time was limited to 100 ms. For the DIA windows, precursors between 379 to 880 $m/z$ were split into 100 windows with an isolation window of 5 $m/z$ and

a +/− 2 $m/z$ window overlap. The precursors were isolated on the quadrupole. The RF lens was set to 30%. Peptide precursors were fragmented by high-energy collision-induced fragmentation (HCD) using a stepped collision energy of 24, 28, and 30%. The normalized AGC target was set to 200% with a dynamic maximum injection time with a desired number of 10 points across the peak. The fragment spectra were acquired in the Orbitrap at 30,000 resolution. Xcalibur Version 4.3.73.11 and Tune 3.4.3072.18 were used to operate the instrument.

The DIA data was processed in Spectronaut (version 17.1.221229.55965, Quasar) with a directDIA search against the human canonical database (downloaded on the 27.11.2023, 20,428 sequences). For the search, the BSG Factory Settings were used with the following adaptations. For fixed modifications, carbamidomethyl on cysteine residues was set and for variable modifications, oxidation on methionine and protein N-terminal acetylation were set. The number of variable modifications was limited to 5. The Pulsar search was limited to trypsin cleavage-specific peptides with a minimum length of 7 and a maximum of 52 amino acids. The number of missed cleavages was set to 2. For the DIA analysis/calibration the maximum intensity was used for the mz extraction employing a local RT regression (non-linear). For the identification, an experiment-wide Q-value cutoff of 0.01 was set on precursor and protein level, respectively. The run Q-value cutoff was set to 0.05. The PEP cutoffs were set to 0.2 and 0.75 on precursor and protein levels, respectively. Duplicated assays were excluded from the analysis. Decoys were generated using the mutated option, and the decoy limit strategy was set to dynamic. No imputation of missing precursors was performed. Only identified proteotypic precursors were considered for qualification. Quantification was performed on MS2 level using the area. Cross-normalization was turned off. The mean precursor quantities (grouped by stripped peptide sequence) were used without setting a limiting top n threshold. The Spectronaut peptide-level results were used for further analysis. Spectronaut quantified 130,075 peptide precursors, corresponding to 7991 unique protein identifiers. Peptide signals were median-normalized by subtracting the median of all intensities within one sample from all intensities and scaling the values to the experiment-wide median. Protein abundances were inferred using the mean of top six peptides per unique protein identifier using the peptide ranks in the negative control. This ensured that the same peptides were quantified across the experiment. If a protein identifier had less than six peptides, the average of the top $n$ peptides was used. No data imputation was performed, due to only 1.9% missing values. To validate that the RNAi targets were knocked down, ratios were derived against the control treatment and log2FC and $P$ values (unpaired, two-tailed, Student $t$ test) were calculated. $P$ values were adjusted for multiple testing (BH correction).

## Data availability

The datasets produced in this study are available in the following databases: AP-MS of SLC interactome: ProteomeXchange Consortium via the PRIDE partner repository PXD055605; Full proteome profiling of seven human cell lines: ProteomeXchange Consortium via the PRIDE partner repository PXD051747; Full proteome profiling of RNAi-treated cell lines: ProteomeXchange

Consortium via the PRIDE partner repository PXD055192; AP-MS of SLC16A6 phosphomutants: ProteomeXchange Consortium via the PRIDE partner repository PXD055141; Predicted structures of SLC interactions: Zenodo, 14731200 (https://zenodo.org/records/14731200); Cytometry files used for protein stability assessment: Zenodo, 12758952 (https://zenodo.org/records/12758952); Subcellular localization data containing images, results and macros: BioImage Archive, S-BIAD1105 (https://doi.org/10.6019/S-BIAD1105); Protein interactions reported in the study: RESOLUTE database, proteomics (https://re-solute.eu/resources/dashboards/proteomics/network/); Protein interacts reported in the study: IMEx consortium through the IntAct database (Del Toro et al, 2022), IM-30161 (https://www.imexconsortium.org/).

The source data of this paper are collected in the following database record: biostudies:S-SCDT-10_1038-S44320-025-00109-1.

## Peer review information

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

## Acknowledgements

This study received funding from the RESOLUTE consortium. RESOLUTE has received funding from the Innovative Medicines Initiative 2 Joint Undertaking under grant agreement No 777372. This Joint Undertaking receives support from the European Union's Horizon 2020 research and innovation programme and EFPIA. The last year of work, including validation of data and writing of the manuscript, was supported mainly by the Austrian Academy of Sciences. This article reflects only the authors' views and neither IMI nor the European Union and EFPIA are responsible for any use that may be made of the information contained therein. GS-F was supported by the Austrian Academy of Sciences throughout. We thank the CeMM Proteomics-Metabolomics facility and the IMP (Research Institute of Molecular Pathology) proteomics facility for support in MS data acquisition, especially Andrea Rukavina for the full proteome profiling of RNAi-treated samples and especially Elisabeth Roitinger and

Gabriela Krssakova from the IMP for MS-data acquisition support. We thank Klaus Kratochwill from the Medical University of Vienna for providing access to the mass spectrometer throughout the project. We want to thank Georg Winter and lab members, especially Natalie Scholes, for assistance in setting up the protein stability assay pipeline. Stefan Kubick and the CeMM Molecular Discovery Platform (MDP) team members for support with the screening equipment and CeMM IT team, especially Patricia Carey for their support. We further want to thank Anthony Orth at Novartis Pharma AG BR DSc for providing interactor cDNA constructs. The prediction of structures by AlphaFold have been achieved using the Vienna Scientific Cluster (VSC). We want to thank Ariel Bensimon for providing the HAP1 cell line expressing MTCH2. We thank Gabriel Onea for critical reading and for providing feedback on the manuscript.

## Author contributions

**Fabian Frommelt**: Conceptualization; Data curation; Formal analysis; Supervision; Validation; Investigation; Visualization; Methodology; Writing—original draft; Project administration; Writing—review and editing. **Rene Ladurner**: Conceptualization; Data curation; Formal analysis; Supervision; Validation; Investigation; Visualization; Methodology; Writing—original draft; Writing—review and editing. **Ulrich Goldmann**: Conceptualization; Data curation; Formal analysis; Funding acquisition; Visualization; Methodology; Writing—review and editing. **Gernot Wolf**: Investigation; Methodology. **Alvaro Ingles-Prieto**: Investigation; Methodology. **Eva Lineiro-Retes**: Data curation; Investigation; Methodology. **Zuzana Gelova**: Validation; Investigation. **Ann-Katrin Hopp**: Visualization. **Eirini Christodoulaki**: Formal analysis; Visualization. **Shao Thing Teoh**: Visualization; Writing—review and editing. **Philipp Leippe**: Visualization; Writing—review and editing. **Brianda L Santini**: Validation; Visualization; Writing—review and editing. **Manuele Rebsamen**: Investigation. **Sabrina Lindinger**: Data curation; Validation; Investigation. **Iciar Serrano**: Data curation; Validation; Investigation. **Svenja Onstein**: Investigation; Methodology. **Christoph Klimek**: Investigation; Methodology. **Barbara Barbosa**: Investigation; Methodology. **Anastasiia Pantielieieva**: Data curation. **Vojtech Dvorak**: Visualization. **J Thomas Hannich**: Supervision; Investigation. **Julian Schoenbett**: Data curation; Validation. **Gilles Sansig**: Data curation; Validation. **Tamara AM Mocking**: Data curation; Formal analysis; Validation. **Jasper F Ooms**: Validation. **Adriaan P IJzerman**: Resources; Supervision; Funding acquisition. **Laura H Heitman**: Resources; Supervision; Funding acquisition. **Peter Sykacek**: Resources; Formal analysis; Methodology; Writing—review and editing. **Juergen Reinhardt**: Conceptualization; Resources; Supervision; Funding acquisition. **André C Müller**: Conceptualization; Resources; Supervision; Funding acquisition; Methodology. **Tabea Wiedmer**: Conceptualization; Supervision; Funding acquisition; Writing—original draft; Project administration; Writing—review and editing. **Giulio Superti-Furga**: Conceptualization; Resources; Supervision; Funding acquisition; Methodology; Writing—original draft; Project administration; Writing—review and editing.

Source data underlying figure panels in this paper may have individual authorship assigned. Where available, figure panel/source data authorship is listed in the following database record: biostudies:S-SCDT-10_1038-S44320-025-00109-1.

## Disclosure and competing interests statement

GS-F is a co-founder and owns shares of Solgate GmbH, an SLC-focused company.

# Expanded View Figures

**Figure EV1.  Characterization of SLC-protein interactions, SLCs as background and SLCs in complexes.**

(A) SLCs and PPIs reported across protein interaction databases. (B) Overlap of SLCs and PPIs across protein interaction databases. (C) Reported PPIs across the PPI library were plotted against frequency of identification in the CRAPome database. 127 SLCs were reported in the PPI library and were also part of CRAPome (CRAPome frequency >20% in violet, <20% in gray). (D) Comparison of the share of PPIs reported in the prey role for SLCs in BioGRID with a CRAPome frequency >20% ($n = 11$) against the share of PPIs reported in the prey role for SLCs with a CRAPome frequency <20% ($n = 75$). Data are presented as mean ± SD. The mean of the two groups showed a significant difference (two-sample t test, $P$ value: 0.005725). In the figure panel, $P$ values below 0.01 are indicated with "**". (E) Count of protein complexes (upper part) and protein complex subunits (lower part) included in CORUM. Subunits/ complexes were grouped into SLC, TM protein and no-TM protein and were counted across all reported protein complexes, taking into consideration multiple occurrences of subunits. (F) GALNT2 signal distribution across the SLC interactome. The left panel displays log2 transformed spectral counts for each SLC AP-MS experiment, grouped by scored interactions ($n = 21$, blue) and background/non-interacting ($n = 230$, gray). Lower and upper hinges of box plots correspond to the 25th and 75th percentiles, respectively. Lower and upper whiskers extend from the hinge to the smallest or largest value no further than the 1.5× interquartile range from the hinge, respectively. Black line represents the median log2 spectral count signal, and the black dots represent the signal per measurement. The right side shows the frequency of GALNT2 identification, scoring, and background in percentages. (G) The upper bar chart shows all SLCs for which GALNT2 was scored within the SLC interactome (log2FC against GFP, threshold of log2FC > 1). The lower section of the bar chart reports the confidence scores of predicted GALNT2-SLC complexes (high confidence, pDockQ threshold of >0.5 dashed black line, medium confidence, pDockQ threshold of >0.25 represented by a dashed gray line). A comparison against randomly sampled SLC-GALNT2 interaction (see "Methods" for details) showed, a significant difference between models of the interactions covered in the SLC interactome ($n = 21$) and the control set ($n = 21$; unpaired Student $t$ test, $P$ value $= 0.00695$). In the figure panel, $P$ values below 0.01 are indicated with "**". Lower and upper hinges of box plots correspond to the 25th and 75th percentiles, respectively. Lower and upper whiskers extend from the hinge to the smallest or largest value no further than the 1.5× interquartile range from the hinge, respectively. Black line represents the median and the black dots represent scores per complex. (H) The AlphaFold model of SLC30A1-GALNT2 is shown on the right side.

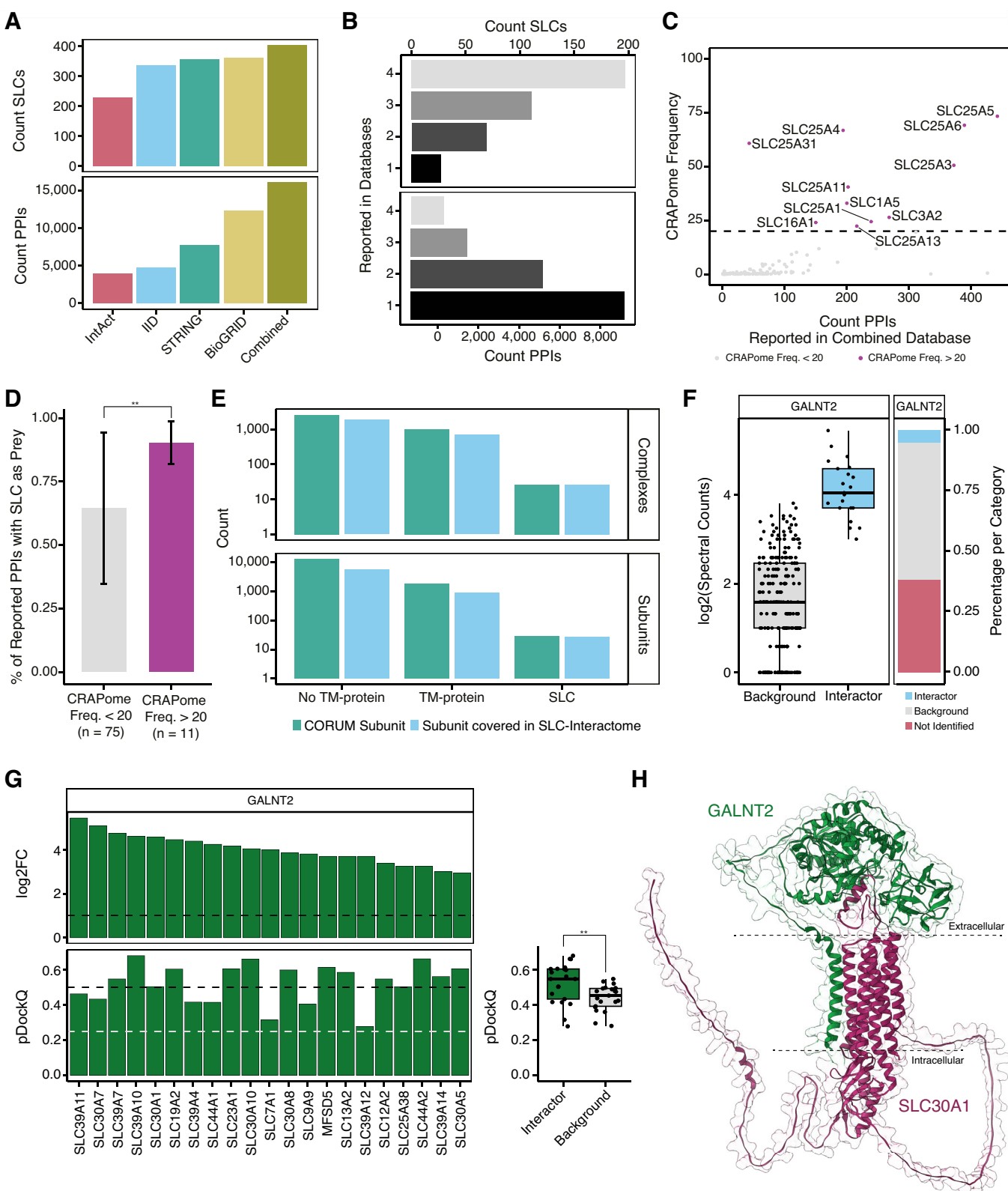

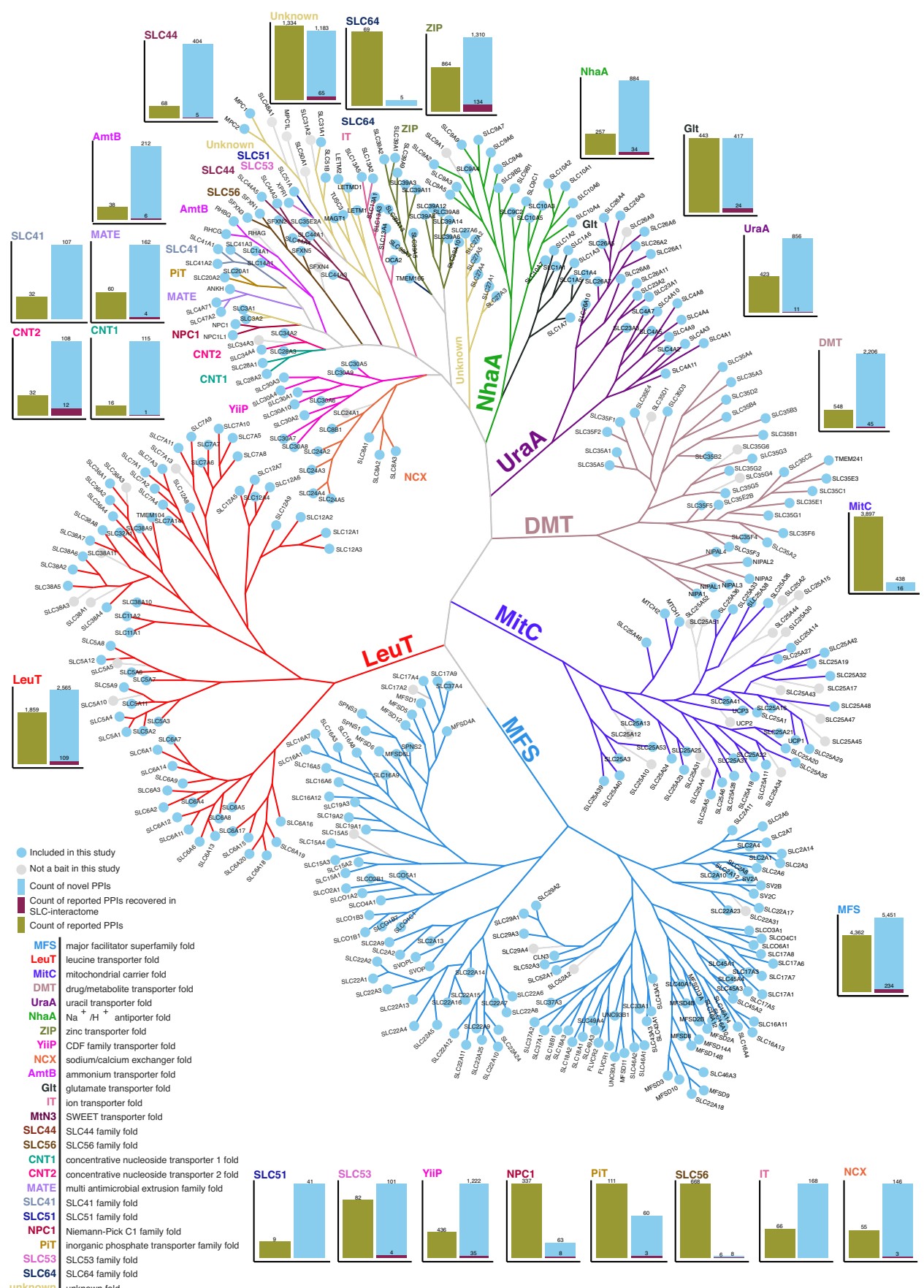

**Figure EV2.   Novel versus known protein interactions for each clade of the structural and evolutionary-based SLCome.**

For the construction of the phyologenetic tree the distance matrix of a previous classification of the SLC superfamily based on structural models was used (Ferrada and Superti-Furga, 2022). Each of the 25 distinct structural clades are represented in a different color. For each clade, the count of novel PPIs (light blue), literature mined PPIs (olive), and the shared PPIs (dark red) are reported. The 405 SLCs included in the study are colored light blue and all SLCs which were not included in the SLC interactome are colored gray.

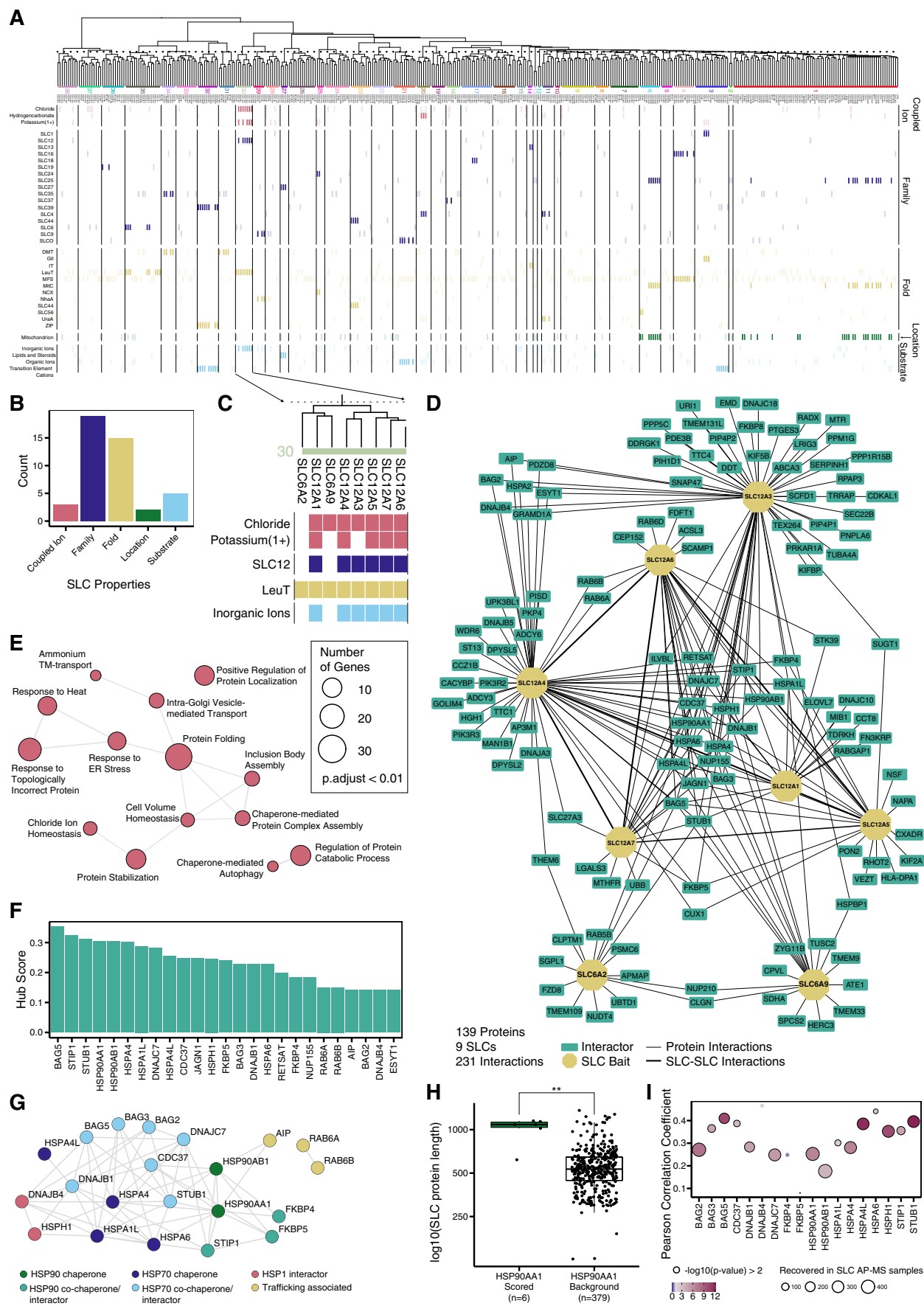

◀ **Figure EV3.  Functional property enrichment analysis showed SLC interactome similarity for SLC12 and SLC6 family members.**

(A) Significantly enriched SLC functional properties identified in the interactome SLC clustering analysis (Fisher's test $P < 0.2$). Colors indicate different SLC functional properties. (B) Sum of enriched SLC functional properties separated by SLC properties. (C) Significantly enriched SLC functional properties for PPI-interactome profile cluster 30 (Fisher's test $P < 0.2$). (D) Cluster-specific PPI-network obtained by the SLC baits (yellow octagons) grouped to cluster 30. Interactors are shown in green, and interactions between SLC baits are highlighted with a thick black line. (E) GO biological processing terms significantly enriched (Gene ontology overrepresentation test with a hypergeometric model, Benjamini–Hochberg corrected, $P$-adjusted $<0.01$) obtained by GSEA for interactors present in the specific PPI-network. The obtained terms were converted to a similarity network with GO semantic similarity and further filtered to showcase the least overlapping significant terms (similarity cutoff of 0.7). (F) The 25 most connected interactors (hub score) in the cluster 30-specific PPI-network. (G) PPI-network retrieved from literature (STRING confidence score $>0.4$, physical interactions only) for the 25 most connected interactors within cluster 30. Interactors were grouped as HSP90 chaperones (green), HSP90 co-chaperones/interactors (teal), HSP70 chaperones (blue), HSP70 co-chaperones/interactors (cyan), HSP1 interactors (red) or trafficking associated proteins (yellow). The four interactors not connected were removed from the PPI-network. (H) Distribution of protein length of SLCs interacting with HSP90AA1 ($n = 6$) and SLCs for which HSP90AA1 was found in the background ($n = 379$). Comparison showed a significant difference in the protein length (Student $t$ test $P$ value $= 1.285e^{-08}$). In the figure panel, $P$ values below 0.01 are marked with "**". Lower and upper hinges of box plots correspond to the 25th and 75th percentiles, respectively. Lower and upper whiskers extend from the hinge to the smallest or largest value no further than the 1.5× interquartile range from the hinge, respectively. Black line represents the median SLC-protein sequence length, and the black dots represent the sequence length per SLC for which HSP90AA1 was identified. (I) Correlation of the chaperone/ chaperone interacting protein abundances with the summed SLC tail length (N-terminal and C-terminal). Significant correlations with a $P$ value $< 0.01$ are indicated with black ring.

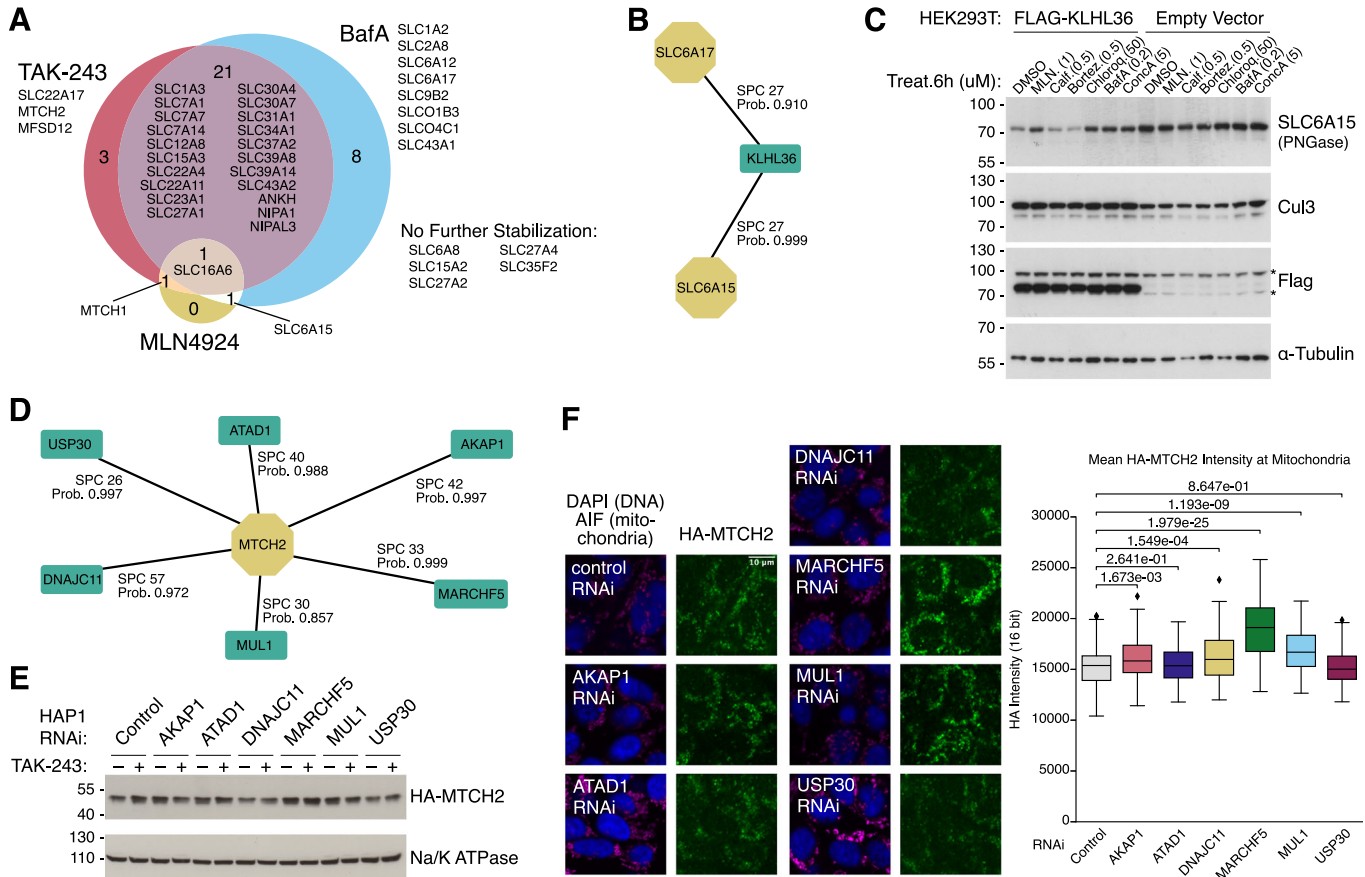

**Figure EV4. SLC levels are regulated by protein stability.**

(A) Venn diagram of drug treatment effects on SLC-protein stability. SLCs were considered to be stabilized by the respective drug if the GFP:RFP ratio was increased more than 10% compared to mock treated cells (*P* value < 0.05, independent *t* test). (B) SLC-protein interactions of KLHL36. The protein was not quantified in the background of any other SLC purification. (C) Western blot result of endogenous SLC6A15 in HEK 293T cells showing SLC6A15 degradation after overexpression of FLAG-tagged KLHL36. The samples shown in the blot were derived from the same experiment and gels/blots were processed in parallel. Uncropped images are provided as source data. (D) MTCH2 interactome showing 7 distinct interactors that were assessed in detail. (E) HAP1 cells expressing endogenously HA-tagged MTCH2 were transfected with RNAi against MTCH2 interactors. Depletion of the E3 ligase proteins MARCHF5 and MUL1 stabilizes MTCH2 levels as strongly as ubiquitination inhibition by TAK-243. (F) Immunofluorescence example images of HA-MTCH2 and quantification at mitochondria as identified by staining for AIF (apoptosis-inducing factor). Over 10,000 cells were imaged per condition; unpaired *t* test was used to compare treatments (*n* = 112 images per sample). Lower and upper hinges of box plots correspond to the 25th and 75th percentiles, respectively. Lower and upper whiskers extend from the hinge to the smallest or largest value no further than the 1.5× interquartile range from the hinge, respectively. Black line represents the mean and the black dots represent outliers. Source data are available online for this figure.

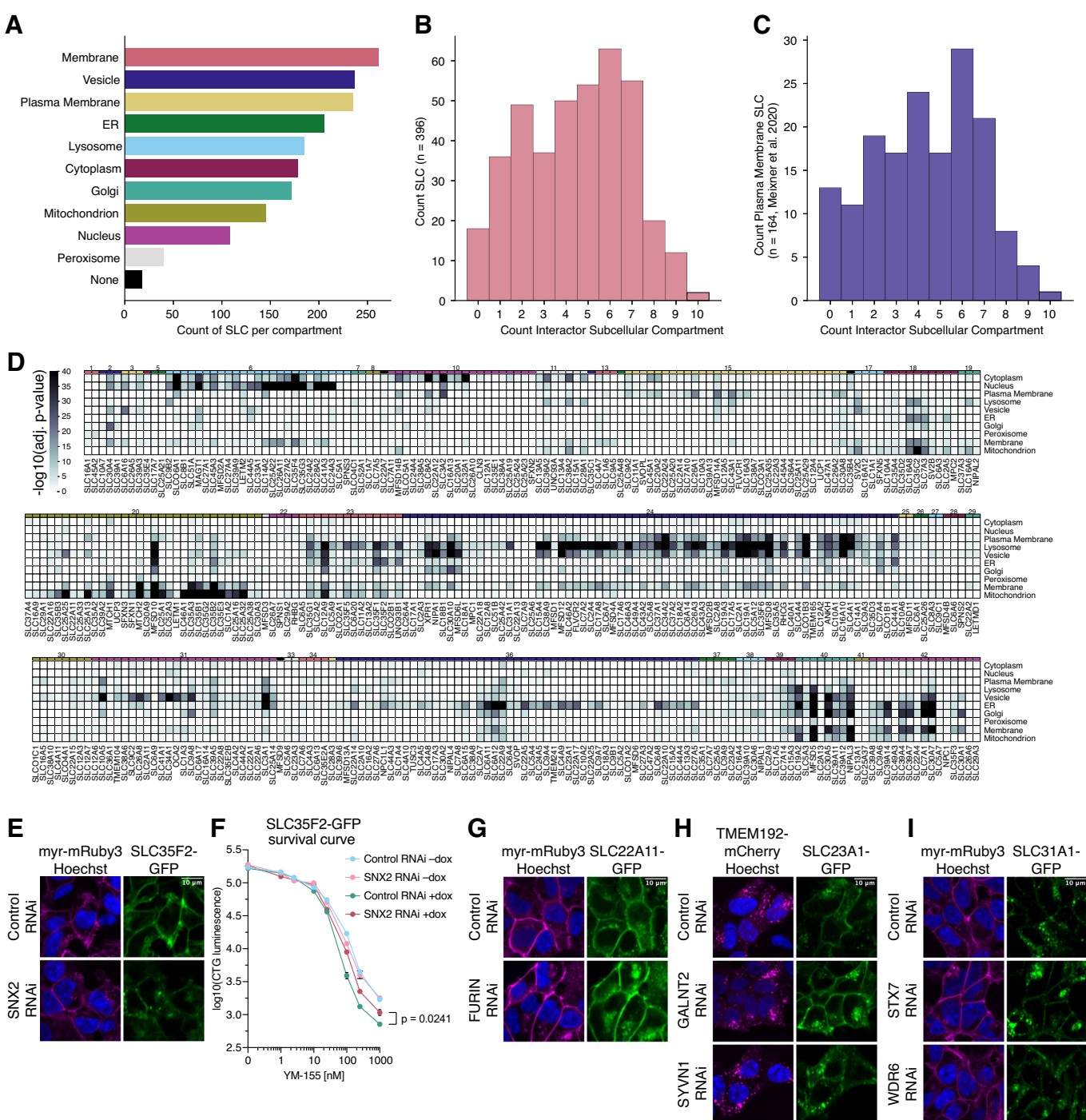

**Figure EV5.** **Proteins affecting trafficking and subcellular localization of SLCs.**

(**A**) GSEA of cellular compartments was performed for each SLC interactome and the resulting GO terms were hierarchically assigned to 10 main compartments (*P* value cutoff 0.05; see "Methods" for details). (**B**) A majority of SLC interactomes contain interaction partners from several subcellular compartments (mean: 4.5 compartments per SLC). (**C**) This notion also holds true for a subset of SLCs described previously as uniquely plasma membrane-associated (mean: 4.3 compartments). (**D**) The results from the interactome–compartment enrichment analysis were used to cluster SLCs. Strong clusters can be identified for some mitochondria baits (e.g., MTCH2), lysosomal baits (e.g., MFSD12) or Golgi baits (e.g., SLC30A5). (**E**) Representative images of plasma membrane- and Golgi-associated SLC35F2 after RNAi (see Fig. 6D for a quantification of the effect; size bar, 10 μm for all images). (**F**) SNX2 depletion attenuates SLC35F2-mediated import of YM-155. After 48 h RNAi, SLC35F2-GFP expression was induced using doxycycline and cells were subjected to YM-155 at the indicated concentrations for 24 h. Cell viability was measured using CellTiter-Glo luminescence (*n* = 3, mean ± SEM). Induced conditions at 1 μM YM-155 were compared using unpaired *t* test. (**G**) Representative images of plasma membrane- and Golgi-associated SLC22A11-GFP. Depletion of endopeptidase FURIN increased the plasma membrane-associated GFP intensity but not the Golgi-associated GFP intensity. (**H**) Representative images of plasma membrane- and lysosome-associated SLC23A1-GFP. Whereas depletion of GalNAc transferase GALNT2 increased GFP signals overall, depletion of SYVN1 led to a pronounced increase of GFP overlapping with the lysosomal RFP reference (see Fig. 6D for quantifications). (**I**) Representative images of plasma membrane- and vesicle-associated SLC31A1-GFP. Depletion of syntaxin-7 increased overall GFP signal while depletion of WDR6 increased GFP intensity at the plasma membrane (see Fig. 6D for quantifications).

