## [Peer Review File · Molecular Systems Biology]

The solute carrier superfamily interactome

Fabian Frommelt, Rene Ladurner, Ulrich Goldmann, Gernot Wolf, Alvaro Ingles-Prieto, Eva Lineiro-Retes, Zuzana Gelova, Ann-Katrin Hopp, Eirini Christodoulaki, Shao Teoh, Philipp Leippe, Brianda L. Santini, Manuele Rebsamen, Sabrina Lindinger, Iciar Serrano, Svenja Onstein, Christoph Klimek, Barbara Barbosa, Anastasiia Pantielieieva, Vojtech Dvorak, J. Hannich, Julian Schoenbett, Gilles Sansig, Tamara Mocking, Jasper Ooms, Adriaan IJzerman, Laura Heitman, Peter Sykacek, Juergen Reinhardt, André Müller, Tabea Wiedmer, and Giulio Superti-Furga

Corresponding author(s): Giulio Superti-Furga (gsupert@cemm.oeaw.ac.at)

Review Timeline:

Submission Date:	17th Oct 24
Editorial Decision:	28th Nov 24
Revision Received:	30th Jan 25
Editorial Decision:	24th Feb 25
Revision Received:	28th Mar 25
Accepted:	11th Apr 25

Editor: Poonam Bheda

Transaction Report:

28th Nov 2024

Manuscript Number: MSB-2024-12688

Title: The solute carrier superfamily interactome

Dear Dr. Superti-Furga,

Thank you for the submission of your manuscript to Molecular Systems Biology. We have now received feedback from the three reviewers who agreed to evaluate your manuscript. As you will see from the reports below, the referees acknowledge the interest of the study and are overall supporting publication of your work pending appropriate revisions.

I think that the recommendations of the reviewers are rather clear and I therefore do not see the need to repeat the comments listed below. All issues raised would need to be satisfactorily addressed. Please let me know in case you would like to discuss in further detail any of the reviewer comments or your proposed revisions, I would be happy to schedule a call.

We require:

1) A .docx formatted version of the manuscript text (including legends for main figures, EV figures and tables). Please make sure that the changes are highlighted to be clearly visible. Alternatively you may choose to submit your manuscript as a LaTeX file.

4) A .docx formatted letter INCLUDING the reviewers' reports and your detailed point-by-point responses to their comments. As part of the EMBO Press transparent editorial process, the point-by-point response is part of the Peer Review File (PRF), which will be published alongside your paper.

5) A complete author checklist, which you can download from our author guidelines (<https://www.embopress.org/page/journal/17574684/authorguide#submissionofrevisions>). Please insert information in the checklist that is also reflected in the manuscript. The completed author checklist will also be part of the PRF.

6) Please note that all corresponding authors are required to supply an ORCID ID for their name upon submission of a revised manuscript.

7) It is mandatory to include a 'Data Availability' section after the Materials and Methods. Before submitting your revision, primary datasets produced in this study need to be deposited in an appropriate public database, and the accession numbers and database listed under 'Data Availability'. Please remember to provide a reviewer password if the datasets are not yet public (see <https://www.embopress.org/page/journal/17574684/authorguide#dataavailability>).

This study includes no data deposited in external repositories.

8) All Materials and Methods need to be described in the main text using our 'Structured Methods' format, which is required for all research articles. According to this format, the Methods section includes a Reagents and Tools Table (listing key reagents, experimental models, software and relevant equipment and including their sources and relevant identifiers) followed by a Methods and Protocols section describing the methods using a step-by-step protocol format. The aim is to facilitate adoption of the methodologies across labs. Please upload the Reagents and Tools table as a separate document when submitting your revised manuscript. More information on how to adhere to this format as well as a downloadable template (.docx) for the Reagents and Tools Table can be found in our author guidelines:

<https://www.embopress.org/page/journal/17444292/authorguide#structuredmethods>

9) For data quantification: please specify the name of the statistical test used to generate error bars and P values, the number (n) of independent experiments (specify technical or biological replicates) underlying each data point and the test used to calculate p-values in each figure legend. The figure legends should contain a basic description of n, P and the test applied. Graphs must include a description of the bars and the error bars (s.d., s.e.m.). Please provide exact p values.

10) Our journal encourages inclusion of *data citations in the reference list* to directly cite datasets that were re-used and obtained from public databases. Data citations in the article text are distinct from normal bibliographical citations and should directly link to the database records from which the data can be accessed. In the main text, data citations are formatted as follows: "Data ref: Smith et al, 2001" or "Data ref: NCBI Sequence Read Archive PRJNA342805, 2017". In the Reference list, data citations must be labeled with "[DATASET]". A data reference must provide the database name, accession number/identifiers and a resolvable link to the landing page from which the data can be accessed at the end of the reference. Further instructions are available at .

11) We replaced Supplementary Information with Expanded View (EV) Figures and Tables that are collapsible/expandable online. A maximum of 5 EV Figures can be typeset. EV Figures should be cited as 'Figure EV1, Figure EV2' etc... in the text and their respective legends should be included in the main text after the legends of regular figures.

<https://www.embopress.org/page/journal/17574684/authorguide#expandedview>

13) Author contributions: CRediT has replaced the traditional author contributions section because it offers a systematic machine readable author contributions format that allows for more effective research assessment. Please remove the Authors Contributions from the manuscript and use the free text boxes beneath each contributing author's name in our system to add specific details on the author's contribution. More information is available in our guide to authors.

Please also suggest a striking image or visual abstract to illustrate your article as a PNG file 550 px wide x 300-600 px high. Share synopsis text and image, as well as eTOC:

Please note that these would be the final versions and changes during proofing are usually not allowed

16) As part of the EMBO Publications transparent editorial process initiative (see our policy here: https://www.embopress.org/transparent-process#Review_Process), Molecular Systems Biology will publish online a Peer Review File (PRF) to accompany accepted manuscripts.

In the event of acceptance, this file will be published in conjunction with your paper and will include the anonymous referee reports, your point-by-point response and all pertinent correspondence relating to the manuscript. Let us know whether you agree with the publication of the PRF and as here, if you want to remove or not any figures from it prior to publication. Please note that the Authors checklist will be published at the end of the PRF.

Molecular Systems Biology has a "scooping protection" policy, whereby similar findings that are published by others during review or revision are not a criterion for rejection. Should you decide to submit a revised version, I do ask that you get in touch after three months if you have not completed it, to update us on the status.

I look forward to receiving your revised manuscript.

Yours sincerely,

Poonam Bheda, PhD
Scientific Editor
Molecular Systems Biology

Reviewer #1:

The manuscript by the Superti-Furga group, titled "The Solute Carrier Superfamily Interactome," systematically characterizes the protein-protein interactions (PPIs) of approximately 450 SLCs in HEK293 cells using affinity-purification followed by mass spectrometry proteomics. For 65 SLC hits, the authors further investigated the abundance and subcellular localization of GFP-tagged SLC proteins via flow cytometry and microscopy, with supporting evidence from western blot biochemistry. In this analysis, ~152 perturbation conditions (including 134 RNAi knockdowns and 18 cDNA overexpression of putative binding partners) were applied to assess the impact on SLC abundance and localization. This manuscript is part of a back-to-back submission of four RESOLUTE team papers that provide a systems-level analysis of human SLCs, covering PPIs (this study), genetic interactions, metabolic/transcriptomic landscapes, and computational annotation. These studies represent valuable resources for exploring cellular metabolism and transporter functions, and the reviewer appreciates the RESOLUTE team's contributions to the field. For this PPI manuscript, the reviewer supports publication with minor text edits.

1) The sentence "Further, we compared the log₂FC of each SLC bait against GFP controls as well as all other AP-MS samples (Appendix Fig. S1D). Several SLCs showed lower enrichment against the membrane background of other SLC AP-MS samples than to GFP controls. We concluded that AP-MS data from other SLCs represented a better strategy to correct for co-purified SLCs." is difficult to understand. Does this imply that some SLCs (e.g., SLC-x) show lower levels in their own bait (SLC-x) AP-MS than in AP-MS samples of other SLCs (e.g., SLC-y bait)? If so, perhaps these SLCs (SLC-x) be excluded from analysis.

2) The limitations of a high-throughput SLC PPI procedures should be clearly stated. For example, SLCs located intracellular organelles of low abundance, may require pure organelle fractions and optimized conditions for the membrane, such as endolysosomes, peroxisomes, and the inner mitochondrial membrane, as seen in the work by Shi et al. (Mol Cell, 2024) on a mitochondrial SLC.

3) The identification of a loop region degron in SLC16A6 is novel and underscores how PPIs can reveal previously uncharacterized regulatory mechanisms in SLCs. However, since the phosphomimetic mutant S240D also stabilizes SLC16A6, the data do not convincingly support a phospho-degron mechanism. Instead, it may suggest an S240-dependent loop degron, with prior evidence indicating this serine as a post-translational modification site in high-throughput phosphoproteomics studies.

Reviewer #2:

In this manuscript, the authors describe comprehensive mapping of the interactome of close to 400 SLC proteins using AP-MS, as well as robust computational validation of their results, and large-scale functional assessment of the identified interactors using RNA interference and protein localization/stability measurements. Additionally, the authors provide a more in-depth examination of the biological role of the interactions of a range of SLCs as further support for the quality of their reported interactions.

Overall, this is excellent work, providing an unprecedented view of the complex interactions of the majority of human SLC proteins. The data provided is richly annotated and well validated, and it should prove to be a valuable resource to the scientific community. The authors have also been very up-front about the limitations of their approach, which is appreciated and should be helpful to readers making use of the data.

I have only minor comments (see below) and no trouble recommending this manuscript for publication.

Minor Comments

- Parts of Figure 1A are hard to read (graphics/text are too light). I'm also not convinced this panel is even necessary (it really doesn't add anything to the text-based description).

- Please provide a better explanation of panel 1F.

- Why are fewer scored interactions reported in Dataset EV1 than are indicated in Figure 1E?
- The legend for Figure 2A appears to indicate the wrong colors for the bars in the graph (SLC vs TM).
- I'm not sure I see the relevance of Figure 2B/C. It really isn't surprising that you'd see more PPIs for well-studied SLCs in a dataset compiled from previously published data. If an SLC was in 10 PPI studies, it is more likely to have more PPIs reported than one that has only been studied once. The fact that the authors' interactome doesn't show this is only mildly more interesting; it's good that their reported interaction counts aren't biased by previously reported information, but again it's not really surprising. The manuscript is already very lengthy and information-dense, so I don't feel this really needs to be included.
- For Appendix Figure S8, why were these measurements only performed in a single well? Showing these results without replication/error makes them hard to properly interpret.
- Page 10, the last paragraph references Fig. EV4H. There is no panel H in this figure. Where is the intended data?
- The destabilization of MTCH1-GFP by USP30 is not very convincing from the data presented in Figure 4G. It seems minor at best. How many times was this experiment performed?
- Paragraph 4, page 11 references Figure 4B. I believe this should be 4C.
- I would like to suggest that the authors acknowledge in their revised version the importance of complementary protein-protein interaction mapping techniques, such as MaMTH, SIMPL, and other protein complementation assays. These approaches would be valuable in providing a fuller and higher-quality view of the SLC interactome and addressing any limitations of AP-MS in this context.

Reviewer #3:

In this paper, the authors tackle one of the most critical and challenging issues in life sciences. Despite the solute carrier (SLC) family's significant role in numerous vital physiological functions, much about their functions remains unknown. Naturally, our understanding of their protein-protein interactions (PPIs) is also insufficient. This study aims to comprehensively elucidate the PPI network of the SLC family by overexpressing 396 types of SLC transporters and analyzing their PPIs using mass spectrometry-based proteomics and machine learning (ML). Some PPIs were further evaluated using RNA interference, successfully detecting interactions primarily related to protein stability and localization. The methodology of conducting numerous wet-lab experiments and integrating large datasets with ML is innovative. The resulting interactome is expected to benefit the research community, and some of the individual validation experiments also yielded intriguing results. On the other hand, there are limitations inherent to the wet experiments, some of which the authors have mentioned in their discussion. However, the most significant limitation—the use of an overexpression system for SLC molecules—has not been explicitly addressed. Additionally, the use of cultured cells may not fully reflect the physiological PPIs of SLCs *in vivo*. While these points do not detract from the study's value, the authors might consider addressing them more explicitly.

Major Points

1. As mentioned above, the experiments were conducted using an overexpression system in cultured cells, raising concerns that the detected interactions may include those occurring outside the transporters' native functional contexts. Indeed, many of the PPIs individually analyzed by the authors are related to localization.
2. The title of the paper is ambitious, reflecting the authors' aim to obtain 'The solute carrier superfamily interactome.' While this study makes significant strides in this direction, it may not yet fully capture the comprehensive map implied by the title. A more specific title might better reflect the current scope of the work.
3. How would the interactome data differ if interactions were categorized based on whether they occur at the locations where SLC molecules naturally perform their transport functions or elsewhere? Although the experimental system may have inadvertently captured PPIs across different stages of an SLC transporter's lifecycle, this is not necessarily a drawback. However, from a biological perspective, it might be beneficial to explicitly distinguish between these differences. Accordingly, it could be worth considering structuring the manuscript to address the interactome with attention to these temporal scales.
4. The PPI examples used to validate the interactome's reliability in Fig. EV1 and Fig. 2 are all located on the plasma membrane. While these are valid examples, they may not fully align with the paper's focus on novel findings related to trafficking and localization. It might strengthen the narrative to include examples more directly tied to these findings.

Minor Points

1. While 405 molecules were tested, were there any molecules excluded due to issues such as lack of expression, like SLC9A1? It would be helpful if the manuscript addressed this point or provided any information to the relevant section if it is already covered.
2. There appears to be inconsistency in the labeling of molecules with SLC names in the text and figures. Ensuring uniform

nomenclature would help avoid confusion.

3. Regarding the example of interactions between MCTs and BSG or EMB, as well as well-known ACE2 and Collectrin with SLC6A19, should there not be a mention of tissue specificity in these cases?
4. p.7, l. 13: CATs are not known to form heterodimers, so they should not be mentioned here.
5. p.7, l. 17: The physiological SLC7A9 binding partner is SLC3A1, not SLC3A2.
6. Fig. EV1G: Showing predicted structures for multiple SLC7 family members and SLCs seems less meaningful. Moreover, the SLC7A7-SLC3A2 complex, while lacking structural analysis, is biochemically well-supported. Highlighting the superiority of the interactome with other examples might be more effective.
7. Similarly, the CryoEM structure of SLC9A1 and CHP1 makes the importance of predicting the structure of SLC9A2 and CHP1 less clear.
8. p. 11, l. 30: The text mentions GFP-SLC16A6, but was the GFP fusion at the N-terminus? Fig. 4A suggests a C-terminal fusion.
9. In some cases, citations, such as those related to complex formation, reference recent structural analyses, even though these facts had been previously established through biochemical and other methods. It might be helpful to acknowledge earlier studies alongside the recent work to provide a more complete context.
10. The inclusion of SciWheel.com links in the references might impact readability and make it more challenging to search for citations during the review process. Removing these links could enhance accessibility and improve the overall review experience.

Point-by-point response for the manuscript: The solute carrier superfamily interactome.

Response to reviewers of "The solute carrier superfamily interactome"

Below, the reviewers' comments are represented in black, followed by our responses in green. To simplify revision, page and line numbers referring to the 'track changes' document in the "simple markup" view were added to each answer. Additionally, a list of all figure panels that were revised or had minor text edits (e.g. axis-labels) was appended.

Response to reviewer #1:

The manuscript by the Superti-Furga group, titled "The Solute Carrier Superfamily Interactome," systematically characterizes the protein-protein interactions (PPIs) of approximately 450 SLCs in HEK293 cells using affinity-purification followed by mass spectrometry proteomics. For 65 SLC hits, the authors further investigated the abundance and subcellular localization of GFP-tagged SLC proteins via flow cytometry and microscopy, with supporting evidence from western blot biochemistry. In this analysis, ~152 perturbation conditions (including 134 RNAi knockdowns and 18 cDNA overexpression of putative binding partners) were applied to assess the impact on SLC abundance and localization. This manuscript is part of a back-to-back submission of four RESOLUTE team papers that provide a systems-level analysis of human SLCs, covering PPIs (this study), genetic interactions, metabolic/transcriptomic landscapes, and computational annotation. These studies represent valuable resources for exploring cellular metabolism and transporter functions, and the reviewer appreciates the RESOLUTE team's contributions to the field. For this PPI manuscript, the reviewer supports publication with minor text edits.

We thank the reviewer for their detailed and positive evaluation of our work. We appreciate the constructive comments and the reviewers' support for the publication of our manuscript.

1. The sentence "Further, we compared the log₂FC of each SLC bait against GFP controls as well as all other AP-MS samples (Appendix Fig. S1D). Several SLCs showed lower enrichment against the membrane background of other SLC AP-MS samples than to GFP controls. We concluded that AP-MS data from other SLCs represented a better strategy to correct for co-purified SLCs." is difficult to understand. Does this imply that some SLCs (e.g., SLC-x) show lower levels in their own bait (SLC-x) AP-MS than in AP-MS samples of other SLCs (e.g., SLC-y bait)? If so, perhaps these SLCs (SLC-x) be excluded from analysis.

We apologize for any confusion caused by the above-mentioned section. In the section we discuss a comparative analysis of the log₂FC for each SLC bait (SLC-x) against GFP negative controls, as well as against all other AP-MS samples (SLCs plus GFP negative controls). The second fold change was derived by calculating the mean across the entire dataset, excluding the AP-MS data for SLC-x. Each SLC bait signal showed an enrichment with a fold change of at least 9.45 compared to GFP and of a fold change of at least 2.8 compared to all other samples. This demonstrates that all SLC baits were enriched in both comparisons.

However, 67.40% of all SLC baits showed an equal or higher fold change against GFP negative controls, indicating that the SLC-x protein was more abundant in other SLC AP-MS experiments compared to the signal of most SLCs retrieved in GFP negative controls. Consequently, we concluded that the entire dataset provides an additional filter to penalize the complex membrane background. In our scoring and

filtering approach, we considered both fold changes. We acknowledge the reviewer's concern that the criticized section could be misleading, and we have tried to explain better. Further we have revised **Appendix Fig. S1D** and updated the figure caption accordingly. Changes in the main manuscript:

“Further, we compared the log₂FC of each SLC bait against GFP controls as well as all other AP-MS samples (SLCs and GFP) (**Appendix Fig. S1D**). Several SLCs showed lower log₂FC enrichment against the membrane background of other SLC AP-MS samples compared to GFP controls. Each SLC bait demonstrated a strong enrichment compared to GFP negative controls (minimum log₂FC of 3.24 across the SLC-interactome) and the background of the same SLC identified in other AP-MS samples (minimum log₂FC of 1.49 across the SLC-interactome). We concluded that AP-MS data from other SLCs provided a more accurate representation of the inherent background co-purified during affinity enrichment. Therefore, we used the entire dataset in the filtering and scoring strategy to correct for co-purified SLCs.” (p.4, l. 157-165)

We revised for **Appendix Fig. S1D** the following: **I**) Scaled the x- and y-axis to the same axis limits and axis-tick intervals, **II**) clarified the x- and y-axis labels, **III**) annotated the three SLCs for which the bait-self loop interactions were not scored, and **IV**) added a line through the origin to increase readability. In addition, we revised the **Appendix Fig. S1D** figure legend. (Appendix p.12, l. 385-393)

The three SLC-bait self-loops were penalized by our scoring approach due to either high entropy, as observed in SLC22A25 (indicating low reproducibility across biological replicates), or a CRAPome frequency score above 99%, indicating that these SLCs were found in almost all negative controls of the CRAPome study, as seen for the SLC25A3 and SLC25A5 baits (Mellacheruvu et al, 2013).

2. The limitations of a high-throughput SLC PPI procedures should be clearly stated. For example, SLCs located intracellular organelles of low abundance, may require pure organelle fractions and optimized conditions for the membrane, such as endolysosomes, peroxisomes, and the inner mitochondrial membrane, as seen in the work by Shi et al. (Mol Cell, 2024) on a mitochondrial SLC.

We thank the reviewer for pointing out the necessity to emphasize this more. We discussed several limitations of our high-throughput AP-MS approach, including the employed detergents for native lysis, the absence of SLC isoforms, and the scoring approach. (p.16, l. 699-705, and p.17, l. 722-729)

We acknowledge the difficulties of performing affinity purifications of baits localized at membrane-bound organelles. However, given the large number of SLC baits included in our study, and our aim to compare the interactomes at the superfamily level, we adopted a standardize procedure, an approach also used by other large-scale interaction proteomics studies coving baits from multiple subcellular compartments (Huttlin et al, 2015, 2017, 2021; Buljan et al, 2020).

During the generation of our dataset, we recognized the necessity to use a combination of subcellular fractionation and affinity enrichment for difficult to purify mitochondrial SLCs. We thus increased protein/cell input (from 40, to 80/120 million cells per replicate; see methods: Cell culture and mitochondrial fractionation, see **Fig. 1D** for count of baits) and performed crude subcellular enrichment by sucrose cushion prior to the affinity enrichment (Frezza et al, 2007). However, we did not implement

specific enrichment procedures for other organelles, as we did not experience the same difficulties (e.g.: for lysosomal localized SLCs we used SLC38A9 for benchmarking).

It is important to note that the localization of many SLCs remains ambiguous, with multiple localizations reported in the literature and protein localization databases (Meixner et al, 2020). The multiple reported localization and inconsistency of reported localizations are thoroughly discussed in the accompanying manuscript data- and knowledge-derived functional landscape of SLC (Goldmann et al, 2024).

Future studies could benefit from initially conducting a comprehensive profiling of purified organelles, using an approach introduced in a recent study (Hein et al, 2024). This would enable the generation of a detailed cell type specific subcellular localization map of the SLC baits. This map would serve to guide the selection of appropriate organellar enrichment procedures for specific SLC baits. However, this approach is constrained by the number of endogenously expressed SLCs per cell line, which averages 246.5 (see **Appendix Fig. S15A,B**), and would not resolve the issue that SLCs might localize to multiple subcellular compartments. While we acknowledge the limitations of not employing specific enrichment procedures, we believe there is significant value in maintaining a standardized and unbiased approach across multiple baits for an integrated analysis such as the one presented in our study.

To address the raised point, we extended the limitation section, now covering low abundance of SLCs at subcellular compartments. Adaptions in the manuscript:

“Second, we used a modified AP-MS workflow to recover low abundant mitochondrial baits by performing organellar enrichment for mitochondria prior to prior to affinity enrichment (see Methods for details). However, we did not implement subcellular enrichment for SLCs localized to other membrane-bound organelles. The diverse subcellular localization – ranging from a single organelle to multiple compartments – and the broad range of expression levels across the SLC superfamily may have impacted the purification of low-expressed SLCs using our standard high-throughput AP-MS workflow. To partially mitigate this, we increased the lysate input for low-expressed SLCs (see Methods for details).”
(p. 16, l. 706-713)

3. The identification of a loop region degron in SLC16A6 is novel and underscores how PPIs can reveal previously uncharacterized regulatory mechanisms in SLCs. However, since the phosphomimetic mutant S240D also stabilizes SLC16A6, the data do not convincingly support a phospho-degron mechanism. Instead, it may suggest an S240-dependent loop degron, with prior evidence indicating this serine as a post-translational modification site in high-throughput phosphoproteomics studies.

We thank the reviewer for pointing out the inconsistency of a phosphorylation-dependent degradation mechanism and the mixed outcome of our attempts at mimicking the phosphorylation using point mutations at the presumed phosphorylation sites. If phosphorylation of serine 240 and threonine 245 is indeed required for their recognition by the SCF as is implied by our experiments using non-phosphorylatable S240A or T245N mutants (**Fig. 5**), one could reasonably expect that a true phosphomimetic mutant would increase interaction with the SCF compared to its wildtype form, leading to a further destabilization or degradation of the SLC16A6 protein.

Contrary to this expectation, replacing serine at position 240 with the negatively charged aspartate strongly stabilized GFP–SLC16A6 (**Fig. 5D**) and strongly reduced its interaction with CUL1, SKP1, BTRC, and FBXW11 (**Fig. 5E**). Although an S240-dependent loop degron as suggested by the reviewer is an intriguing possibility, one would expect the T245E mutations to affect stability through a similar mechanism because of its proximity to S240, introducing negatively charged glutamate at the position of threonine, showed similar degradation to the wildtype form in our protein stability assay (**Fig. 5D**) and slightly increased interaction with CUL1 in AP-MS experiments (**Fig. 5E**). Perhaps in line with the expectation, this mutant was significantly affected in its plasma membrane association (**Appendix Fig. S11F**).

While we generated the aspartate and glutamate mutations in S240 and T245 “to potentially mimic phosphorylated residues at these positions and elicit increased degradation” (p.12, l.507-508), it is obvious that the S240D mutation could not successfully mimic pS240. We therefore already mentioned in our initial submission that “The S240D mutant was the most stable, suggesting that the aspartate substitution cannot substitute for phosphorylated serine.” (p.12, l.510-511)

We have searched the literature and have found only one attempt at mimicking phosphorylation of a β TrCP substrate (Zhao et al, 2010). In their experiments, substitution of serine 384 in YAP, which is part of the DSG recognition motif of β TrCP, with aspartate completely abrogated interaction with TRCP similar to an alanine substitution at this position. The authors explained this result with prior crystallographic results of the β TrCP– β -catenin interaction that showed direct hydrogen bonding with the phosphate group (Wu et al, 2003). Importantly, this result did not dissuade the authors from using the term “phosphodegron” for the recognition motif in YAP.

In summary, prior analysis of β TrCP–phospho-degron mechanisms are in line with and further extended with our analysis of phosphorylation mutants. We would therefore like to maintain that our extensive analysis of mutants and depletions in protein interaction, stability and localization assays support a phospho-degron mechanism of SLC16A6 degradation. However, to explain our observations better, we would like to link them to the literature by amending the results as follows:

“The S240D mutant was the most stable, suggesting that the aspartate substitution cannot substitute for phosphorylated serine. This is in line with previous observations that aspartate at the serine position of the DSG motif cannot mimic phosphorylated serine, presumably because the phosphate group is required for hydrogen bonding with β TrCP (Zhao et al. *Genes Dev* 2010; Wu et al. *Mol Cell* 2003).” (p. 12, l. 510-514)

Response to reviewer #2:

In this manuscript, the authors describe comprehensive mapping of the interactome of close to 400 SLC proteins using AP-MS, as well as robust computational validation of their results, and large-scale

functional assessment of the identified interactors using RNA interference and protein localization/stability measurements. Additionally, the authors provide a more in-depth examination of the biological role of the interactions of a range of SLCs as further support for the quality of their reported interactions. Overall, this is excellent work, providing an unprecedented view of the complex interactions of the majority of human SLC proteins. The data provided is richly annotated and well validated, and it should prove to be a valuable resource to the scientific community. The authors have also been very up-front about the limitations of their approach, which is appreciated and should be helpful to readers making use of the data. I have only minor comments (see below) and no trouble recommending this manuscript for publication.

We thank the reviewer for the very positive assessment of our work. We are grateful for their comments and feedback. We also appreciate their strong support for publishing our manuscript with only minor edits.

1. Parts of **Figure 1A** are hard to read (graphics/text are too light). I'm also not convinced this panel is even necessary (it really doesn't add anything to the text-based description).

We appreciate the reviewer's feedback regarding the colors and overall purpose of **Fig. 1A**. We agree that the colors and structure of **Fig. 1A** were not effectively conveying our intended message. Our goal was to highlight that the interactome is part of a larger effort to systematically characterize human solute carriers using the same cell line system with a multi-omics approach. **Fig. 1A**, is used as well in the three other accompanying manuscripts to ensure readers are aware of the other studies.

However, we see now the figure panel did not convey the intended message. To address this, we have updated **Fig. 1A**. The revised version shifts the focus from the method perspectives to our original intention emphasizing that there are accompanying manuscripts offering a complementary characterization of SLCs at different molecular organization layers. The updated figure now aligns better with the text in the manuscript and guides readers to the parallel metabolomics/transcriptomics characterization of the SLC-cell lines (Wiedmer, Shao et al.), the genetic interactions of SLCs (Leippe, Wolf et al.), and the functional landscape of SLCs (Goldmann, Wiedmer et al.) (see updated **Fig. 1A**).

2. Please provide a better explanation of panel 1F.

We extended the captions for **Fig. 1F**, providing now a detailed description for each of the graphs. We added the total number of SLCs, SLC-families and the corresponding numbers covered in the SLC-interactome. The changed figure caption reads now:

“(F) AP-MS coverage across the SLC superfamily. The left bar chart shows the coverage of SLC-baits (396 of 447 SLCs investigated in RESOLUTE) and families (68 of 70 SLC families) included in the SLC-interactome. The upper part of the split graph reports the counts of SLC-baits used within the study separated per SLC-family. The count of SLCs (y-axis) is plotted against the SLC-families (x-axis). SLCs used as bait in the SLC-interactome are marked in blue, SLCs not included in this study or filtered after scoring are coloured in yellow. The lower part of the bar chart represents the percent coverage per SLC-family, indicating in blue the percentage of family members reported in the SLC-interactome and in yellow the

percent of family members which are not included. The y-axis shows percentage coverage by family and the x-axis indicates SLC-families.” (p. 19, l.769-779)

3. Why are fewer scored interactions reported in **Dataset EV1** than are indicated in **Figure 1E**?

We thank the reviewer for recognizing this inconsistency between the data table and **Figure 1E**. This discrepancy arises from the removal of scored SLC-bait hits (the bait scored itself). A total of 402 baits were scored in their own AP-MS experiments. As these do not represent PPIs, we removed them from the result table. However, we report them in the complete unfiltered dataset used as input for our scoring (**Dataset EV9**) and provide the unfiltered SLC-interactome dataset under <https://resolute.eu/resources/dashboards/proteomics/downloads/> for download, thus ensuring full transparency and reproducibility.

Three SLC-bait self-enrichments were penalized by our scoring approach either due to high entropy as for SLC22A25 (low reproducibility across the biological replicates) or due to a too high CRAPome frequency (SLC25A3 and SLC25A5). The final dataset comprises PPIs of 396 SLCs, as for six SLCs (SLC25A27, SLC25A28, SLC25A39, SLC25A40, SLC25A41, SLC25A53, SLC17A4, SLC5A11) only the bait itself was passing our scoring and filtering approach. We thus report only the PPI dataset of 396 SLCs (**Fig. 1C**).

The count of scored PPIs **Fig. 1E** is misleading, thus we changed the legend labels from “Scored PPIs (n= 19,393)” to “**Scored PPIs (n = 18,991) and scored SLC-baits (n = 402)**”. This allows us to report the correct distribution of PPI-probabilities and ensures consistency with the number of reported PPIs in **Dataset EV1** and **Fig. 1C**. Further we adapted the Figure caption of **Fig. 1E** to “**(E)** A set of 18 features (**Table EV2**) and a curated list of labelled PPIs served as input for the scoring. Distribution of scored interactions **and scored SLC-baits as preys** (dark blue) versus background proteins (grey). For visualization of both distributions, the y-axis was cut at a density of 10.” (p. 19, l.769-772)

4. The legend for **Figure 2A** appears to indicate the wrong colors for the bars in the graph (SLC vs TM). I'm not sure I see the relevance of **Figure 2B/C**. It really isn't surprising that you'd see more PPIs for well-studied SLCs in a dataset compiled from previously published data. If an SLC was in 10 PPI studies, it is more likely to have more PPIs reported than one that has only been studied once. The fact that the authors' interactome doesn't show this is only mildly more interesting; it's good that their reported interaction counts aren't biased by previously reported information, but again it's not really surprising. The manuscript is already very lengthy and information-dense, so I don't feel this really needs to be included.

We thank the reviewer for noticing. We have updated the color indications in the figure caption to match the colors used in the bar plot of **Fig. 2A** panel.

We believe that **Fig. 2B/2C** highlight important benchmarking results. **Fig. 2B** underscores the necessity for a systematic study of SLCs, as the majority (375 out of 446) have a low number of reported interactions. Note that the PPI count in **Fig. 2B** does not distinguish whether an SLC was reported in the bait or prey role in the databases. Therefore, a high number of associated studies does not necessarily mean the SLC was used as bait. This figure sets the stage by highlighting the research need to systematically map interactions for SLCs. The threshold of 10 associated studies (although arbitrarily

chosen) reveals a research bias, with only 72 SLCs having more than 10 studies reporting interactions in PPI databases. This figure panel, therefore, emphasizes the need to systematically address this research gap and highlights the coverage achieved by the SLC-interactome (blue dots in the scatterplot). **Fig. 2C** demonstrates to a critical reader that the presented SLC-interactome is not biased by previously deposited PPI data. This benchmark figure convincingly shows that our methodology remains unbiased, even though well-characterized PPIs were used as labels for the ML-based scoring.

Therefore, we disagree and argue that figure panels **Fig. 2B** and **Fig. 2C** effectively fulfill their intended purposes, that we have generated a comprehensive and systematic dataset.

5. For **Appendix Figure S8**, why were these measurements only performed in a single well? Showing these results without replication/error makes them hard to properly interpret

The reviewer is right in identifying this as a weak point of the study and that without explanation it is difficult to understand. After a long debate, we propose to leave the figure in, but decided to explain the context much better and clarify the experimental limitations. The data are meant to represent a trend in the pharmacological confirmation of the proteostatic regulation. We reanalyzed the data and now show log-transformed median GFP:RFP ratios, with error bars representing the first and third quartiles and the minimal numbers of events per graph indicated in the figure legends.

The new **Appendix Figure S10** depicts the effects of RNAi and drug treatment for each SLC-GFP cell line in a separate panel. **Appendix Figure S11** depicts the effects of cDNA overexpression and drug treatment for each SLC-GFP cell line in a separate panel. We are confident that these revised figures allow for a more meaningful and careful interpretation.

We changed the main text to highlight this representation:

"We also tested pharmacological inhibition of degradation in addition to depleting or overexpressing the SLC interactor in HEK293 cells (**Appendix Fig. S10, S11**). The increase in protein levels observed after RNAi-mediated depletion was not further increased in some cases, suggesting that the main degrader of a specific SLC had already been eliminated (e.g. **Appendix Figure S10B**, SLC2A8–RNF167) or that the interactor was not directly linked to these degradation pathways (e.g. **Appendix Figure S10D**, SLC6A15–GOLGA5). In other cases, SLC levels were additionally increased by inhibiting ubiquitination or lysosome acidification (e.g. MTCH2–MUL1; SLC39A8–NFXL1), indicating either incomplete depletion, additional degraders or other stabilization mechanisms." (p.11, l. 444-451)

6. Page 10, the last paragraph references **Fig. EV4H**. There is no panel H in this figure. Where is the intended data?

We thank the reviewer for identifying the incorrect **Fig. EV4H** reference. During the manuscript preparation, we rearranged the panels of **Fig. 4** and **Fig. EV4**. The current **Fig. 4C** was previously **Fig. EV4H**. We have corrected this in the manuscript accordingly. (p.11, l. 458)

7. The destabilization of MTCH1-GFP by USP30 is not very convincing from the data presented in **Figure 4G**. It seems minor at best. How many times was this experiment performed?

In our interpretation, the destabilization of MTCH1-GFP by depletion of USP30 should occur because ubiquitinated MTCH1-GFP, of which a certain pool is normally deubiquitinated by USP30, can now be more efficiently degraded. In accordance with this hypothesis, USP30 depletion led to a 26% lower GFP:RFP ratio (p value = 1.99E-05, n = 4 wells of one experiment). This was the third strongest destabilization effect among all RNAi mediated protein stabilization experiments and can be appreciated in the blue cytometry histogram on the left side of **Fig. 4G**.

We also measured MTCH1-GFP stability using cDNA over expression of MUL1 and USP30 (n = 4 wells of one experiment). While overexpression of the E3 ligase MUL1 led to strong depletion of the GFP signal (**Fig. 4G**, right side; to a similar extent as USP30 siRNA), overexpression of the deubiquitinase increased the median GFP:RFP signal by 7.5%, possibly because a large majority of the overexpressed USP30 is not tethered to the outer mitochondrial membrane and subsequently could not prevent MTCH1-GFP degradation. Importantly, the USP30 overexpression effect did not make the threshold we set for robust changes in protein stability and is not included in **Fig. 4E** showing protein stability changes after cDNA overexpression of an interactor.

We nevertheless included the cytometry data in **Figure 4G** to show our reciprocal validation efforts, using siRNA depletion and cDNA overexpression. We have now highlighted the weak effect of USP30 overexpression by addition of an asterisk in the graph and expanding the caption for **Fig. 4G**. Additionally, we changed the font color of "USP30 cDNA" in the legend of the right cytometry histogram (**Fig. 4G**) to gray to create a distinction to the robust changes observed using USP30 RNAi. (p.29, l. 886)

8. Paragraph 4, page 11 references **Figure 4B**. I believe this should be **4C**.

We thank the reviewer for spotting the embarrassing mix-up. We apologize for the confusion in labeling **Fig. 4B** with **Fig. 4C**. The figure reference has been corrected to **Fig. 4C**. (p.11, l. 481)

9. I would like to suggest that the authors acknowledge in their revised version the importance of complementary protein-protein interaction mapping techniques, such as MaMTH, SIMPL, and other protein complementation assays. These approaches would be valuable in providing a fuller and higher-quality view of the SLC interactome and addressing any limitations of AP-MS in this context.

We thank the reviewer for highlighting the need to include orthogonal technological approaches which enable the characterizations of transient protein interaction of transmembrane proteins. Consequently, we have revised the discussion to include the following section to the discussion:

“Notably, our dataset serves as a starting point for orthogonal protein interaction mapping approaches, such as APEX (Hung et al, 2016), MaMTH (Petschnigg et al, 2014), and SIMPL (Yao et al, 2024), to further validate and complement the SLC-interactome. Complementation/split assays can capture transient interactions which are often missed by AP-MS. A combination of approaches enables to generate a more comprehensive PPI-landscape of SLCs.” (p.17, l. 736-741)

Response to reviewer #3:

In this paper, the authors tackle one of the most critical and challenging issues in life sciences. Despite the solute carrier (SLC) family's significant role in numerous vital physiological functions, much about their functions remains unknown. Naturally, our understanding of their protein-protein interactions (PPIs) is also insufficient. This study aims to comprehensively elucidate the PPI network of the SLC family by overexpressing 396 types of SLC transporters and analyzing their PPIs using mass spectrometry-based proteomics and machine learning (ML). Some PPIs were further evaluated using RNA interference, successfully detecting interactions primarily related to protein stability and localization. The methodology of conducting numerous wet-lab experiments and integrating large datasets with ML is innovative. The resulting interactome is expected to benefit the research community, and some of the individual validation experiments also yielded intriguing results. On the other hand, there are limitations inherent to the wet experiments, some of which the authors have mentioned in their discussion. However, the most significant limitation—the use of an overexpression system for SLC molecules—has not been explicitly addressed. Additionally, the use of cultured cells may not fully reflect the physiological PPIs of SLCs *in vivo*. While these points do not detract from the study's value, the authors might consider addressing them more explicitly.

We thank the reviewer for the detailed and overall positive assessment of our manuscript. We particularly appreciate the recognition of the potential impact that the presented SLC-interactome could have for the research community and agree with the reviewer.

We acknowledge the reviewers' concern regarding the overexpression of bait proteins in the HEK293 cellular system. We agree with the reviewers' statement that these limitations are inherent to AP-MS and that they should be discussed. Given that SLCs are a heterogeneous family regarding their expression across different cell types, tissues and organelles, a systematic mapping approach employing AP-MS will capture in addition to physiologically relevant interactions also interactions which might not be relevant for the transporter's activity. For a detailed explanation of the changes made to address these points, we refer to our answer for the comment of reviewer #3's major point 1.

Major Points:

1. As mentioned above, the experiments were conducted using an overexpression system in cultured cells, raising concerns that the detected interactions may include those occurring outside the transporters' native functional contexts. Indeed, many of the PPIs individually analyzed by the authors are related to localization.

We appreciate the thoughts and suggestions regarding the limitation of the overexpression system and that we should explicitly state the cellular system as limitation to the already listed technological limitations in the manuscript.

A large part of the RESOLUTE project focused on systematically investigating the SLC superfamily using multiple “omics” approaches. Several techniques, including high-throughput imaging, metabolomics (+/- doxycycline induction), and transcriptomics (+/- doxycycline induction, see accompanying manuscript by (Wiedmer *et al*, 2024)), used the same cellular system as the one in this study. One of our objectives was to use a consistent cellular background to facilitate the integration of different data layers (see

accompanying manuscript by (Goldmann et al, 2024)). This is also discussed in the manuscript and provides the rationale for choosing HEK293 cells. (p. 2, l. 70-75)

As mentioned by the reviewer, AP-MS inherently relies on the usage of overexpressed affinity-tagged baits. This ectopic expression can lead to false positive hits and may lead to the identification of interactors not directly related to the transporter's primary function (e.g., trafficking, degradation interactions). Additionally, it may fail to capture relevant physiological functional PPIs, as they might not be expressed or only expressed at low levels in the chosen cellular system (e.g., such as the example of SLC7A9 and SLC3A1 discussed by the reviewer, or TASL a known interactor of SLC15A4 which is absent in HEK293 cells).

However, this is inherent to the methodological approach, and to our knowledge, AP-MS as shown by numerous studies over the decades remains one of the most specific and sensitive MS-based approach to map PPIs, capturing functionally relevant protein interactions (Huttlin et al, 2021; Uliana et al, 2023; Salokas et al, 2022). At the beginning of the multiple year-long RESOLUTE project, HEK293 cells were chosen as the model system due to their comprehensive expression of SLCs (n = 246, see **Appendix Fig. S15A**) and their extensive use for the overexpression of recombinant proteins.

Although our study would have benefitted greatly from only working with endogenous proteins, mapping the protein interactome at endogenous levels at this scale was not feasible. High-quality affinity binder reagents, such as antibodies, are not available for many members of the SLC superfamily. As part of the RESOLUTE project, a handful of high-affinity binders were generated and characterized (Gelová et al, 2024), but unfortunately far from reaching a superfamily-wide coverage.

Other orthogonal MS-based interaction mapping techniques such as BioID (Roux et al, 2012) or APEX (Hung et al, 2016), which allow the characterization of proximity interaction partner face the same limitations as AP-MS. For these techniques, the bait protein needs to be fused to protein biotin ligase or an ascorbate peroxidase, respectively, and is often stably overexpressed as well.

Recent improvements in *in vivo* cross-linking mass spectrometry (XL-MS), including cleavable and enrichable cross-linkers, show great potential to enable interactome mapping at the endogenous level (Wheat et al, 2021; Burke et al, 2015; Wang et al, 2022). However, *in vivo* XL-MS currently does not offer the same depth and coverage as AP-MS or proximity interactome approaches. Ongoing developments in instrumentation, fractionation of cross-linked peptides, cross-linker reagents and data analysis workflows indicate great potential for the application of *in vivo* XL-MS to map full protein interactomes in the near future (Müller et al, 2024).

With this being said, XL-MS is limited to endogenously expressed SLCs, preventing comprehensive mapping of the entire superfamily within a single cellular system. Additionally, endogenously low expressed SLCs would be penalized by XL-MS approaches. Similar limitations apply to co-fractionation MS-based approaches, which directly profile protein complexes and enable the deconvolution of protein complexes into PPIs (Bludau et al, 2020; Scott et al, 2017; Havugimana et al, 2012).

We revised the limitation section of our manuscript and added a limitation of the usage of our cellular system and the overexpression of the tagged-SLCs.

“Third, for generating our interactome dataset, we relied on an overexpression system for the tagged SLC baits. Although this is comparable to other PPI-mapping approaches (BioID, APEX) and necessary due to the lack of sufficient affinity binders for SLCs, it may not accurately represent endogenous expression levels and led to identifying PPIs related to overexpression, such as trafficking and folding-associated interactors. Using cultured cells, while beneficial for systematically profiling the entire family in one background, may not fully capture the complexity and physiological relevance of PPIs of SLCs not endogenously expressed. Several well-characterized interactions of SLCs are tissue/cell line-specific, such as TASL-SLC15A4 (Heinz et al, 2020) or ACE2-SLC6A19 (Danilczyk et al, 2006), which are missed with our approach.”(p16,17, l.713-721)

2. The title of the paper is ambitious, reflecting the authors' aim to obtain 'The solute carrier superfamily interactome.' While this study makes significant strides in this direction, it may not yet fully capture the comprehensive map implied by the title. A more specific title might better reflect the current scope of the work.

We thank the reviewer for the suggestion regarding the changing of the title of our manuscript. We dare to disagree. According to the logic of proteomics, an interactome can never be complete, as it is dynamic and tissue-, time-dependent. Nevertheless, it is appropriate to define a study that is more than 100 times larger than any previous effort and represents almost a decade of work, with six years of RESOLUTE project duration and many preparatory before, done with great care, as what it is. Everything else would be apologetic.

3. How would the interactome data differ if interactions were categorized based on whether they occur at the locations where SLC molecules naturally perform their transport functions or elsewhere? Although the experimental system may have inadvertently captured PPIs across different stages of an SLC transporter's lifecycle, this is not necessarily a drawback. However, from a biological perspective, it might be beneficial to explicitly distinguish between these differences. Accordingly, it could be worth considering structuring the manuscript to address the interactome with attention to these temporal scales.

We thank the reviewer for an intriguing thought. During the conceptualization and writing of the manuscript, we extensively debated organizing our study according to the reviewer's suggested structure. However, in this process, we identified certain inherent limitations to the proposed restructuring. Following, we will explain the impediment to further dissect the interactome according to a temporal and spatial organization.

- l) The localization of many SLCs is ambiguous. While most have reported localizations, many SLCs are found in multiple subcellular locations according to literature and databases such as human protein atlas (Meixner et al, 2020; Goldmann et al, 2024; Uhlén et al, 2015). In the RESOLUTE project, the same HEK293 WT overexpression cell lines used for AP-MS were also used to determine the subcellular locations of SLCs by IF. The data showed that most SLCs were confidently found in multiple locations. The IF-dataset overlapped with other subcellular location annotations (Human Protein Atlas, GO subcellular location, and UniProtKB) (see Goldmann et al. 2024, **Fig. 3A-H**), but overall, there were huge differences

- regarding SLC localization across all the resources. Computational spatial deconvolution of the PPI-network would thus be alone complicated by multiple localizations of SLCs, and the nonexistence of a ground truth for many of them.
- II) Within the interactome, protein interactions and complexes are convoluted. One way of PPI-network deconvolution is to integrate database knowledge. However, this would mean retrieving and trusting localization information for the 4,381 preys and 396 SLCs. Inconsistencies in localization annotations in online resources and multiple localizations per protein further contribute to the complexity of this task. We used GO subcellular compartment enrichment to infer the most likely SLC localization using prey information (**Fig. 6A-D**). This inference is crude, and fully deconvoluting the SLC-interactome by spatial organization would require extensive additional experimental data. A potential approach could involve mapping the subcellular location of proteins experimentally before the interactome study through subcellular fractionation, IPs of intact organelles, or combinations (Hein et al, 2024; Itzhak et al, 2016). We deemed this out of scope of the current study.
 - III) Similarly, the temporal aspect is convoluted within our dataset. We attempted to deconvolute it (see **Fig. 4, Fig. 5**) for several well-characterized PPIs. We used annotations of proteins involved in degradation and folding to better understand the organization of the SLC-interactome, discussing only proteins with literature or database evidence of association with folding, glycosylation, or degradation. In an ongoing study, we are focusing on several interactors of the SLC-interactome to experimentally investigate temporal and cell line specific aspects to elucidate rewiring of PPI-networks.
 - IV) For this study, which maps PPIs at steady-state (except for the validation experiments of SLC16A6 variants), we considered it out of scope to include differential mapping after cell line perturbations.

In summary, the chosen approach limits the temporal and spatial interpretation of the data. We anyhow still structured the paper around proteostasis and trafficking themes, as these were most prominently present in the dataset, thus deconvoluting part of the temporal and spatial PPI-networks. We extensively highlight our findings regarding folding/degradation interactions (**Fig. 4, Fig. 5, Fig. EV4, Appendix Fig. S11**) and trafficking (**Fig. 6, Fig. EV5**). We carefully selected proteins with previously demonstrated functional associations, linking already reported biology to novel biological findings for SLCs.

4. The PPI examples used to validate the interactome's reliability in Fig. EV1 and Fig. 2 are all located on the plasma membrane. While these are valid examples, they may not fully align with the paper's focus on novel findings related to trafficking and localization. It might strengthen the narrative to include examples more directly tied to these findings.

We thank the reviewer for pointing this out. We have addressed the raised point by incorporating new data into the manuscript, leading to an extensive re-organization of the protein complex prediction section and **Fig. EV1**. As the reviewer pointed out, the presented examples of predicted protein complexes are valid and partially well-studied protein interactions. The original purpose was to choose well-characterized PPIs and use predicted structures, to validate *in silico* validate novel interactions of the same interactor with other SLCs. We adapted the following strategy to address the reviewer's point:

We kept the BSG and EMB sections in manuscript and in **Fig. 2I-J** to demonstrate the validity our SLC-interactome by orthogonal structural modelling. Meanwhile, we moved the SLC3A2 and CHIP1 sections to the Appendix.

We chose all SLC interactions with GALNT2 as new examples. We predicted an additional 21 SLC-GALNT2 complexes. We also chose 21 randomly selected controls sampled from the lowest 25% quantile of GALNT2 signal characterized as background in AP-MS. We chose this example, as it is localized in the Golgi and vesicles (UniProtKB: Golgi/ Golgi stacks (UniProt Consortium, 2023), Human Protein Atlas: localized to the Golgi apparatus (Uhlén et al, 2015)) and it is involved in post-translational processing by performing the initial reaction in O-linked oligosaccharide. So far only 3 SLC-GALNT2 interactions (with SLC25A4, NIPAL1, NPC1) were reported in PPI-databases.

Major point 4 overlaps with minor points 4, 5, 6, 7 of reviewer #3, which we all addressed too. We adapted the following changes: Moved **Fig. EV1F-I** to **Appendix Fig. S4** and **S5**. Restructured the two Appendix Figures. Added new data and corresponding figure panels to **Fig. EV1F-H**. Added figure captions to all figures. Added a new Appendix Result section “Structural modeling of SLC complexes with SLC3A2 and CHP1”, by moving the corrected parts from the main manuscript to the appendix. Summarizing the SLC3A2 and CHP1 part in the main manuscript:

“To provide further benchmarking examples, we obtained structural models from SLC3A2 interactions identified within our dataset. SLC3A2 forms heterodimeric complexes with multiple members of the L-type amino acid transporters (LATs), which belong to the SLC7-family (Rodriguez et al, 2021; Lee et al, 2022; Oda et al, 2020; Mastroberardino et al, 1998; Estévez et al, 1998; Kandasamy et al, 2018). The obtained predicted models showed a high interaction confidence, and significant structural scores compared to non-reported interactions (see details Appendix Results, **Dataset EV3, Appendix Fig. S4A-D**). Further we modelled the interactions of SLCs with the Calcineurin B homologous protein 1 (CHP1). CHP1 was reported to form heterodimeric complexes with SLC9A1 and SLC9A3, stabilizing their plasma membrane localization and increasing pH sensitivity (Dong et al, 2021). We scored interactions with additional SLC9-, SLC6-, and SLC7-family members (see details Appendix Results, **Dataset EV3, Appendix Fig. S5A-E**).” (p. 7, l. 283-293)

Changed manuscript text for SLC-GALNT2 protein complex models:

“We also evaluated novel SLC interactions with GALNT2 (polypeptide N-acetylgalactosaminyltransferase 2), which was so far only reported to interact with three SLCs (NPC1, NIPAL1, SLC25A4; see reference PPI-library). GALNT2, expressed ubiquitously and localized to the cis- and trans-Golgi apparatus, where it catalyzes the initial steps of the O-linked glycosylation of proteins (Kurze et al, 2019). O-linked glycosylation was found to affect protein stability of receptors, such as EGFR (Wu et al, 2022). GALNT2 was scored as interactor for 21 SLCs (**Fig. EV1F**), of which eleven were metal transporters of the SLC30 and SLC39 families. (**Fig. EV1G upper part**). We obtained 14 high-confidence and seven medium confidence structures of the SLC-GALNT2 complexes (Fig. EV1G lower part). We compared the structures of these heterodimers against a set of non-reported SLC-GALNT2 complexes, randomly sampled from the lowest 25% quantile of GALNT2 abundances quantified within our dataset. The structures of scored interactions showed a significantly higher pDockQ score (unpaired student t-test, p-value = 0.00695, **Fig.**

EV1G right side). Lastly, we investigated one of the high-scoring models, the complex of SLC30A1 with GALNT2 (**Fig. EV1H**). SLC30A1 is a Zn²⁺/Ca²⁺ exchanger localized at the plasma membrane, and forms a homodimer (Sun et al, 2024). The predicted interaction interface of GALNT2 is largely distinct from the homodimer interaction interface. SLC30A1 was reported to be N-glycosylated at position 299 (Asn-299) in the extracellular loop between transmembrane domain affecting its stability (Nishito & Kambe, 2019). Profiling of O-linked glycosylated proteins identified two O-linked sites on SLC30A1, one being either S300 or T301 (Steentoft et al, 2013), again at the extracellular loop, which is predicted to interact with GALNT2.” (p.7/8, l. 294-313)

Minor Points:

1. While 405 molecules were tested, were there any molecules excluded due to issues such as lack of expression, like SLC9A1? It would be helpful if the manuscript addressed this point or provided any information to the relevant section if it is already covered.

We are glad to be able to address the point raised by the reviewer. Yes, there were additional SLC-baits which were filtered by different QC steps prior to assembly of the dataset of 405 SLC baits. We report all SLC baits, with annotations (e.g.: alternative identifiers, tagged terminus) and the QC status in **Table EV1**.

Our QC procedure contained several steps: Briefly, we first evaluated expression of the SLC baits using a full lysate extract by western blotting. We cross-referenced this with the results obtained from the high-throughput imaging of the same cell line (Goldmann et al, 2024). In total we considered 686 cell lines (see **Table EV1**) expressing one of the 447 SLCs plus the two different GFP cell lines. By default, we tried first the C-terminal tagged version and generated the N-terminal construct if we observed low expression. Next, we performed the AP and subsequently used an aliquot of ~2% of the eluate to assess the SLC's enrichment by western blotting. SLC baits with low performance were rejected from being further processed. Lastly, after the MS-acquisition we removed bait-proteins with low signal intensity (< 24 spectral counts) and samples with technical issues (e.g. low reproducibility were rejected and/or repeated). Overall, for AP-MS we tried 526 cell lines expressing 439 SLC baits, for which 119 were rejected either due to low expression and/or a better performing cell line tagged on the N-terminus

For SLC9A1 we excluded a C-terminal tagged cell line due to low expression and re-generated an C- and N-terminal tagged cell line which were both used for AP-MS, but both were removed prior to the final dataset assembly due to low signal. We added to the revised manuscript a summary of the cell line assessment to the methods section described above (p. 36, l. 1016-1025). This should help readers appreciate the challenges of expressing and purifying SLCs and may guide future studies. We further added a cross reference to **Table EV1** and the method section when discussing the low expression of SLC9A1 in the appendix results. (Appendix p. 3, l. 100-101)

2. There appears to be inconsistency in the labeling of molecules with SLC names in the text and figures. Ensuring uniform nomenclature would help avoid confusion.

We thank the reviewer for recognizing these inconsistencies. We checked the manuscript, figures, extended view figures and appendix figures and revised the following inconsistencies.

- Changed multiple instances of SLC16A to SLC16 family. (p.3, l. 101, p.7, l.268, l.276)
- Revised multiple instances of SLC7A family to SLC7 family. (p.3, l 102)
- Revised SLC39A to SLC39 family (p. 3., l. 104)
- Changed multiple instances of SLC9A, SLC6A and SLC7A family notations to SLC9, SLC6 and SLC7 family to be consistent with the family notations. (Appendix, p.3, l. 82-105)
- Changed multiple instances of SLC12A and SLC6A family notations to SLC12 and SLC6 families as used in Fig. EV3A and Fig. EV3C (p.9, l.362; l.378; p.10, l.401)
- Removed in the manuscript the alternative name for SLC22A11 (p. 14, l.581).
- Removed in the manuscript the alternative name for SLC31A1 (p.14, l.588).
- Removed in the manuscript the alternative name for SLC43A2 (p.14, l.594).

We hope these changes increase readability and apologize for the inconsistencies.

3. Regarding the example of interactions between MCTs and BSG or EMB, as well as well-known ACE2 and Collectrin with SLC6A19, should there not be a mention of tissue specificity in these cases?

We thank the reviewer for this remark. We have expanded the description of the chaperones BSG/EMB to include the tissue-specific expression of SLC16 family members and chaperones respectively.

“Several SLC16 family members, such as SLC16A1, SLC16A3, and SLC16A7, are widely expressed, while others exhibit tissue-specific expression patterns. For instance, SLC16A8 is specifically expressed in the retinal pigment epithelium (Philp et al., 2003). The chaperone BSG is ubiquitously expressed, whereas EMB expression is more restricted (Guenette et al, 1997).” (p.7, l. 270 – 273)

We included the tissue-specific expression of ACE2-SLC6A19 in the limitation section of the discussion and addressed it in our response to reviewer #3 major point 1. (Page 17, l.720-721)

4. p.7, l. 13: CATs are not known to form heterodimers, so they should not be mentioned here.

We thank the reviewer for recognizing this mistake. We revised the sentence and shortened the paragraph in the manuscript and moved the SLC3A2 and CHIP1 sections to the Appendix.

“To provide further benchmarking examples, we obtained structural models from SLC3A2 interactions identified within our dataset. SLC3A2 forms heterodimeric complexes with multiple members of the L-type amino acid transporters (LATs), which belong to the SLC7-family (Rodriguez et al, 2021; Lee et al, 2022; Oda et al, 2020; Mastroberardino et al, 1998; Estévez et al, 1998; Kandasamy et al, 2018). (p.7, l. 283-286)

5. p.7, l. 17: The physiological SLC7A9 binding partner is SLC3A1, not SLC3A2.

We thank the reviewer for pointing out this error. The incorrect statement with SLC7A9 has been removed from the manuscript.

6. **Fig. EV1G:** Showing predicted structures for multiple SLC7 family members and SLCs seems less meaningful. Moreover, the SLC7A7-SLC3A2 complex, while lacking structural analysis, is

biochemically well-supported. Highlighting the superiority of the interactome with other examples might be more effective.

We agree with the reviewer. Our intention was to use well-characterized interactions for the structural modelling to support the benchmarking of our dataset. We moved the SLC3A2 complex predictions to the Appendix and the figure panels to the Appendix figures. We added as a new example predicted structures of SLC-GALNT2 protein complexes (see reviewer #3 major point 4). (p.7, l. 283-293, and appendix p.3, l. 82-93)

7. Similarly, the CryoEM structure of SLC9A1 and CHP1 makes the importance of predicting the structure of SLC9A2 and CHP1 less clear.

We agree with the reviewer's remark. To address this point, we moved the CHP1 section to the Appendix. As a new example predicted complexes of GALNT2-SLC were added to **Fig. EV2F-G** (see reviewer #3 major point 4). (p.7, l. 283-293, and Appendix p.3, l. 94-105)

8. p. 11, l. 30: The text mentions GFP-SLC16A6, but was the GFP fusion at the N-terminus? Fig. 4A suggests a C-terminal fusion.

The reviewer is correct in pointing out that GFP was fused to the N-terminus of SLC16A6. The GFP tag was put at the N-terminus of 4 other SLCs when C-terminal tagging was not successful. In the case of SLC16A6, preliminary results showed slightly stronger plasma membrane localization of N-terminally tagged SLC16A6 compared to the C-terminally fused protein and was therefore selected for further analyses. We however suspect that the terminal tag position does not have strong effects on the observed behavior of SLC16A6 since all SLC16A6 proteomics data shown in our manuscript, which served as the basis for the validation experiments, was generated using the C-terminally fused protein.

Since 61 SLCs were tagged with GFP at their C-terminus, we decided to simplify the experimental validation outline in **Fig. 4A** in this respect and included the tagging location in **Table EV4** and the precise SLC-GFP and GFP-SLC protein sequence in our web resource (<https://re-solute.eu/resources/reagents>). We realized however that the tagging strategy could be mentioned more explicitly in our manuscript. Because **Fig. 4A** is in our opinion already quite complex, we amended the main text as follows:

"We selected 65 SLCs with interactions involved in these processes and expressed them with an N- or C-terminal GFP tag from an inducible promoter (**Fig. 4A**)."

 (p.10, l. 414-415)

We also clarified the tagging strategy in the Materials and Methods section as follows:

"The GFP tag was located at the C-terminus except for SLC6A2, SLC6A3, SLC9B2, and SLC30A4 where N-terminal tagging did not result in GFP positive cells, and for SLC16A6 (wild type and mutants) where the N-terminal GFP fusion protein showed slightly stronger plasma membrane localization. In these cases, the RFP reporter was expressed upstream of an IRES sequence followed by the GFP-SLC fusion." (p.35, l. 985-988)

9. In some cases, citations, such as those related to complex formation, reference recent structural analyses, even though these facts had been previously established through biochemical and

other methods. It might be helpful to acknowledge earlier studies alongside the recent work to provide a more complete context.

We thank the reviewer for this point and added references for the discussed protein complexes of the SLC7 and SLC16 family members. For SLC7-SLC3A2 interactions we added the following references: (Rodriguez et al, 2021; Lee et al, 2022; Oda et al, 2020; Mastroberardino et al, 1998; Estévez et al, 1998; Kandasamy et al, 2018). (p.7, l. 283-286)

For the complex of BSG/EMB together with SLC16 family members we added the following references: (Bosshart et al, 2021; Wilson et al, 2005; Poole & Halestrap, 1997; Guenette et al, 1997; Philp et al, 2003). (p.7, l. 267-272)

10. The inclusion of SciWheel.com links in the references might impact readability and make it more challenging to search for citations during the review process. Removing these links could enhance accessibility and improve the overall review experience.

We thank the reviewer for the feedback. We removed the SciWheel.com links to improve readability and accessibility for revisions.

Revised figure panels:

- Revised **Figure 1A** to highlight better the interconnectivity of the four back-to-back submitted studies.
- Revised **Fig. 1E** panel category variables.
- **Revised Appendix Fig. S1D** to increase readability of the plot (same x- and y-axis scale and ticks; line through the origin; axis labels).
- Major revision and new data and analysis for **Fig. EV1**:
 - New panels **Fig. EV1F-H**
 - Moved previous panels **Fig.EV1F,G** to new **Appendix Fig. S4**
 - Moved previous panels **Fig. EV1H,I** to new **Appendix Fig. S5**
 - New **Appendix Fig. S6** (SLC30A1-GALNT2)
- Revised **Appendix Fig. S8** (now **Appendix Fig. S10**)
- To comply with journal policy of maximum five Extended View Figures, Fig.EV5 was moved to the Appendix Figures.
- Due to the revision, the number of figures in the Appendix increased and were adapted accordingly.

References

Bludau I, Heusel M, Frank M, Rosenberger G, Hafen R, Banaei-Esfahani A, van Drogen A, Collins BC, Gstaiger M & Aebersold R (2020) Complex-centric proteome profiling by SEC-SWATH-MS for the parallel detection of hundreds of protein complexes. *Nat Protoc* 15: 2341–2386

Bosshart PD, Charles R-P, Garib Singh R-AA, Schlessinger A & Fotiadis D (2021) SLC16 family: from atomic structure to human disease. *Trends Biochem Sci* 46: 28–40

Buljan M, Ciuffa R, van Drogen A, Vichalkovski A, Mehnert M, Rosenberger G, Lee S, Varjosalo M, Pernas LE, Spegg V, *et al* (2020) Kinase interaction network expands functional and disease roles of human kinases. *Mol Cell* 79: 504-520.e9

Burke AM, Kandur W, Novitsky EJ, Kaake RM, Yu C, Kao A, Vellucci D, Huang L & Rychnovsky SD (2015) Synthesis of two new enrichable and MS-cleavable cross-linkers to define protein-protein interactions by mass spectrometry. *Org Biomol Chem* 13: 5030–5037

Danilczyk U, Sarao R, Remy C, Benabbas C, Stange G, Richter A, Arya S, Pospisilik JA, Singer D, Camargo SMR, *et al* (2006) Essential role for collectrin in renal amino acid transport. *Nature* 444: 1088–1091

Dong Y, Gao Y, Ilie A, Kim D, Boucher A, Li B, Zhang XC, Orlowski J & Zhao Y (2021) Structure and mechanism of the human NHE1-CHP1 complex. *Nat Commun* 12: 3474

Estévez R, Camps M, Rojas AM, Testar X, Devés R, Hediger MA, Zorzano A & Palacín M (1998) The amino acid transport system y⁺L/4F2hc is a heteromultimeric complex. *FASEB J* 12: 1319–1329

Frezza C, Cipolat S & Scorrano L (2007) Organelle isolation: functional mitochondria from mouse liver, muscle and cultured fibroblasts. *Nat Protoc* 2: 287–295

Gelová Z, Ingles-Prieto A, Bohstedt T, Frommelt F, Chi G, Chang Y-N, Garcia J, Wolf G, Azzollini L, Tremolada S, *et al* (2024) Protein binder toolbox for studies of solute carrier transporters. *J Mol Biol* 436: 168665

Goldmann U, Wiedmer T, Garofoli A, Sedlyarov V, Bichler M, Wolf G, Christodoulaki E, Ingles-Prieto A, Ferrada E, Frommelt F, *et al* (2024) Data- and knowledge-derived functional landscape of human solute carriers. *BioRxiv*

Guenette RS, Sridhar S, Herley M, Mooibroek M, Wong P & Tenniswood M (1997) Embigin, a developmentally expressed member of the immunoglobulin super family, is also expressed during regression of prostate and mammary gland. *Developmental Genetics*

Havugimana PC, Hart GT, Nepusz T, Yang H, Turinsky AL, Li Z, Wang PI, Boutz DR, Fong V, Phanse S, *et al* (2012) A census of human soluble protein complexes. *Cell* 150: 1068–1081

Hein MY, Peng D, Todorova V, McCarthy F, Kim K, Liu C, Savy L, Januel C, Baltazar-Nunez R, Sekhar M, *et al* (2024) Global organelle profiling reveals subcellular localization and remodeling at proteome scale. *Cell*

Heinz LX, Lee J, Kapoor U, Kartnig F, Sedlyarov V, Papakostas K, César-Razquin A, Essletzbichler P, Goldmann U, Stefanovic A, *et al* (2020) TASL is the SLC15A4-associated adaptor for IRF5 activation by TLR7-9. *Nature* 581: 316–322

Hung V, Udeshi ND, Lam SS, Loh KH, Cox KJ, Pedram K, Carr SA & Ting AY (2016) Spatially resolved proteomic mapping in living cells with the engineered peroxidase APEX2. *Nat Protoc* 11: 456–475

- Huttlin EL, Bruckner RJ, Navarrete-Perea J, Cannon JR, Baltier K, Gebreab F, Gygi MP, Thornock A, Zarraga G, Tam S, *et al* (2021) Dual proteome-scale networks reveal cell-specific remodeling of the human interactome. *Cell* 184: 3022-3040.e28
- Huttlin EL, Bruckner RJ, Paulo JA, Cannon JR, Ting L, Baltier K, Colby G, Gebreab F, Gygi MP, Parzen H, *et al* (2017) Architecture of the human interactome defines protein communities and disease networks. *Nature* 545: 505–509
- Huttlin EL, Ting L, Bruckner RJ, Gebreab F, Gygi MP, Szpyt J, Tam S, Zarraga G, Colby G, Baltier K, *et al* (2015) The bioplex network: A systematic exploration of the human interactome. *Cell* 162: 425–440
- Itzhak DN, Tyanova S, Cox J & Borner GH (2016) Global, quantitative and dynamic mapping of protein subcellular localization. *eLife* 5
- Kandasamy P, Gyimesi G, Kanai Y & Hediger MA (2018) Amino acid transporters revisited: New views in health and disease. *Trends Biochem Sci* 43: 752–789
- Kurze A-K, Buhs S, Eggert D, Oliveira-Ferrer L, Müller V, Niendorf A, Wagener C & Nollau P (2019) Immature O-glycans recognized by the macrophage glycoreceptor CLEC10A (MGL) are induced by 4-hydroxy-tamoxifen, oxidative stress and DNA-damage in breast cancer cells. *Cell Commun Signal* 17: 107
- Lee Y, Wiriyasermkul P, Kongpracha P, Moriyama S, Mills DJ, Kühlbrandt W & Nagamori S (2022) Ca²⁺-mediated higher-order assembly of heterodimers in amino acid transport system b₀,+ biogenesis and cystinuria. *Nat Commun* 13: 2708
- Luck K, Kim D-K, Lambourne L, Spirohn K, Begg BE, Bian W, Brignall R, Cafarelli T, Campos-Laborie FJ, Charloreaux B, *et al* (2020) A reference map of the human binary protein interactome. *Nature* 580: 402–408
- Mastroberardino L, Spindler B, Pfeiffer R, Skelly PJ, Loffing J, Shoemaker CB & Verrey F (1998) Amino-acid transport by heterodimers of 4F2hc/CD98 and members of a permease family. *Nature* 395: 288–291
- Meixner E, Goldmann U, Sedlyarov V, Scorzoni S, Rebsamen M, Girardi E & Superti-Furga G (2020) A substrate-based ontology for human solute carriers. *Mol Syst Biol* 16: e9652
- Mellacheruvu D, Wright Z, Couzens AL, Lambert J-P, St-Denis NA, Li T, Miteva YV, Hauri S, Sardiou ME, Low TY, *et al* (2013) The CRAPome: a contaminant repository for affinity purification-mass spectrometry data. *Nat Methods* 10: 730–736
- Müller F, Stejskal K & Mechtler K (2024) Breaking Barriers in Crosslinking Mass Spectrometry: Enhanced Throughput and Sensitivity with the Orbitrap Astral Mass Analyzer. *BioRxiv*
- Nishito Y & Kambe T (2019) Zinc transporter 1 (ZNT1) expression on the cell surface is elaborately controlled by cellular zinc levels. *J Biol Chem* 294: 15686–15697
- Oda K, Lee Y, Wiriyasermkul P, Tanaka Y, Takemoto M, Yamashita K, Nagamori S, Nishizawa T & Nureki O (2020) Consensus mutagenesis approach improves the thermal stability of system xc⁻ transporter, xCT, and enables cryo-EM analyses. *Protein Sci* 29: 2398–2407

- Petschnigg J, Groisman B, Kotlyar M, Taipale M, Zheng Y, Kurat CF, Sayad A, Sierra JR, Mattiazzi Usaj M, Snider J, *et al* (2014) The mammalian-membrane two-hybrid assay (MaMTH) for probing membrane-protein interactions in human cells. *Nat Methods* 11: 585–592
- Philp NJ, Wang D, Yoon H & Hjelmeland LM (2003) Polarized expression of monocarboxylate transporters in human retinal pigment epithelium and ARPE-19 cells. *Invest Ophthalmol Vis Sci* 44: 1716–1721
- Poole RC & Halestrap AP (1997) Interaction of the erythrocyte lactate transporter (monocarboxylate transporter 1) with an integral 70-kDa membrane glycoprotein of the immunoglobulin superfamily. *J Biol Chem* 272: 14624–14628
- Rodriguez CF, Escudero-Bravo P, Díaz L, Bartoccioni P, García-Martín C, Gilabert JG, Boskovic J, Guallar V, Errasti-Murugarren E, Llorca O, *et al* (2021) Structural basis for substrate specificity of heteromeric transporters of neutral amino acids. *Proc Natl Acad Sci USA* 118
- Roux KJ, Kim DI, Raida M & Burke B (2012) A promiscuous biotin ligase fusion protein identifies proximal and interacting proteins in mammalian cells. *J Cell Biol* 196: 801–810
- Salokas K, Liu X, Öhman T, Chowdhury I, Gawryski L, Keskitalo S & Varjosalo M (2022) Physical and functional interactome atlas of human receptor tyrosine kinases. *EMBO Rep* 23: e54041
- Scott NE, Rogers LD, Prudova A, Brown NF, Fortelny N, Overall CM & Foster LJ (2017) Interactome disassembly during apoptosis occurs independent of caspase cleavage. *Mol Syst Biol* 13: 906
- Steenfot C, Vakhrushev SY, Joshi HJ, Kong Y, Vester-Christensen MB, Schjoldager KT-BG, Lavrsen K, Dabelsteen S, Pedersen NB, Marcos-Silva L, *et al* (2013) Precision mapping of the human O-GalNAc glycoproteome through SimpleCell technology. *EMBO J* 32: 1478–1488
- Sun C, He B, Gao Y, Wang X, Liu X & Sun L (2024) Structural insights into the calcium-coupled zinc export of human ZnT1. *Sci Adv* 10: eadk5128
- Uhlén M, Fagerberg L, Hallström BM, Lindskog C, Oksvold P, Mardinoglu A, Sivertsson Å, Kampf C, Sjöstedt E, Asplund A, *et al* (2015) Tissue-based map of the human proteome. *Science* 347: 1260419
- Uliana F, Ciuffa R, Mishra R, Fossati A, Frommelt F, Keller S, Mehnert M, Birkeland ES, van Droogen F, Srejjic N, *et al* (2023) Phosphorylation-linked complex profiling identifies assemblies required for Hippo signal integration. *Mol Syst Biol* 19: e11024
- UniProt Consortium (2023) Uniprot: the universal protein knowledgebase in 2023. *Nucleic Acids Res* 51: D523–D531
- Wang Y, Hu Y, Höti N, Huang L & Zhang H (2022) Characterization of In Vivo Protein Complexes via Chemical Cross-Linking and Mass Spectrometry. *Anal Chem* 94: 1537–1542
- Wheat A, Yu C, Wang X, Burke AM, Chemmama IE, Kaake RM, Baker P, Rychnovsky SD, Yang J & Huang L (2021) Protein interaction landscapes revealed by advanced in vivo cross-linking-mass spectrometry. *Proc Natl Acad Sci USA* 118

Wiedmer T, Teoh ST, Christodoulaki E, Wolf G, Tian C, Sedlyarov V, Jarret A, Leippe P, Frommelt F, Ingles-Prieto A, *et al* (2024) Metabolic mapping of the human solute carrier superfamily. *BioRxiv*

Wilson MC, Meredith D, Fox JEM, Manoharan C, Davies AJ & Halestrap AP (2005) Basigin (CD147) is the target for organomercurial inhibition of monocarboxylate transporter isoforms 1 and 4: the ancillary protein for the insensitive MCT2 is EMBIGIN (gp70). *J Biol Chem* 280: 27213–27221

Wu G, Xu G, Schulman BA, Jeffrey PD, Harper JW & Pavletich NP (2003) Structure of a beta-TrCP1-Skp1-beta-catenin complex: destruction motif binding and lysine specificity of the SCF(beta-TrCP1) ubiquitin ligase. *Mol Cell* 11: 1445–1456

Wu L, Cheng Y, Geng D, Fan Z, Lin B, Zhu Q, Li J, Qin W & Yi W (2022) O-GlcNAcylation regulates epidermal growth factor receptor intracellular trafficking and signaling. *Proc Natl Acad Sci USA* 119: e2107453119

Yao Z, Kim J, Geng B, Chen J, Wong V, Lyakisheva A, Snider J, Dimlić MR, Raić S & Stajljarić I (2024) A split intein and split luciferase-coupled system for detecting protein-protein interactions. *Mol Syst Biol*

Zhao B, Li L, Tumaneng K, Wang C-Y & Guan K-L (2010) A coordinated phosphorylation by Lats and CK1 regulates YAP stability through SCF(beta-TRCP). *Genes Dev* 24: 72–85

24th Feb 2025

Manuscript Number: MSB-2024-12688R

Title: The solute carrier superfamily interactome

Dear Dr. Superti-Furga,

Thank you for the submission of your revised manuscript to Molecular Systems Biology. I am pleased to inform you that we will be able to accept your manuscript pending the following final amendments and appropriate response to reviewers:

- 1) Please ensure that we have a functional email address for all authors. Currently the address for Jasper F. Ooms (j.f.ooms@umail.leidenuniv.nl) bounces back emails.
- 2) Please note that Reviewers 1 and 3 have some remaining concerns that should be addressed textually.
- 3) Please upload the manuscript in .docx (or LaTeX) format with no track changes and no figures, only the figure legends should remain placed below the References.
- 4) Please format the Data availability section according to the example below:
"The datasets and computer code produced in this study are available in the following databases:
- Chip-Seq data: Gene Expression Omnibus GSE46748 (<https://www.ncbi.nlm.nih.gov/geo/query/acc.cgi?acc=GSE46748>)
- Modeling computer scripts: GitHub (<https://github.com/SysBioChalmers/GECKO/releases/tag/v1.0>)
- [data type]: [full name of the resource] [accession number/identifier] ([doi or URL or identifiers.org/DATABASE:ACCESSION])"
- 5) Data availability: The IM-30161 dataset in the IntAct database does not appear to be available, this needs to be rectified. Please ensure that all other deposited datasets are now publicly accessible, that reviewer access information is removed, and that specific URLs for each accession code are given as in the example above.
- 6) Please rename "Conflict of Interest" to "Disclosure and competing interests statement". We updated our journal's competing interests policy in January 2022 and request authors to consider both actual and perceived competing interests. Please review the policy <https://www.embopress.org/competing-interests> and update your competing interests if necessary.
- 7) Our journal encourages inclusion of *data citations in the reference list* to directly cite datasets that were re-used and obtained from public databases. Data citations in the article text are distinct from normal bibliographical citations and should directly link to the database records from which the data can be accessed. In the main text, data citations are formatted as follows: "Data ref: Smith et al, 2001" or "Data ref: NCBI Sequence Read Archive PRJNA342805, 2017". In the Reference list, data citations must be labeled with "[DATASET]". A data reference must provide the database name, accession number/identifiers and a resolvable link to the landing page from which the data can be accessed at the end of the reference. Further instructions are available at .
- 8) In the Methods, please take care of the following:
 - The Materials and Methods section should be renamed to "Methods".
 - Cell lines: As you have indicated in the Author Checklist, please also be sure to include a sentence in the Methods as to whether or not the cell lines were recently authenticated and tested for mycoplasma contamination.
 - Primers: please ensure that the actual primer sequences are included in Table EV5, and not links to the primer sequence via IDT.
 - Please ensure that a statement on whether or not blinding was done is included in the Methods even if no blinding was done. Please also be sure to update the Author Checklist with this information and where it can be found in the manuscript.
- 9) Please place individual sections of the manuscript in the following order: Title page - Abstract & Keywords - Introduction - Results - Discussion - Methods - Data Availability - Acknowledgements - Disclosure and Competing Interests Statement - References - Figure Legends - Expanded View Figure Legends.
- 10) Please remove the "Supplementary Information" section from the main manuscript file.
- 11) For the figures and figure legends, please take care of the following:
 - The callout for Appendix Fig. 6A,B should be corrected to Appendix Fig. S6A,B
 - All figure callouts should be listed sequentially.
 - Please note that the exact p values are not provided in the legends of figures 2H, 4D, E; 5B, D; 6A, EV4 F
 - Please indicate the statistical test used for data analysis in the legends of figures EV3 E.
 - Please indicate what */ **/ ***/ **** represents; if this represents p value(s), please indicate the exact p value in the legend(s) of figure(s) 2C, J; EV1 D, G; EV3 H
 - Please note that the box plots need to be defined in terms of minima, maxima and percentile in the legends of figures 2A, C, I, J; EV1 F, G; EV3 H
 - Please note that the box plots need to be defined in terms of minima, maxima, centre, bounds of box and whiskers, and percentile in the legend of figure EV4 F
 - Please note that information related to n is missing in the legends of figures 2A, I; 6C, EV1 F
 - Please note that the error bars are not defined in the legend of figure EV1 D.
 - Please note that the measure of center for the error bars needs to be defined in the legend of figure 6D
- 12) Tables: The labels and legends for Tables EV1-EV8 should not be uploaded as a separate tab/sheet, but above the table in each Excel file.

13) Appendix file: Please upload the Appendix as a single PDF (no separate image files are needed) and add page numbers to the Table of Contents. In addition, please move the Results and Methods currently in the Appendix to the main manuscript file, and include the reagents in the Reagents and Tools table. There are no strict word limits for the main manuscript file and this information should be available to readers in the main manuscript.

14) Source Data: In a routine image check, we have identified a potential issue with Figure EV4C that the blot may have been spliced together. Please provide the source data for this blot. If you make any changes to the figure or manuscript, please provide an explanation.

15) Funding: Please ensure that all funding sources are entered in the Comments box in our submission system are included in the list of funders using the "More Funders" option.

16) Synopsis:

- Synopsis image: We would suggest simplifying the synopsis image, as when resized to the requested 550 pixels wide x 300-600 pixels high, the text is quite small and difficult to read.

17) As part of the EMBO Publications transparent editorial process initiative (see our policy here:

https://www.embopress.org/transparent-process#Review_Process), Molecular Systems Biology will publish online a Peer Review File (PRF) to accompany accepted manuscripts. This file will be published in conjunction with your paper and will include the anonymous referee reports, your point-by-point response and all pertinent correspondence relating to the manuscript. Let us know whether you agree with the publication of the PRF and as here, if you want to remove or not any figures from it prior to publication. Please note that the Authors checklist will be published at the end of the PRF.

18) After your paper is published, we will promote it on social media. If you have any handles or hashtags for Bluesky or X you would like included, please let us know.

19) Please provide a point-by-point letter INCLUDING my comments as well as the reviewer's reports and your detailed responses (as Word file).

I look forward to reading a new revised version of your manuscript as soon as possible.

Yours sincerely,

Poonam Bheda, PhD
Scientific Editor
Molecular Systems Biology

Reviewer #1:

The reviewer remains unconvinced on the annotation of "phosphor-degron" mechanism of SLC16A6 and believed stronger evidence should be provided if kinase and phosphatase are identified and found to affect SLC16A6 stability. Nonetheless, I support the publication of this systems level study.

Reviewer #2:

The authors have satisfactorily addressed all my concerns raised during the primary review. I therefore recommend the manuscript for publication and look forward to seeing it in press.

Reviewer #3:

The authors have made a sincere effort to address the comments, and while the response may not be in a perfectly ideal state as requested by the reviewer, the reviewer finds it reasonably sufficient.

Regarding the title of this manuscript, the reviewer fully understands the authors' perspective; however, from the readers' standpoint, some concerns remain.

Point-by-point response for the manuscript: The solute carrier superfamily interactome. MSB-2024-12688R

Response to reviewers of "The solute carrier superfamily interactome"

Thank you for the submission of your revised manuscript to Molecular Systems Biology. I am pleased to inform you that we will be able to accept your manuscript pending the following final amendments and appropriate response to reviewers:

Response to editorial changes

- 1) Please ensure that we have a functional email address for all authors. Currently the address for Jasper F. Ooms (j.f.ooms@umail.leidenuniv.nl) bounces back emails.

We have updated the contact information to: j.f.ooms@lumc.nl.

- 2) Please note that Reviewers 1 and 3 have some remaining concerns that should be addressed textually.

We addressed the comments of Reviewers #1 and #3 below.

- 3) Please upload the manuscript in .docx (or LaTeX) format with no track changes and no figures, only the figure legends should remain placed below the References.

We uploaded the manuscript as "*.docx".

- 4) Please format the Data availability section according to the example below: "The datasets and computer code produced in this study are available in the following databases: - Chip-Seq data: Gene Expression Omnibus GSE46748 (<https://www.ncbi.nlm.nih.gov/geo/query/acc.cgi?acc=GSE46748>) - Modeling computer scripts: GitHub (<https://github.com/SysBioChalmers/GECKO/releases/tag/v1.0>) - [data type]: [full name of the resource] [accession number/identifier] ([doi or URL or identifiers.org/DATABASE:ACCESSION])"

The data availability section was reformatted.

- 5) Data availability: The IM-30161 dataset in the IntAct database does not appear to be available, this needs to be rectified. Please ensure that all other deposited datasets are now publicly accessible, that reviewer access information is removed, and that specific URLs for each accession code are given as in the example above.

The PPI dataset is accessible via IntAct with the next release (1st of April 2025).

- 6) Please rename "Conflict of Interest" to "Disclosure and competing interests statement". We updated our journal's competing interests policy in January 2022 and request authors to consider both actual and perceived competing interests. Please review the policy <https://www.embopress.org/competing-interests> and update your competing interests if necessary.

We have revised the statement.

- 7) Our journal encourages inclusion of *data citations in the reference list* to directly cite datasets that were re-used and obtained from public databases. Data citations in the article text are distinct from normal bibliographical citations and should directly link to the database records from which the data can be accessed. In the main text, data citations are formatted as follows: "Data ref: Smith et al, 2001" or "Data ref: NCBI Sequence Read Archive PRJNA342805, 2017". In the Reference list, data citations must be labeled with "[DATASET]". A data reference must provide the database name, accession number/identifiers and a resolvable link to the landing page from which the data can be accessed at the end of the reference. Further instructions are available at <https://www.embopress.org/page/journal/17574684/authorguide#referencesformat>.

We have revised the data citations for published datasets.

- 8) In the Methods, please take care of the following:
- The Materials and Methods section should be renamed to "Methods".

We have revised the title to "Methods".

- Cell lines: As you have indicated in the Author Checklist, please also be sure to include a sentence in the Methods as to whether or not the cell lines were recently authenticated and tested for mycoplasma contamination.

We added a paragraph to the cell line generation section, referencing authentication and mycoplasma testing in the methods.

- Primers: please ensure that the actual primer sequences are included in Table EV5, and not links to the primer sequence via IDT.

For predesigned DsiRNAs by IDT, we only have the design IDs and cannot provide the sequences as they are proprietary by IDT and not provided. To clarify, we have added a column indicating for each DsiRNA whether it is custom or predesigned by IDT. For custom DsiRNAs, we have included sequences.

- Please ensure that a statement on whether or not blinding was done is included in the Methods even if no blinding was done. Please also be sure to update the Author Checklist with this information and where it can be found in the manuscript.

We added a statement in the methods addressing blinding.

- 9) Please place individual sections of the manuscript in the following order: Title page - Abstract & Keywords - Introduction - Results - Discussion - Methods - Data Availability - Acknowledgements - Disclosure and Competing Interests Statement - References - Figure Legends - Expanded View Figure Legends.

We organized the main manuscript accordingly.

- 10) Please remove the "Supplementary Information" section from the main manuscript file.

We removed the supplementary information from the manuscript.

11) For the figures and figure legends, please take care of the following:

- The callout for Appendix Fig. 6A,B should be corrected to Appendix Fig. S6A,B

We thank the editor for recognizing, we have corrected the callout as proposed.

- All figure callouts should be listed sequentially.

We referenced **Fig. EV2** before calling all subpanels of **Fig. EV1**. This was intentional, as **Fig. EV2** spans the entire figure space. Further, we reorganized the panels of **Fig. 6G-I** to follow the order.

- Please note that the exact p values are not provided in the legends of figures 2H, 4D, E; 5B, D; 6A, EV4 F

For **Fig. 2H** we added a statement regarding the p-value in the legend and in the methods. For **Fig. 4D, E** we updated the legend. We added a statement that the exact p-values are reported in **Dataset EV7**, updated column headers and added a statement in the methods. For **Fig. 6A**, we added the reference to **Dataset EV7**. For **Fig. 5B, Fig. 5D** and **Fig. EV4F** we exact p-values to the graphs.

- Please indicate the statistical test used for data analysis in the legends of figures EV3 E.

We provide the missing information in the figure caption of **Fig. EV3E**.

- Please indicate what */ **/ ***/ **** represents; if this represents p value(s), please indicate the exact p value in the legend(s) of figure(s) 2C, J; EV1 D, G; EV3 H

For panels **Fig. 2C,J, Fig. EV1D,G** and **Fig. EV3H** we added an explanation regarding the asterisk in the figure legends.

- Please note that the box plots need to be defined in terms of minima, maxima and percentile in the legends of figures 2A, C, I, J; EV1 F, G; EV3 H

We updated the figure legends for **Fig. 2A,C,I,J** and **Fig. EV1F-G** and **Fig. EV3H** regarding the boxplot. In addition to unify descriptions in figure legends, we updated the boxplot description in **Appendix Fig. S2A, S4A, S4B, S5B, S19**.

- Please note that the box plots need to be defined in terms of minima, maxima, centre, bounds of box and whiskers, and percentile in the legend of figure EV4 F

We updated the **Fig. EV4F** legend, which includes now statement regarding the box plot definition.

- Please note that information related to n is missing in the legends of figures 2A, I; 6C, EV1 F

We updated the figure legends of **Fig. 2A, I, J, Fig. 6C** and **Fig. EV1F** regarding the number of observations. To ensure consistency, we too updated the number of observations for **Fig. EV1G, Appendix Fig. S2A, S4A, S4B**.

- Please note that the error bars are not defined in the legend of figure EV1 D.

We added the following explanation to the legend for **Fig. EV1D**: "Data are presented as mean \pm SD."

- Please note that the measure of center for the error bars needs to be defined in the legend of figure 6D

We added to the caption of **Fig. 6D**: “bars denote mean \pm 95% confidence intervals”.

- 12) Tables: The labels and legends for Tables EV1-EV8 should not be uploaded as a separate tab/sheet, but above the table in each Excel file.

We have restructured **Tables EV1-EV8** to combine the column descriptions with the column headings.

- 13) Appendix file: Please upload the Appendix as a single PDF (no separate image files are needed) and add page numbers to the Table of Contents. In addition, please move the Results and Methods currently in the Appendix to the main manuscript file, and include the reagents in the Reagents and Tools table. There are no strict word limits for the main manuscript file and this information should be available to readers in the main manuscript.

We added page numbers to the table of content for appendix figures. We moved the results sections to different result sections (p.5, p.8, p.15). The appendix methods were moved to the main manuscript.

- 14) Source Data: In a routine image check, we have identified a potential issue with Figure EV4C that the blot may have been spliced together. Please provide the source data for this blot. If you make any changes to the figure or manuscript, please provide an explanation.

The figure panel **Fig. EV4C** was assembled from multiple western blots. We included a statement in the figure caption, that the blots are cropped and assembled. The uncropped blots are provided as source data.

- 15) Funding: Please ensure that all funding sources are entered in the Comments box in our submission system are included in the list of funders using the "More Funders" option.

All funding sources are included in the statement.

- 16) Synopsis:

- Synopsis image: We would suggest simplifying the synopsis image, as when resized to the requested 550 pixels wide x 300-600 pixels high, the text is quite small and difficult to read.
- Please check your synopsis text and image before submission with your revised manuscript.
- Please be aware that in the proof stage minor corrections only are allowed (e.g., typos).

We simplified the synopsis image, increased font sizes, and increased the contrast of the image.

- 17) As part of the EMBO Publications transparent editorial process initiative (see our policy here: https://www.embopress.org/transparent-process#Review_Process), Molecular Systems Biology will publish online a Peer Review File (PRF) to accompany accepted manuscripts. This file will be published in conjunction with your paper and will include the anonymous referee reports, your point-by-point response and all pertinent correspondence relating to the manuscript. Let us know whether you agree with the publication of the PRF and ask here, if you want to remove or

not any figures from it prior to publication. Please note that the Authors checklist will be published at the end of the PRF.

We agree to publish the PRF and wish to retain all content from the point-by-point answers.

18) After your paper is published, we will promote it on social media. If you have any handles or hashtags for Bluesky or X you would like included, please let us know.

X: “@gsf_lab”, “@giuliosf”, “@CeMM_News”, “@RESOLUTE_IMI”, “@oeaw”; Bluesky: “@supertifurgalab.bsky.social”, “@giuliosupertifurga.bsky.social”, “@oeaw.bsky.social”, “@ihieurope.bsky.social”.

19) Please provide a point-by-point letter INCLUDING my comments as well as the reviewer's reports and your detailed responses (as Word file).

We have included this file that addresses all points.

Reviewer #1:

The reviewer remains unconvinced on the annotation of "phosphor-degron" mechanism of SLC16A6 and believed stronger evidence should be provided if kinase and phosphatase are identified and found to affect SLC16A6 stability. Nonetheless, I support the publication of this systems level study.

We wish to express our gratitude to the reviewer for their constructive feedback and support in the publication of our system-wide SLC-interactome study.

In the previous revision, we transparently addressed the mixed results of mimicking S240 phosphorylation. We amended the results to reflect the uncertainty of the phospho-degron mechanism and referenced a similar S384 substitution experiment for YAP. Like our SLC16A6 substitutions, S384 substitution in YAP led to loss of SCF complex binding.

We searched for kinases in our AP-MS dataset but did not find strongly enriched kinases in the SLC16A6 experiment due to AP-MS capturing mostly stable interactions. Proximity labeling is more suitable for transient interactions. Additional BioID experiments are beyond our study's scope, but we believe this study will help identify the corresponding kinase in a targeted approach.

Reviewer #2:

The authors have satisfactorily addressed all my concerns raised during the primary review. I therefore recommend the manuscript for publication and look forward to seeing it in press.

We sincerely appreciate the reviewer's support in the publication of our manuscript.

Reviewer #3:

The authors have made a sincere effort to address the comments, and while the response may not be in a perfectly ideal state as requested by the reviewer, the reviewer finds it reasonably sufficient.

Regarding the title of this manuscript, the reviewer fully understands the authors' perspective; however, from the readers' standpoint, some concerns remain.

We thank the reviewer for recognizing our effort to address their comments and for supporting the publication of our manuscript.

We emphasize that the SLC-interactome is a multi-year effort, surpassing previous datasets in size and coverage. A fundamental characteristic of interactomics— regardless of the interaction mapping technology employed — is that it leads to non-exhaustive PPI-networks. Therefore, we firmly believe that the title accurately reflects the scope and significance of the presented work.

11th Apr 2025

Manuscript number: MSB-2024-12688RR

Title: The solute carrier superfamily interactome

Dear Dr. Superti-Furga,

Congratulations on an excellent manuscript, I am pleased to inform you that your manuscript has been accepted for publication in Molecular Systems Biology. Thank you for your comprehensive response to referee concerns. It has been a pleasure to work with you to get this to the acceptance stage.

Yours sincerely,

Poonam Bheda, PhD
Scientific Editor
Molecular Systems Biology
